# The motor neuron m6A repertoire governs neuronal homeostasis and FTO inhibition mitigates ALS symptom manifestation

Ya-Ping Yen [1] ✉, Ting-Hsiang Lung [1], Ee Shan Liau [1], Chuan-Che Wu [1], Guan-Lin Huang [1], Fang-Yu Hsu [1], Mien Chang[1], Zheng-Dao Yang[2], Chia-Yi Huang[2], Zhong Zheng[3], Wei Zhao [4], Jui-Hung Hung [2], Chuan He [3,5,6], Qing Nie[4] & Jun-An Chen [1] ✉

Amyotrophic lateral sclerosis (ALS) is a swiftly progressive and fatal neuro-degenerative ailment marked by the degenerative motor neurons (MNs). Why MNs are specifically susceptible in predominantly sporadic cases remains enigmatic. Here, we demonstrated $N^6$-methyladenosine (m6A), an RNA modification catalyzed by the METTL3/METTL14 methyltransferase complex, as a pivotal contributor to ALS pathogenesis. By conditional knockout *Mettl14* in murine MNs, we recapitulate almost the full spectrum of ALS disease characteristics. Mechanistically, pervasive m6A hypomethylation triggers dysregulated expression of high-risk genes associated with ALS and an unforeseen reduction of chromatin accessibility in MNs. Additionally, we observed diminished m6A levels in induced pluripotent stem cell derived MNs (iPSC~MNs) from familial and sporadic ALS patients. Restoring m6A equilibrium via a small molecule or gene therapy significantly preserves MNs from degeneration and mitigates motor impairments in ALS iPSC~MNs and murine models. Our study presents a substantial stride towards identifying pioneering efficacious ALS therapies via RNA modifications.

In recent years, the intricate world of RNA modifications, often referred to as the "epitranscriptome," has emerged as a pivotal regulatory axis governing developmental processes[1]. Over 170 types of RNA modifications have been identified since the 1950s. High-throughput sequencing has revealed a diversity of messenger RNA (mRNA) modifications in various organisms[2,3]. Among these, m6A has received the most attention as it is the most abundant form of mRNA modification in mammals[4]. Various studies have shown that m6A levels can be dynamic and reversible. m6A is deposited by the m6A methyltransferase complex (termed the "writer"), with METTL3 (methyltransferase-like 3) acting as the catalytically active methyltransferase and METTL14 playing an essential structural role in facilitating catalysis. The larger methyltransferase holo complex contains accessory units, including WTAP (Wilms tumor 1-associated protein), VIRMA (vir-like m6A methyltransferase associated), RBM15 (RNA-binding motif protein 15) and its paralog (RBM15B), ZC3H13, and HAKAI[5–11]. Conversely, m6A marks are subject to removal by m6A demethylases, aptly termed "erasers". Notable players in this dynamic process include FTO (fat mass and obesity-associated protein) and ALKBH5 (alkB homolog 5)[12–15]. This intricate interplay between writers and erasers crafts a finely tuned regulatory mechanism, orchestrating the reversible m6A

[1]Institute of Molecular Biology, Academia Sinica, Taipei, Taiwan. [2]Department of Computer Science, National Yang Ming Chiao Tung University, Hsinchu City, Taiwan. [3]Department of Chemistry and Institute for Biophysical Dynamics, University of Chicago, Chicago, IL, USA. [4]Department of Mathematics, NSF-Simons Center for Multiscale Cell Fate Research, Department of Developmental and Cell Biology, University of California, Irvine, Irvine, CA, USA. [5]Howard Hughes Medical Institute, Chicago, IL, USA. [6]Department of Biochemistry and Molecular Biology, University of Chicago, Chicago, IL, USA.
✉ e-mail: yapingyen@gate.sinica.edu.tw; jac2210@gate.sinica.edu.tw

modifications that play critical roles in shaping RNA function and developmental processes.

The m[6]A RNA modification acts on multiple molecular pathways, including in splicing, stability, nuclear export, localization, translational efficiency, and activation and decay of targeted mRNAs[5,7,16,17]. Recent studies have shown that constitutive knockout of *Mettl14*—a key facilitator of the m[6]A methyltransferase complex—is embryonically lethal in mice, whereas conditional knockout (cKO) of *Mettl14* in neural progenitor cells disrupts cortical development and leads to premature death in mice[18,19]. Remarkably, levels of m[6]A are relatively low in mouse brain tissue during embryogenesis, but drastically increase by adulthood[20], suggesting that m[6]A RNA modification plays a unique role in the adult central nervous system. That latter finding has also raised the possibility that m[6]A might play an important role in adult RNA homeostasis, with imbalances potentially leading to the onset or progression of neurodegeneration. This hypothesis is supported by studies demonstrating a positive correlation between m[6]A modification and gene expression homeostasis across tissues, as well as tissue-type-specific aging-associated m[6]A dynamics in primates[21]. In humans, functional impairment of m[6]A has also been shown to play a pivotal role in cancer[22,23], cell fate transition and determination[24,25], and disease[26–28]. Although a concordant decrease in m[6]A RNA methylation of brain tissue (specifically, the cingulate gyrus) from an Alzheimer's disease (AD) mouse model and in human patient brain tissues has been reported[29], whether m[6]A exerts a direct causative role in human neurodegeneration remains obscure.

To date, research efforts have focused almost exclusively on DNA sequencing (usually whole exome) to identify the genetic causes of neurodegenerative diseases. This is arguably the primary reason why the genetic and molecular bases for many neurodegenerative diseases remain unknown, as most neurodegenerative diseases are sporadic[30,31]. Accumulating evidence indicates that most aging-associated diseases, including amyotrophic lateral sclerosis (ALS), are linked to RNA metabolism, perhaps explaining why probing gene mutations by DNA sequencing fails to identify more ALS-causative genes[32,33]. Though dysregulated RNA processing has been identified in the majority of ALS patients, it remains unclear which aspects of RNA metabolism are critical and if they are directly causative of spinal motor neuron (MN) degeneration[31,34]. There are two major shortcomings of ALS-associated research efforts to date: (1) next-generation sequencing technologies are DNA-based and cannot directly sequence RNA or RNA isoforms with long reads and modifications, hindering analyses of RNA modifications from patient transcriptomes; and (2) only a small number of mouse models mimic to varying degrees the MN pathology of sporadic ALS (sALS), with most of them presenting relatively minor phenotypes when compared to familial ALS (fALS) models[30]. Accordingly, there is still no robust sALS animal model (>90% of ALS patients are sporadic) that fully recapitulates MN degeneration pathology. Although employing a gene mutation identified in ALS patients to generate an ALS murine model remains a robust methodology, it frequently only elicits some aspects of ALS pathology, occasionally resulting in excessively shortened lifespan[30]. Therefore, to advance research in this field necessitates: (1) discovering common disease-causing mechanisms present in both familial and sporadic ALS patients; and (2) establishing an ALS animal model based on these mechanisms. Such efforts would aim to replicate the primary hallmarks of familial and sporadic ALS at molecular (e.g., TARDBP/TDP43 aggregation), cellular (e.g., motor neuron degeneration), and physiological levels (manifesting muscle weakness and shortened lifespan).

Recently, two studies emphasized the roles of m[6]A in ALS. Barmada's group reported that m[6]A hypermethylation modulates RNA binding by TDP43 and the disease pathogenesis of ALS and frontotemporal dementia (FTD)[35], whereas Sun's group indicated that globally reduced m[6]A levels in C9ORF72-associated ALS and FTD dysregulate RNA metabolism and contribute to neurodegeneration[36].

Although these two studies emphasize the importance of m[6]A homeostasis in ALS, their seemingly contradictory results necessitate further clarification. As both studies were performed primarily on cell models and postmortem spinal cord sections from patients, it is imperative to explore if manipulating m[6]A levels in animal models in vivo recapitulates ALS pathology.

Here, we used two different sets of MN Cre drivers to remove *Mettl14*, corroborating that m[6]A hypomethylation (hypo-m[6]A) elicits an ALS-like phenotype in vivo. Impairment of the m[6]A repertoire elicits dysregulation of many known ALS-related pathways, with a concomitant change in the chromatin landscape of spinal MNs. Additionally, iPSC-MNs from several familial and sporadic ALS patients exhibited hypo-m[6]A, and restoration of m[6]A homeostasis by means of a small molecule largely spared the MNs from degeneration in both familial and sporadic ALS contexts. Most importantly, intrathecal delivery of *Fto*-shRNA to knock down *Fto*, an m[6]A eraser enzyme, ameliorated the motor deficits of *SOD1^{G93A}* mice (an ALS mouse model) and extended their lifespan. Accordingly, we speculate that m[6]A hypomethylation contributes to ALS and restoring the m[6]A reservoir can mitigate the symptoms of familial and sporadic ALS.

## Results

### m[6]A levels are reduced in human ALS iPSC-MNs and hypo-m[6]A leads to MN degeneration

Given existing contradictory results[35,36], we examined the extensive transcriptomic dataset derived from Answer ALS to establish if m[6]A hypermethylation (hyper-m[6]A) or hypo-m[6]A is associated with ALS. Our analysis focused on assessing expression levels of the methyltransferases *METTL3/5/14/16*, in which METTL3/14 are the 'writer' complex responsible for m[6]A for mRNAs, while METTL16 is largely for ncRNA and METTL5 is known for rRNA m[6]A modifications[3] (Supplementary Fig. 1). Interestingly, only *METTL3/14* exhibited a trend of down-regulation in numerous familial or sporadic human ALS iPSC-MNs (Supplementary Fig. 1a, b). In agreement with this outcome, we observed reduced *METTL3* and *METTL14* expression in the majority of postmortem cortex samples from ALS patients (Supplementary Fig. 1c), as well as reduced protein levels as examined by Li et al.[36]. To confirm this result, we assessed another independent study that conducted transcriptome analysis on a different set of human ALS iPSC-MNs and observed that most of those iPSC-MNs also exhibited consistent down-regulation of *METTL3* and *METTL14* (Supplementary Fig. 1d). Together, these findings indicate that compromised m[6]A pathways might be an overlooked aspect of ALS. To scrutinize if compromised m[6]A writer expression leads to hypo-m[6]A in a human context, we differentiated three familial ALS iPSC lines (*SOD1^{+/L144F}*, *C9ORF72^{exp-800 G4C2}*, and *TDP43^{G298S}*), together with their corresponding isogenic rescue controls (Ctrl #1 and Ctrl #2) and healthy control (Ctrl #3), into spinal MNs, and then profiled their m[6]A dynamics along the differentiation process under stress-induced conditions (illustrated in Fig. 1a, results in Fig. 1b–g)[37,38]. To accelerate ALS disease progression, we applied cyclopiazonic acid (CPA), an endoplasmic reticulum stressor, as CPA has been shown previously to act as a selective stressor to accelerate the degeneration of human *SOD1^{G93A}* iPSC-MNs but not wild-type controls[37,38]. Consistent with this scenario, we found that all of our Ctrl iPSC-MNs were relatively resistant to CPA stress, unlike the ALS iPSC-MNs that exhibited drastic loss after seven days of CPA treatment (Fig. 1b–d). No obvious degeneration was displayed by either Ctrl or ALS iPSC-MNs on day 4 (Fig. 1b–d). Thus, we could capture the progressive MN degeneration displayed by the familial ALS iPSC lines. Notably, even before the drastic MN loss following stress treatment on day 4, we consistently detected reduced m[6]A levels in the ALS iPSC-MNs (Fig. 1b–d). This trend was sustained at day 7 with a concomitant decrease in *METTL3/14* expression, together with significantly increased expression of the demethylases *FTO* and *ALKBH5* in most of the ALS iPSC-MNs (Supplementary Fig. 1e). In agreement

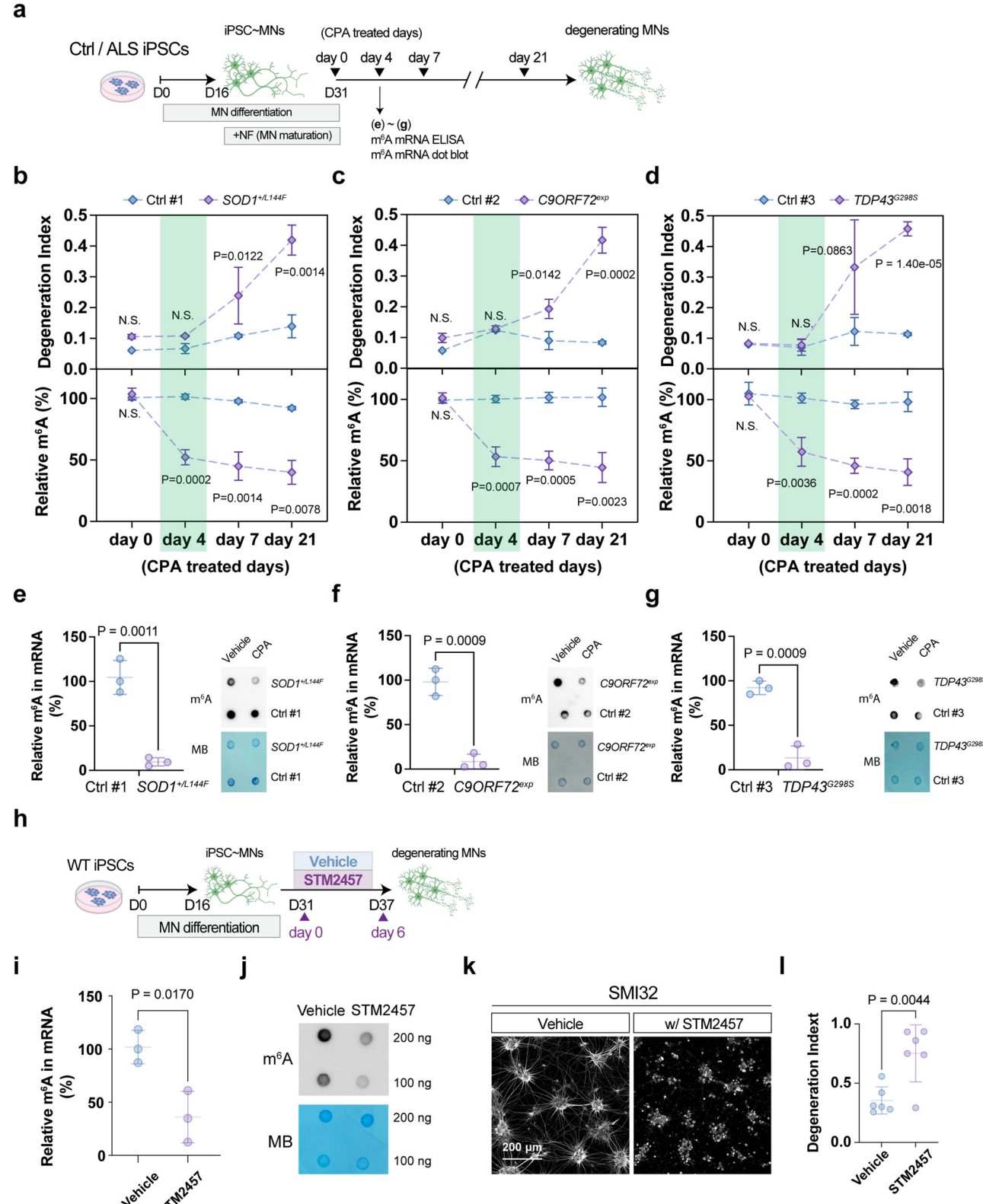

with these data, we found that all three ALS iPSC lines displayed hypo-m⁶A before the MN loss caused by stressor treatment (Fig. 1e–g) revealed by m⁶A ELISA and dot blot (see methods for details).

To confirm if hypo-m⁶A leads to MN degeneration, we adopted two approaches. First, we used a specific METTL3 inhibitor (METTL3i), STM2457[22], to impair m⁶A production during human MN differentiation (Fig. 1h), which revealed that METTL3i reduces the m⁶A-mRNA

repertoire assayed by m⁶A ELISA and dot blot (Fig. 1i, j), with con-comitantly drastic neurite degeneration and a reduced MN population (Fig. 1k, l). Secondly, we transfected the HEK293T with *METTL3 or METTL14*-shRNA (Supplementary Fig. 2a) and revealed a reduction in m⁶A-mRNA levels (Supplementary Fig. 2b). Furthermore, we infected the Lentivirus (LV)-sh*METTL3* and sh*METTL14* observed a concomitant neurite degeneration (Supplementary Fig. 2c–e). Thus, our results

**Fig. 1 | m⁶A RNA modification levels are downregulated in ALS; loss of m⁶A methyltransferases *METTL3* and *METTL14* leads to neurodegeneration.**
**a** Timeline of MN differentiation in control (Ctrl) and ALS ($SOD1^{+/L144F}$, $C9ORF72^{exp-800 G4C2}$, $TDP43^{G298S}$) iPSC lines. CPA (cyclopiazonic acid) was applied to accelerate MN degeneration (annotated as basal time point day 0). NF neurotrophic factors. **b–d** MN degeneration index and m⁶A RNA methylation levels of Ctrl and ALS iPSC-MNs. m⁶A percentages of Ctrl/ALS types at indicated time points were normalized to those at day 0. ALS iPSC-MNs undergo dramatic degeneration from day 7 to day 21 post-CPA treatment, while a significant reduction in m⁶A level was already exhibited at day 4. The degeneration index measures the neurite fragmentation. **e–g** Quantification of the m⁶A ratio in mRNAs from day 4 post-CPA treatment of Ctrl and ALS ($SOD1^{+/L144F}$, $C9ORF72^{exp-800 G4C2}$, $TDP43^{G298S}$) iPSC lines. The mRNAs were extracted by poly(A) purification. The panels at the right show representative images of an m⁶A dot blot and methylene blue staining (for loading controls) (**b–g:** $n = 3$ independent experiments). **h** Timeline of STM2457 (20 μM)-mediated inhibition of m⁶A modification during MN differentiation of wild-type iPSC lines. STM2457 was applied for 6 days to accelerate MN degeneration. **i** and **j** Inhibition of METTL3-mediated m⁶A modification in human wild-type iPSC-MNs results in a sharp decline in m⁶A levels after four days of treatment for STM2457, as assayed by m⁶A ELISA in mRNA (**i**) and mRNA m⁶A dot blot (**j**) (**i, j:** $n = 3$ independent experiments). m⁶A methylation was normalized to the vehicle (DMSO), which served as a non-stressed control. Note that dramatic neurite degeneration was observed on day 6 in **k** upon STM2457 treatment. Scale bar, 200 μm. **l** Quantification of the degeneration index at an indicated time point normalized to the vehicle control ($n = 6$ independent experiments). Illustrations in **a** and **h** created in BioRender. Chen, J. (2025) https://BioRender.com/3nizfu7. All Data are presented as mean ± SD, significant *P* values from two-tailed *t*-tests. N.S. non-significant. Source data are provided as a Source data file.

---

together with the large set of available ALS patient data support the notion that METTL3/METTL14 might be a critical regulatory complex linked to both familial and sporadic ALS disease, and that MNs appear to be more sensitive to *METTL3/METTL14* down-regulation. Most notably, global down-regulation of the m⁶A methylation repertoire elicited obvious MN degeneration.

### Impairment of the m⁶A production enzyme Mettl14 in spinal MNs elicits MN degeneration

To determine if hypo-m⁶A promotes MN degeneration in vivo, we conditionally deleted *Mettl14* specifically in MNs either at the developmental stage by using *Olig2-Cre*[39] or at the postmitotic and postnatal stage by adopting *ChAT-Cre* (see Methods for details)[18]. First, we verified that both conditional mouse lines displayed a significant reduction in the population of Mettl14ᵒⁿ cells in the ventral horn of the spinal cord (Supplementary Fig. 3a, b). Although *Olig2-Cre; Mettl14ᶠˡᵒˣᵉᵈ* mice mostly exhibited early postnatal lethality (~P24 to P28, Supplementary Fig. 3c) with a shivering phenotype (Supplementary Movie 1), all MN subtypes appeared normal based on immunostaining (Supplementary Fig. 3d–g). As Olig2 is known to be expressed at a later stage in oligodendrocytes, we consider that the shivering phenotype might be a reflection of compromised oligodendrocyte precursors (Olig2ᵒⁿ and Sox9ᵒⁿ double-positive cells) (Supplementary Fig. 3h, i), consistent with a previous study[40]. Conversely, the *ChAT-Cre; Mettl14ᶠˡᵒˣᵉᵈ* mice displayed normal MN development and an ordinary appearance at the postnatal and juvenile stages (Fig. 2a). Nevertheless, two months later, we observed a gradual decline in body weight from P70 (Fig. 2a) and a kyphosis phenotype from P100 for all of the *ChAT-Cre; Mettl14ᶠˡᵒˣᵉᵈ* mice (Supplementary Fig. 3j). Strikingly, all of the *ChAT-Cre; Mettl14ᶠˡᵒˣᵉᵈ* mice (both male and female, $n > 90$) exhibited premature death at P160-P300 (Fig. 2b).

Apart from their kyphotic appearance and movement defects, we further investigated a series of molecular and cellular ALS disease features in the *ChAT-Cre; Mettl14ᶠˡᵒˣᵉᵈ* mice. At the molecular level, we observed (1) that the numbers of ChATᵒⁿ MNs in the lumbar region of spinal cords were comparable before P70 but gradually declined after P100 (Fig. 2c, d). However, C-boutons, a source of cholinergic input to MNs, already showed a prominent decrease from P70 (Fig. 2e, f); (2) prominent neuroinflammation upon microglia (Iba1ᵒⁿ) activation in the spinal cords of the *ChAT-Cre; Mettl14ᶠˡᵒˣᵉᵈ* mice relative to controls at P160 (Fig. 3a, b), but not before P120 (Supplementary Fig. 4a, b); and (3) significant cytoplasmic aggregation of Tdp43 in the *ChAT-Cre; Mettl14ᶠˡᵒˣᵉᵈ* mice, whereas control littermates mainly presented nuclear localizations for that protein after P120 (Fig. 3c, d, and Supplementary Fig. 4c, d). Notably, another RNA-binding protein, Fus, which is often shown as mislocalization in ALS patients, also exhibited cytoplasmic aggregation in the *ChAT-Cre; Mettl14ᶠˡᵒˣᵉᵈ* mice (Supplementary Fig. 4e, f). Furthermore, the *ChAT-Cre; Mettl14ᶠˡᵒˣᵉᵈ* mice exhibited drastically reduced endplate area and muscle denervation (Fig. 3e–h). Overall, our findings indicate that the *ChAT-Cre; Mettl14ᶠˡᵒˣᵉᵈ* mouse

model demonstrates progressive MN degeneration, mirroring several key molecular pathological features observed in human ALS patients.

### Mice with m⁶A hypomethylation recapitulate ALS-associated behavioral phenotypes

Although some previous ALS mouse models exhibit molecular hallmarks of ALS pathology, their MN physiology or gross behaviors appeared relatively normal[30,34]. To examine if the *ChAT-Cre; Mettl14ᶠˡᵒˣᵉᵈ* mice represent an improved potential ALS mouse model, we conducted a series of behavioral assays on the *ChAT-Cre; Mettl14ᶠˡᵒˣᵉᵈ* mice to further characterize the phenotypes observed from P40 to P210. First, the results of a rotarod test and forelimb grip strength assay corroborated that motor ability gradually declined, together with concomitant muscle weakness, recapitulating two major pathological manifestations observed for ALS patients (Fig. 4a–d, Supplementary Movie 2). Next, through an open field test, we noted that general activity levels of the *ChAT-Cre; Mettl14ᶠˡᵒˣᵉᵈ* mice gradually became compromised and their exploratory behavior in a novel environment was reduced, reflecting a motor deficit and a frontotemporal dementia (FTD)-like phenotype (Fig. 4e, f, and Supplementary Fig. 5). Finally, we performed a kinematic analysis, which revealed that whereas the spinal interneuron (IN) circuit remained largely intact, motor outputs were interrupted in the *ChAT-Cre; Mettl14ᶠˡᵒˣᵉᵈ* mice (Fig. 4g-I, Supplementary Fig. 5a, and Supplementary Movie 3). This detailed scrutiny of the *ChAT-Cre; Mettl14ᶠˡᵒˣᵉᵈ* mice at the molecular, cellular, physiological, and behavioral levels indicates that the m⁶A reservoir is a critical factor in maintaining adult MN function, and that compromising m⁶A levels prompts a MN degeneration process that recapitulates ALS disease progression.

### Uncovering the dysregulated m⁶A-modified genes leading to MN degeneration

To gain insights into the potential mechanisms underlying how hypo-m⁶A promotes MN degeneration, we aimed to systematically identify the dysregulated genes possessing m⁶A modifications in our *ChAT-Cre; Mettl14ᶠˡᵒˣᵉᵈ* mice (Figs. 5, 6). To identify m⁶A-modified transcripts, we adopted a direct RNA sequencing platform, which enables the identification of the MN m⁶A epitranscriptome at single-nucleotide resolution (Fig. 5). Since it is technically challenging to obtain sufficient adult MNs from the mouse spinal cord for direct RNA sequencing, we employed an enhanced method using mouse ESC-derived MNs (mESC-MNs), matured with a conditioned medium (Fig. 5a–c)[38,41]. First, we confirmed that this approach successfully generated MNs expressing mature neuronal markers, with longer and more mature neurite structures revealed by discrete Syn1-positive puncta (Fig. 5b, c). We then subjected these mature MNs to the Oxford Nanopore Technologies (ONT) platform, which provides a powerful framework for detecting RNA modifications through advanced machine-learning algorithms applied to sequencing metrics (Fig. 5d, details in Methods). As expected, several adult mature MN

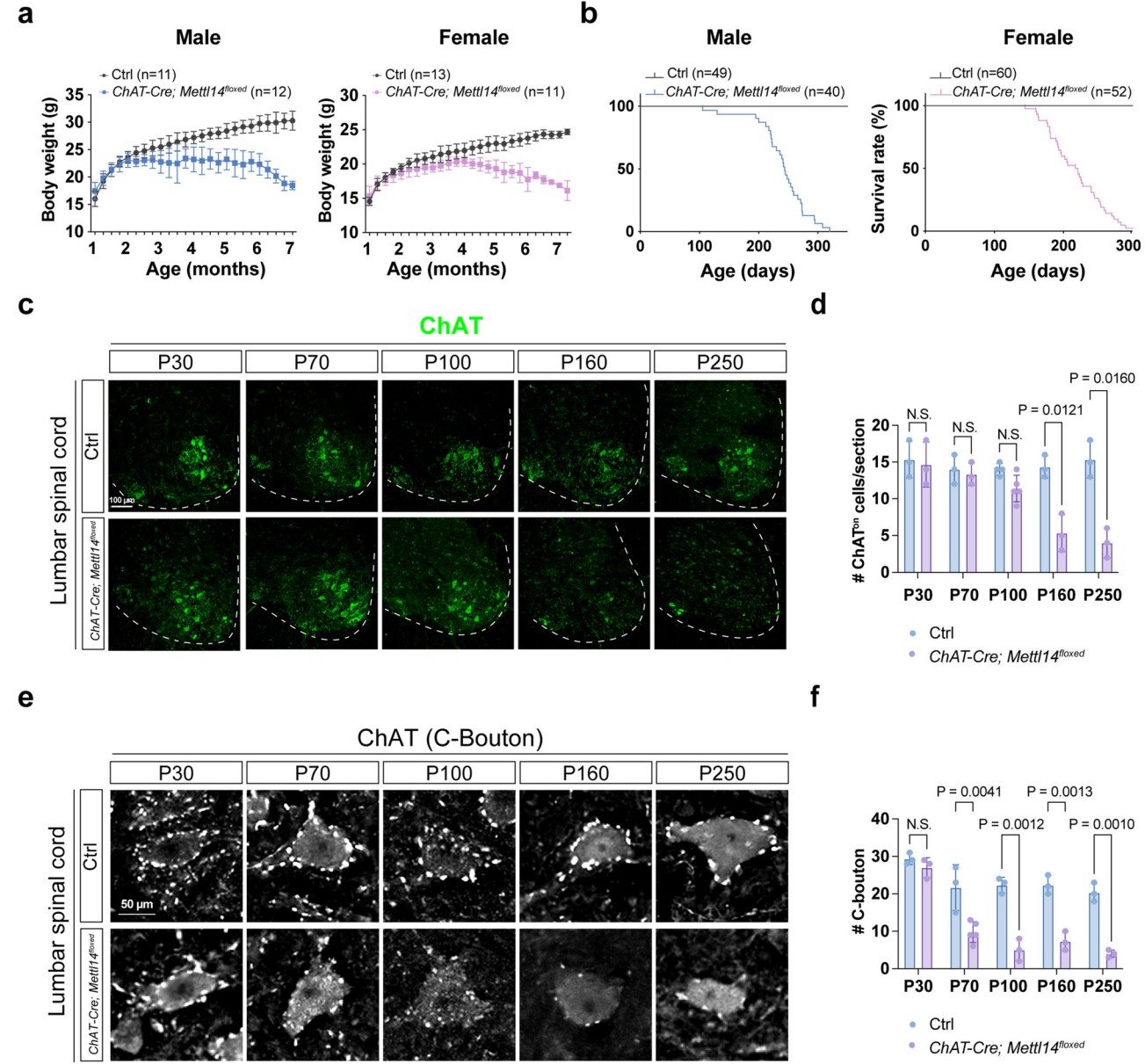

**Fig. 2 | Phenotypic characterization of *ChAT-Cre; Mettl14^floxed* mutant mice.**
**a** Body weight of male and female *ChAT-Cre; Mettl14^floxed* and littermate control mice. Data are presented as mean ± SD. **b** Kaplan–Meier survival curves reflect that both male and female *ChAT-Cre; Mettl14^floxed* mice die prematurely compared to littermate controls. Immunostaining (**c**) and quantification (**d**) of lumbar ChAT^on MN numbers reveal a gradual reduction starting after P100 and a significant loss of *ChAT-Cre; Mettl14^floxed* MNs at P160 (Data are presented as mean ± SD, with significant *P* values from two-tailed *t*-tests. NS non-significant. *n* = 3 in *ChAT-Cre;*

*Mettl14^floxed* and control mice at P30, P70, P160, and P250, respectively; *n* = 5 in *ChAT-Cre; Mettl14^floxed* mice and *n* = 4 in control mice at P100, Scale bar, 100 μm). Preferential loss of the cholinergic C-bouton nerve terminals of MNs in the *ChAT-Cre; Mettl14^floxed* mice from P70 (**e**), with respective quantification in (**f**). Scale bar, 50 μm. Data are presented as mean ± SD, *n* = 3 mice in *ChAT-Cre; Mettl14^floxed* and control at P30, P100, P160, and P250, respectively; *n* = 5 in *ChAT-Cre; Mettl14^floxed* mice and *n* = 3 in control mice at P70, with significant *P* values from two-tailed *t*-tests. N.S. non-significant. Source data are provided as a Source data file.

genes are abundantly expressed in the Nanopore direct RNA-seq results (Fig. 5e). We subsequently employed two supervised machine learning tools, namely EpiNano and m6Anet[42,43], which set a stringent criterion that the predicted m6A sites need to occur in at least two samples for either one of the algorithms (see Methods for details). We identified 30,340 high-confidence m6A modification sites (probability > 0.5) corresponding to 7,921 genes (Fig. 5d, genes of interest are illustrated in Supplementary Data 1; refer to the Methods for details). Interestingly, a deeper estimate of the predicted m6A stoichiometry indicated that the high variation levels are displayed across different m6A sites in the same gene (Supplementary Data 3). Consistent with previous findings[29], our analysis also revealed

enriched distributions of m6A sites in coding sequences (CDSs) and 3′ UTRs, especially near the stop codons (Fig. 5f and Supplementary Fig. 6a). By analyzing the enrichment of Gene Ontology (GO) and KEGG pathways for the m6A-modified transcripts, we noticed a striking enrichment for ALS-related genes (Fig. 5g). In addition, although only 35.56% of transcripts (7921 out of 22,280) contained m6A in our dataset, among the 81 identified ALS risk genes to date, our Nanopore direct RNA-seq demonstrated that 41.98% of ALS risk genes (34 out of 81) are m6A-modified ($P = 2.06e\text{-}6$, one-tailed hypergeometric test; Fig. 5h and Supplementary Data 2).

To validate our computationally predicted m6A sites, we used *Tardbp* (*Tdp43*) as a benchmark to validate our methodology.

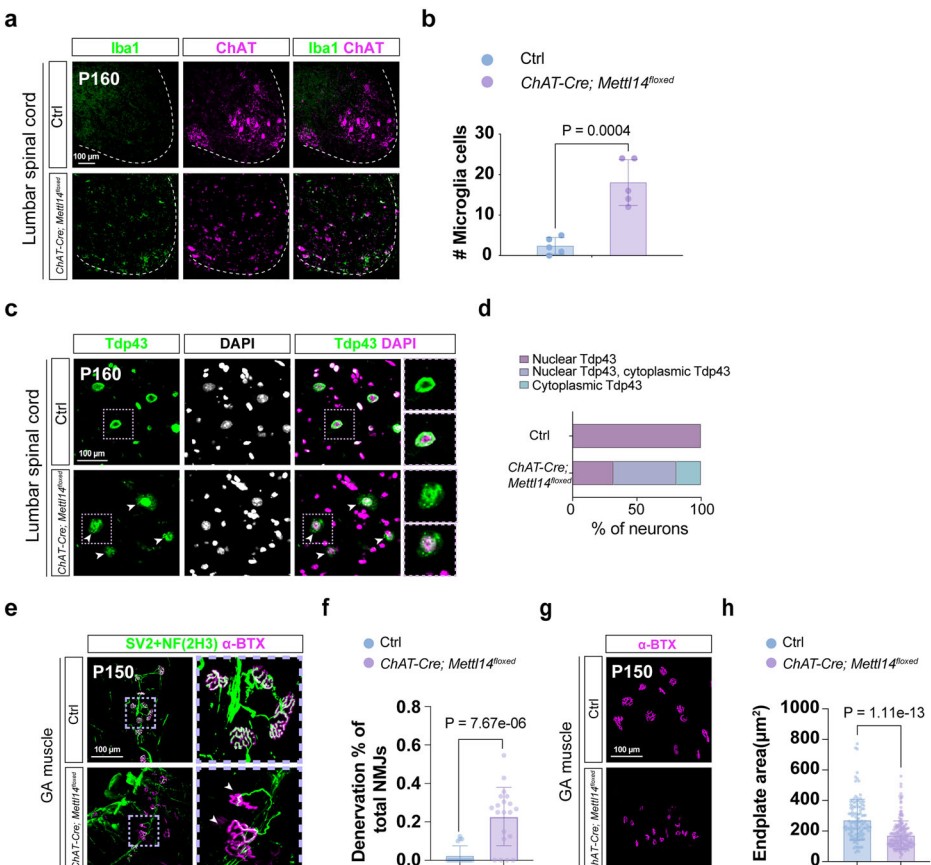

**Fig. 3 | Molecular characterization of *ChAT-Cre; Mettl14^floxed* mutant mice.** Images illustrate microglial activation, as determined by immunostaining for Iba1 (**a**), and quantification (**b**) of lumbar Iba1^on numbers at the ventral region, revealing significant microglial activation in *ChAT-Cre; Mettl14^floxed* mice compared to littermate controls. Staining was repeated on *n* = 5 mice. **c** Tdp43 (green) is localized in the nucleus of the MNs of control mice. In the *ChAT-Cre; Mettl14^floxed* mutant mice, numerous Tdp43 inclusions exist in the cytoplasm (arrow). High magnifications of the highlighted Tdp43 aggregates in MNs are shown in the rightmost panels. Respective quantification is presented in **d**, (*n* = 3 mice). **e, g** Representative z-stack

confocal images of neuromuscular junctions (NMJs) in gastrocnemius (GA) muscles dissected from P150 *ChAT-Cre; Mettl14^floxed* and littermate control mice. Motor nerves were visualized using a combination of SV2/NF(2H3) (green) and postsynaptic AChRs with α-BTX (magenta). Arrowheads identify denervated synapses, abnormal axonal swellings (**e**), and smaller endplates (**g**). **f, h** Quantification of the denervation ratio of NMJs and endplate area from (**e, g**). (*n* = 3 mice, quantification for all NMJs from all views of captured images). Scale bar, 100 μm. Data are presented as mean ± SD with significant *P* values from two-tailed *t*-tests. Source data are provided as a Source data file.

Consistent with a previous report[35], we confirmed the existence of a previously identified m⁶A site in *Tardbp* with a high probability rate with stoichiometry ranging from 57.38 to 92.53% (Supplementary Data 3) and verified via m⁶A antibody pull-down assay (Fig. 5i, site 1). Additionally, we uncovered and validated an additional high m⁶A-modified site within the *Tardbp* transcript from Nanopore direct RNA-seq (Fig. 5i, site 2). Moreover, we substantiated the existence of predicted m⁶A sites in *Atp13a2* (*Park9*) (Fig. 5j)[44], an ALS risk gene not previously shown to have m⁶A modifications in MNs. Finally, we further verified several newly identified m⁶A-modified sites in ALS risk genes, including *Dctn1, Epha4, C9orf72, Glt8d1, Cacna1h, Chrna3, Bscl2, Fig4, Hnrnpa2b1, Ubqln2, Hnrnpa1, Tuba4a, Sod1, Chmp2b,* and *PIKfyve* (Supplementary Fig. 6b). Thus, these observations confirm the sensitivity, accuracy, and reliability of Nanopore technology to identify m⁶A-modified sites in MNs.

Next, we reasoned that most m⁶A-modified ALS risk genes could potentially contribute to the observed ALS-like pathologies upon m⁶A impairment in the *ChAT-Cre; Mettl14^floxed* mice, so we probed the consequence of hypo-m⁶A for MNs by performing 10x Genomics single-nuclei multimodal profiling of ATAC/RNA (snATAC/RNAseq) on Ctrl (*ChAT-Cre; Mettl14^f/+*) and *ChAT-Cre; Mettl14^floxed* mice, allowing us to assess chromatin accessibility and gene dysregulation in a range of MN subtypes simultaneously (Fig. 6a, b). To selectively enrich for nuclei

from spinal cholinergic neurons, we bred *ChAT-Cre; Mettl14^floxed* mice expressing the nuclear envelope reporter CAG-Sun1/sfGFP[45] (Fig. 6b). We harvested lumbar spinal cords at P100-120, a stage when the MN population is not greatly diminished, and collected GFP^on cells by fluorescence-activated cell sorting (FACS) for single-nuclei multimodal profiling (Fig. 6c and Supplementary Fig. 7a). Each sample underwent rigorous quality control (QC) measures and was subsequently filtered to retain only cells that only met our QC criteria (Supplementary Fig. 7a) (see Methods for details). We then integrated three replicates from the Ctrl and *Sun1^sfGFP; ChAT-Cre; Mettl14^floxed* (KO) samples (Supplementary Fig. 7b, c). Principal component analysis (PCA) revealed negligible sequencing and batch confounding variables among our sample preparations (Supplementary Fig. 7d). PC1 and PC2 largely separated the major cell types in the population, and PC3 segregated the total population based on whether the cells were from the Ctrl or *Sun1^sfGFP; ChAT-Cre; Mettl14^floxed* cohorts. Using recognized markers for spinal cord cell type annotation[45,46], we detected the three major cholinergic cell populations, i.e., skeletal MNs (*Tns1^on/Bcl6^on*), visceral MNs (*Nos1^on*), and cholinergic INs (*Pax2^on*) (Fig. 6d and Supplementary Fig. 7e). We did not observe any changes in the proportions of these major cell types, consistent with our in vivo characterization of the pre-onset stage of *ChAT-Cre; Mettl14^floxed* mice (Fig. 6c, d). When we further analyzed subtypes within the skeletal MNs, we identified α (*Htr1d^low*,

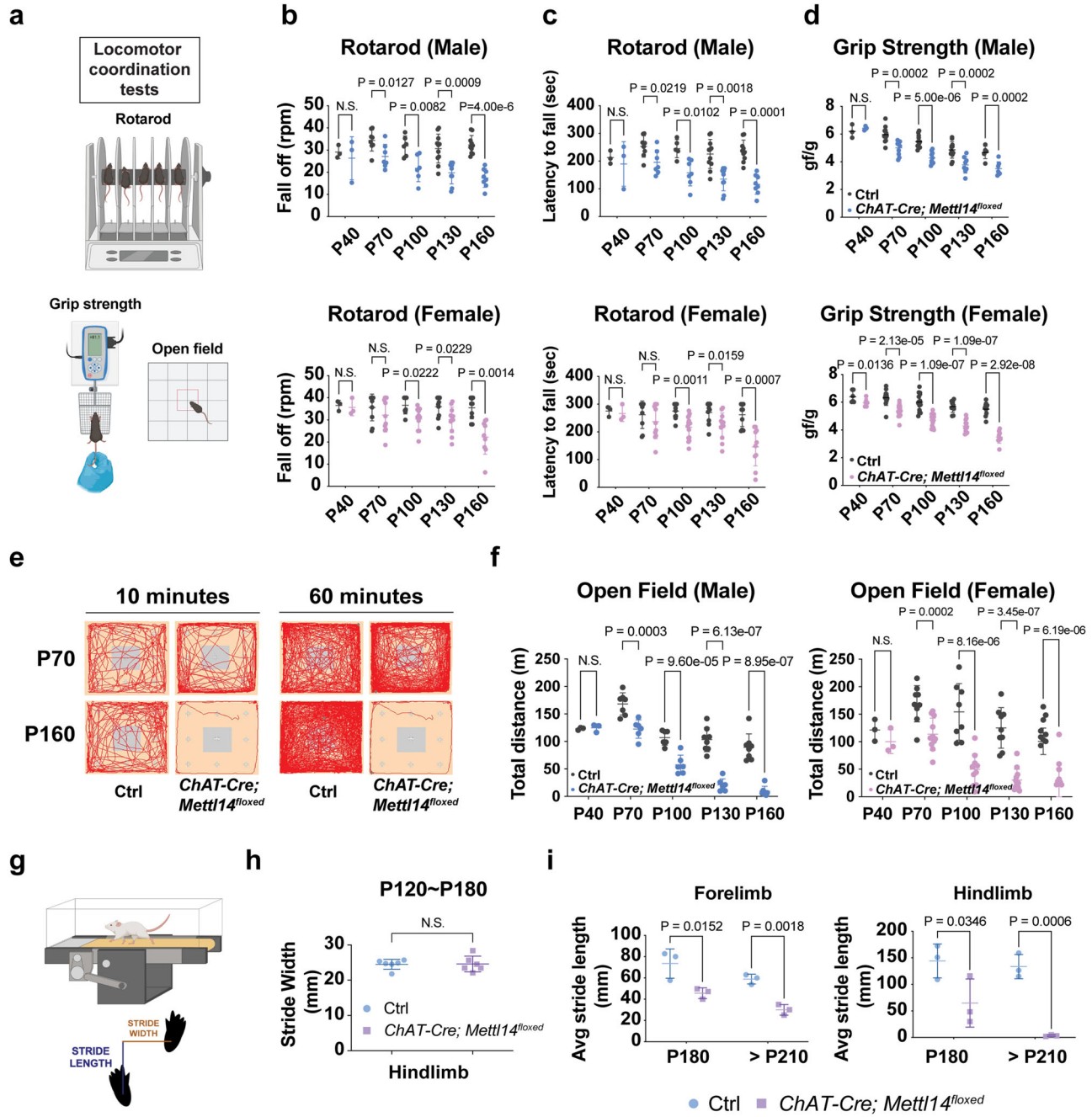

**Fig. 4 | m⁶A-deficient mice display motor deficits that recapitulate ALS.**
**a** Schematic illustration of the behavioral tests conducted to assess motor functions. Created in BioRender. Chen, J. (2025) https://BioRender.com/mrtnwlk. Locomotor coordination on an accelerating rotarod is displayed as the rotation speed at which mice fell off (**b**) and the latency to fall (**c**). (Ctrl: *n* = 3/3, 8/9, 5/10, 11/10, and 10/10 mice; *ChAT-Cre; Mettl14^floxed*: *n* = 3/3, 7/11, 7/15, 9/12, and 8/10 mice at P40, P70, P100, P130, and P160 from males/females, respectively). There was a significant decrease in locomotor activity at P70 and thereafter for males and at P100 and thereafter for females. **d** Forelimb grip strengths for *ChAT-Cre; Mettl14^floxed* male and female mutant mice. (Ctrl: *n* = 3/6, 12/12, 12/14, 12/10, and 6/10 mice and *ChAT-Cre; Mettl14^floxed*: *n* = 3/6, 10/14, 10/17, 8/12, and 7/9 mice at P40, P70, P100, P130, and P160 from males/females, respectively). **e** Travel pathways (red) of

representative trajectory diagrams filmed for 10 and 60 minutes in the open field test arena (square perimeter) for the early-onset (P70) and disease progression (P160) stages of *ChAT-Cre; Mettl14^floxed* and littermate control mice. **f** Total distance traveled in the open field test. (Ctrl: *n* = 3/3, 7/9, 6/8, 8/9, and 8/9 mice; *ChAT-Cre; Mettl14^floxed*: *n* = 3/3, 6/11, 6/11, 7/12, and 7/9 mice at P40, P70, P100, P130, and P160 from males/females, respectively). **g** Schematic illustration of the behavioral tests from the treadmill conducted to assess motor function. Created in BioRender. Chen, J. (2025) https://BioRender.com/mrtnwlk. **h, i** Stride width (usually mediated by INs) is not compromised, whereas stride length (mediated by MNs) is drastically reduced in the *ChAT-Cre; Mettl14^floxed* mice (Speed = 15 cm/s). *n* = 6 mice (3 from P180 and 3 from > P210 mice). Data are presented as mean ± SD, significant *P* values from two-tailed *t*-tests. N.S. non-significant. Source data are provided as a Source data file.

*Rbfox3^high*, *Vipr2^on*), γ (*Htr1d^high*, *Rbfox3^low*, *Spp1^low*, *CrebS^on*, *Pard3b^on*) and γ* (*Htr1d^high*, *Rbfox3^low*, *Spp1^low*, *Stxbp6^on*, *Plch1^on*) MNs (Supplementary Fig. 8a-d). Among α MNs, we could further distinguish fast-fatigue-resistant (*Chodl^on*, *KcnqS^on*), slow-firing (*Sv2a^on*), and fast-fatigable (*Chodl^on*, *KcnqS^off*) cell types (Supplementary Fig. 8e, f). Therefore,

our snATAC/RNAseq dataset encompasses all major adult MN subtypes identified from other studies[45,46].

To discern the molecular alterations underlying the MN degeneration observed in *Sun1^sfGFP; ChAT-Cre; Mettl14^floxed* mice, we concentrated on differentially expressed genes (DEGs) within distinct

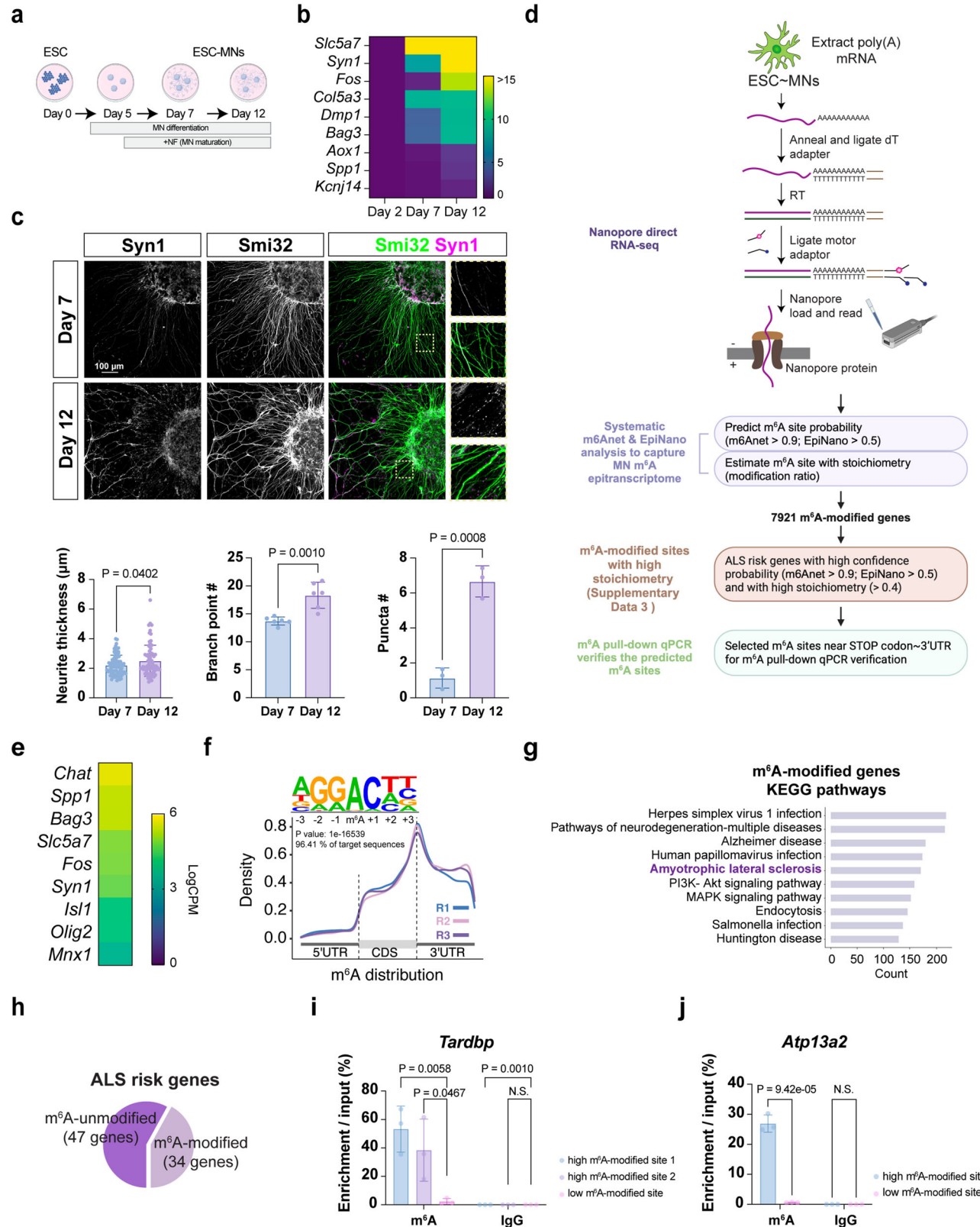

types of cholinergic neurons. By performing a differential expression analysis between *ChAT-Cre; Mettl14^floxed* and Ctrl samples in individual cell types, we identified 652, 500, and 604 DEGs (with p-adjusted <0.05) for skeletal MNs (down-regulated = 291 and up-regulated = 361), visceral MNs (down-regulated = 190 and up-regulated = 310), and cholinergic INs (down-regulated = 165 and up-regulated = 439), respectively (Supplementary Data 4). To identify specific candidates

presenting direct dysregulation due to m⁶A modifications and thus potentially contributing to neuronal degeneration, we conducted GO and KEGG pathway analyses focusing on DEGs exclusively recognized as m⁶A-modified based on Nanopore direct RNA-seq across all cholinergic neurons from *ChAT-Cre; Mettl14^floxed* mice (Fig. 6e). In alignment with our findings of MN denervation and compromised neuromuscular junction (NMJ) size in the *ChAT-Cre; Mettl14^floxed* mice (Fig. 3e-h),

**Fig. 5 | Nanopore direct RNA-seq identifies ALS risk genes as m⁶A modified.**
**a** Timeline of in vitro differentiation and maturation from mESC-MNs. Created in BioRender. Chen, J. (2025) https://BioRender.com/xap28jf. **b** The heatmap from qPCR verification shows that postnatal upregulation of functionally relevant genes. *Slc5a7* serves as a positive control as it is constantly expressed after the postmitotic stage. **c** Upper panel: Immunostaining of Syn1 in Day 7 and Day 12 mESC-MNs. Smi32 labels the MNs and neurites. Scale bar, 100 µm. *n* = 3 independent experiments. All quantification information is provided in the Methods and Source data file. Lower panel: Quantifications reveal a significant increase in neurite thickness (left), neurite complexity (middle), and mature structure revealed by the puncta number (right) in the Day 12 mature mESC-MNs. **d** Overview of the experimental and analysis workflow for conducting Nanopore direct RNA-seq on mature mESC-MNs. Created in BioRender. Chen, J. (2025) https://BioRender.com/cq89p4h. **e** The heatmap from Nanopore direct RNA-seqshows that several feature adult genes (*ChAT, Spp1, Bag, Slc5a7, Fos,* and *Syn1*) are more enriched in the Day12

mature mESC-MNs compared to embryonic genes (*Isl1, Olig2,* and *Mnx1*). **f** Motif preference of m⁶A peaks identifies the DRACH consensus motif (D = A, T, or G, R = A or G, and H = A, T, or C). Metagene profile of enrichment of m⁶A-modified sites across the mRNA transcriptome. 5'UTR, 5' untranslated region; CDS, coding sequence; 3'UTR, 3' untranslated region. Replicates 1, 2, and 3 (R1, R2, and R3) represent the triplicate biological repeats. **g** KEGG pathway analysis of the m⁶A-modified MN epitranscriptome reveals distinct biological pathways related to neurodegenerative diseases. Terms of interest in this study are highlighted in bold purple. **h** Schematic for analyzing the m⁶A-modified MN epitranscriptome, showing that 41.98% of m⁶A-modified genes are ALS risk genes. **i, j** The verification of the predicted m⁶A-modified sites in *Tardbp* and *Atp13a2* by m⁶A pull-down qPCR of selected high m⁶A-modified sites and low m⁶A-modified sites. Points represent individual biological experiments. All data are presented as mean ± SD, *n* = 3, with significant *P* values from two-tailed *t*-tests. N.S. non-significant. Source data are provided as a Source data file.

---

together with neurite degeneration in METTL3i-treated and *METTL3/METTL14* knockdown iPSC-MNs (Fig. 1h-l and Supplementary Fig. 2c-e), GO analysis of m⁶A-modified and down-regulated genes revealed enrichment for pathways related to axonogenesis, synapse organization, cytoskeleton, and tubulin-related gene terms (Fig. 6e and Supplementary Fig. 9a). Notably, we observed the down-regulation of RNA splicing-related genes (e.g., *Malat1, Srsf2, Fus, Srek1, Hnrnph1, Tra2a*) in the *ChAT-Cre; Mettl14^floxed^* samples (Supplementary Fig. 9b), corroborating previous studies showing that m⁶A-mediated splicing induces changes in gene expression[3]. Among this group of analyzed genes, we noted that ALS-associated genes were prominent in our KEGG pathway analysis (Fig. 6f). Specifically, in MNs, these down-regulated genes are linked to neurofilament (*Nefm, Nefl*), tubulin (*Kif5c, Tubb3*), gene encoding DNA/RNA binding protein (*Fus*), and nucleoporin (*Nup93*) (Supplementary Fig. 9c). Mutation or aberrant expression of these genes has been shown to contribute to ALS pathologies, such as cytoskeletal defects and nucleocytoplasmic transport[31]. Moreover, the down-regulation of calcium signaling, which is often linked to neurodegenerative diseases[47], appeared as the top enriched pathway for m⁶A-modified and down-regulated genes in cholinergic neurons (Fig. 6f). Among the m⁶A-modified ALS risk genes (34 out of 81 genes in Supplementary Data 2), expression of two genes—*Fus* and *Bscl2*—was significantly reduced, whereas that of *PIKfyve* was increased in the skeletal MNs of *ChAT-Cre; Mettl14^floxed^* mice (Fig. 6g, h), in accordance with dysregulated ALS risk genes identified from patients[48]. These alterations are likely major contributors to MN degeneration and the subsequent motor behavior deficits observed in our *ChAT-Cre; Mettl14^floxed^* mice. Taken together, these findings strongly imply a direct association between decreased m⁶A levels, neurodegeneration, and ALS pathology.

## Hypo-m⁶A MNs exhibit an increase of closed chromatin regions

In contrast to the down-regulated DEGs, the up-regulated DEGs with m⁶A modifications exhibited conspicuous enrichment in genes responsible for regulating chromatin and histone modification (Fig. 7a and Supplementary Fig. 9d). Notably, several of these genes upregulated in response to hypo-m⁶A—including members of the ATP-dependent *Chd* family, *Bcl7c, Ncoa6,* and *Ube2b*—have been implicated in the DNA damage response and apoptosis. These pathways are commonly implicated in diverse neurodegenerative diseases[49] (Supplementary Fig. 9e). Accordingly, we noticed that the MNs of *ChAT-Cre; Mettl14^floxed^* mice displayed a drastic increase of γH2AX signals, together with a striking increase in repressive histone modification marks (i.e., H3K9me3) (Fig. 7b-e, Supplementary Fig. 10a-c). Our discovery of up-regulatedgenes linked to chromatin/histone modification prompted us to examine if changes in chromatin are associated with the aforementioned identified DEGs. To do so, we probed changes in chromatin accessibility in our snATAC/RNAseq dataset. Among the cholinergic cell types, *ChAT-Cre; Mettl14^floxed^* skeletal MNs displayed the

most drastic peak changes (open peaks = 3278; closed peaks = 10,744), followed by visceral MNs (open peaks = 2495; closed peaks = 920), with cholinergic INs exhibiting the fewest peak changes (open peaks = 171; closed peaks = 602), most of these peaks are in intergenic or intronic regions (Fig. 7f). Subsequently, we integrated and scrutinized the snRNAseq and snATACseq data, revealing that only a modest subset of DEGs align with alterations in chromatin accessibility (Supplementary Fig. 9f and Supplementary Data 5). To further investigate the relationship between changes in chromatin accessibility and dysregulated gene expression, we first identified linked peak-to-gene associations and performed a correlation analysis between gene expression and changes in chromatin accessibility following *Mettl14* ablation (Supplementary Fig. 10d and Supplementary Data 5). Our analysis revealed a low correlation between these two factors. Notably, only 17%, 11%, and 5% of the genes associated with regions of differential chromatin accessibility for skeletal MNs, visceral MNs, and cholinergic neurons, respectively, were both m⁶A-modified and differentially expressed upon *Mettl14* ablation. These discoveries underscore the critical importance of preserving a nuanced equilibrium in the m⁶A transcriptome within adult MNs to maintain neuronal homeostasis. Diminished m⁶A levels may lead to compromised expression of pivotal neuronal and disease-associated genes governed by versatile regulatory mechanisms, i.e., either through direct modification of m⁶A-affected transcripts or by reshaping the chromatin landscape within MNs.

## Restoring m⁶A homeostasis rescues MN degeneration in both familial and sporadic ALS models

The consistent manifestation of hypo-m⁶A in human ALS iPSC-MNs, together with our *ChAT-Cre; Mettl14^floxed^* mice recapitulating ALS pathology, prompted us to explore if bolstering the m⁶A reservoir could represent a therapeutic strategy. Thus, we deployed several familial ALS patient (*C9ORF72^exp-800 G4C2^*, *SOD1^+/L144F^*, *TDP43^G298S^*) iPSC-MN lines and one sALS iPSC-MN line to reflect MN degeneration (Fig. 8a–d). Then, we treated these lines with FB23-2, an inhibitor of FTO (an m⁶A eraser)[50], to see if this approach could be applied to rescue MN degeneration (Fig. 8a). First, we differentiated the ALS iPSCs under defined conditions to cause MN degeneration through a selective ER stressor, CPA (Fig. 8a, b). Subsequently, we applied FB23-2 to determine if doing so could elevate m⁶A levels and thereby restore the m⁶A repertoire to rescue MN degeneration in different contexts of ALS. As expected, we observed a consistently significant increase in m⁶A levels upon applying FB23-2, albeit to varying degrees (Fig. 8c). By using SMI32 to assess MN degeneration and neurite complexity, we observed that FB23-2 promotes MN survival upon CPA stressor treatment for both familial and sporadic ALS MNs (Fig. 8d). Thus, our results indicate that fortifying basal m⁶A levels by adding a m⁶A eraser inhibitor can rescue human ALS iPSC-MNs from degeneration.

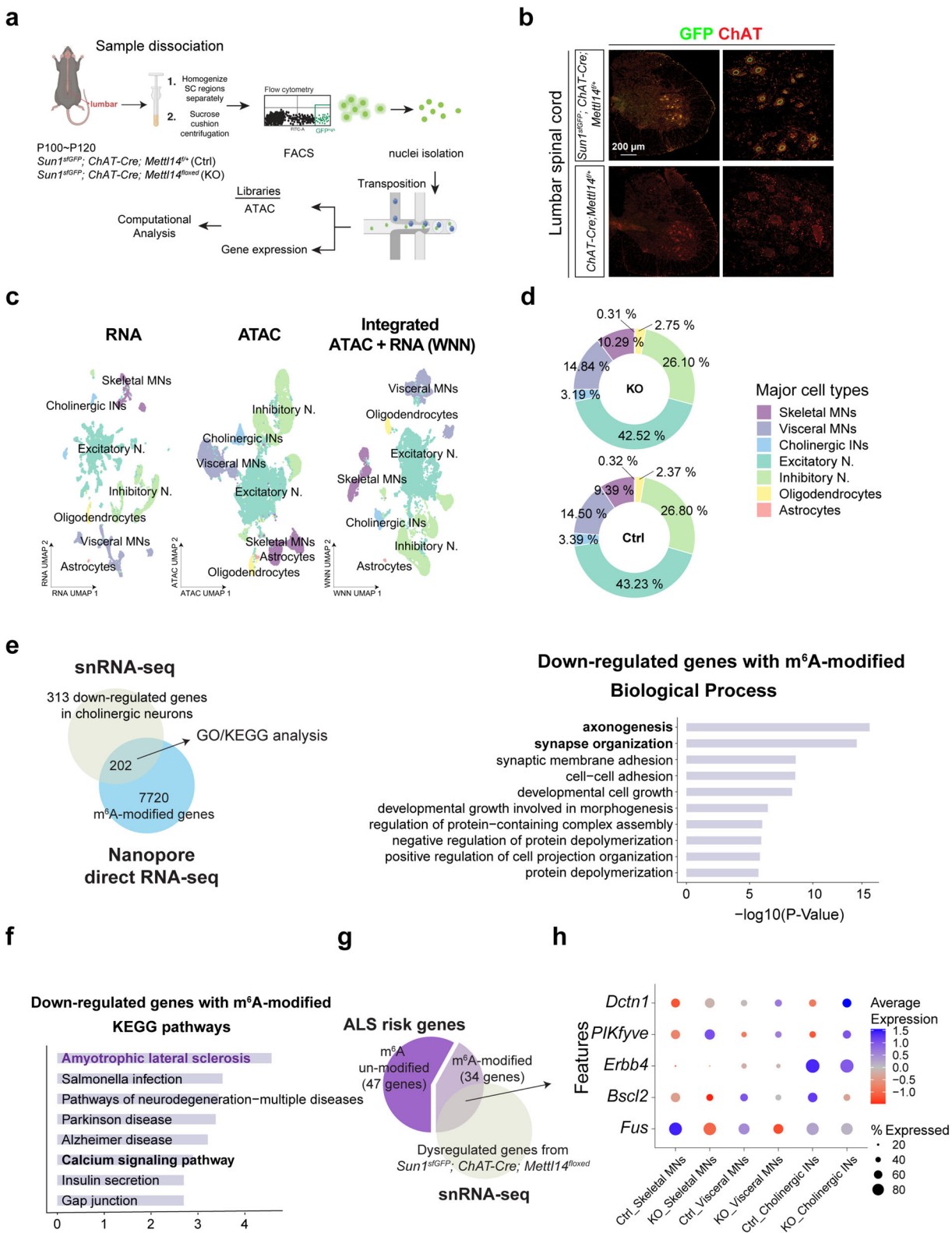

To determine if the neuroprotective effects of the FTO inhibitor on MN degeneration in ALS are mediated through its regulation of m⁶A-modified ALS risk genes (Fig. 6g, h), we performed RNA-seq analysis on ALS iPSC-MNs treated with FB23-2. Differential expression analysis revealed that following FB23-2 treatment, several m⁶A-modified genes involved in synaptic function, RNA metabolism, and chromatin and histone modifications were restored to levels similar to controls (Supplementary Fig. 11). Notably, the expression of multiple m⁶A-modified ALS risk genes was returned to control-like levels (vehicle-treated), suggesting that FTO inhibition may mitigate MN degeneration in ALS by modulating m⁶A-modified gene expression (Fig. 8e). These findings indicate that enhancing m⁶A levels in ALS iPSC-MNs using small molecules could help restore the balance of the m⁶A epitranscriptome and preserve MN integrity.

**Fig. 6 | Identification of m⁶A-modified genes contributing to MN degeneration. a** Overview of the experimental workflow for single nucleus multiomics. Created in BioRender. Chen, J. (2025) https://BioRender.com/esjho7r. **b** Immunostaining showing localization of Sun1-sfGFP-myc in MNs that carry *R26-CAG-LSL-Sun1-sfGFP-myc* together with a Cre driver. Scale bar, 200 μm. The staining experiment was independently repeated. (*n* = 3 mice). **c** Uniform manifold approximation and projection (UMAP) representation of all nuclei that passed quality filtering. Dimensionality reduction and clustering were performed based on gene expression (RNA, left), chromatin accessibility (ATAC, middle), and weighted nearest neighbor (WNN) integration of RNA and ATAC data (right). Clusters are color-coded and annotated using label transfer prediction, referencing Blum et al., 2021. **d** Major cell

type proportions are unaffected at P100-P120 in *Sun1^sfGFP; ChAT-Cre; Mettl14^floxed* mice, a stage before MN degeneration. **e** Schematic for cross-referencing DEGs, particularly those down-regulated in *Sun1^sfGFP; ChAT-Cre; Mettl14^floxed* MNs, and the m⁶A-modified MN epitranscriptome. The resulting data reveals distinct biological pathways (Gene Ontology, right) and KEGG (**f**) that might cause MN degeneration in the *Sun1^sfGFP; ChAT-Cre; Mettl14^floxed* mice. Terms of interest in this study are high-lighted in bold and purple. Schematic for analyzing the dot-plot data (**g**), with the outcome (**h**) showing ALS disease risk genes displaying significant changes in expression in *Sun1^sfGFP; ChAT-Cre; Mettl14^floxed* mice in each cholinergic neuronal subtype.

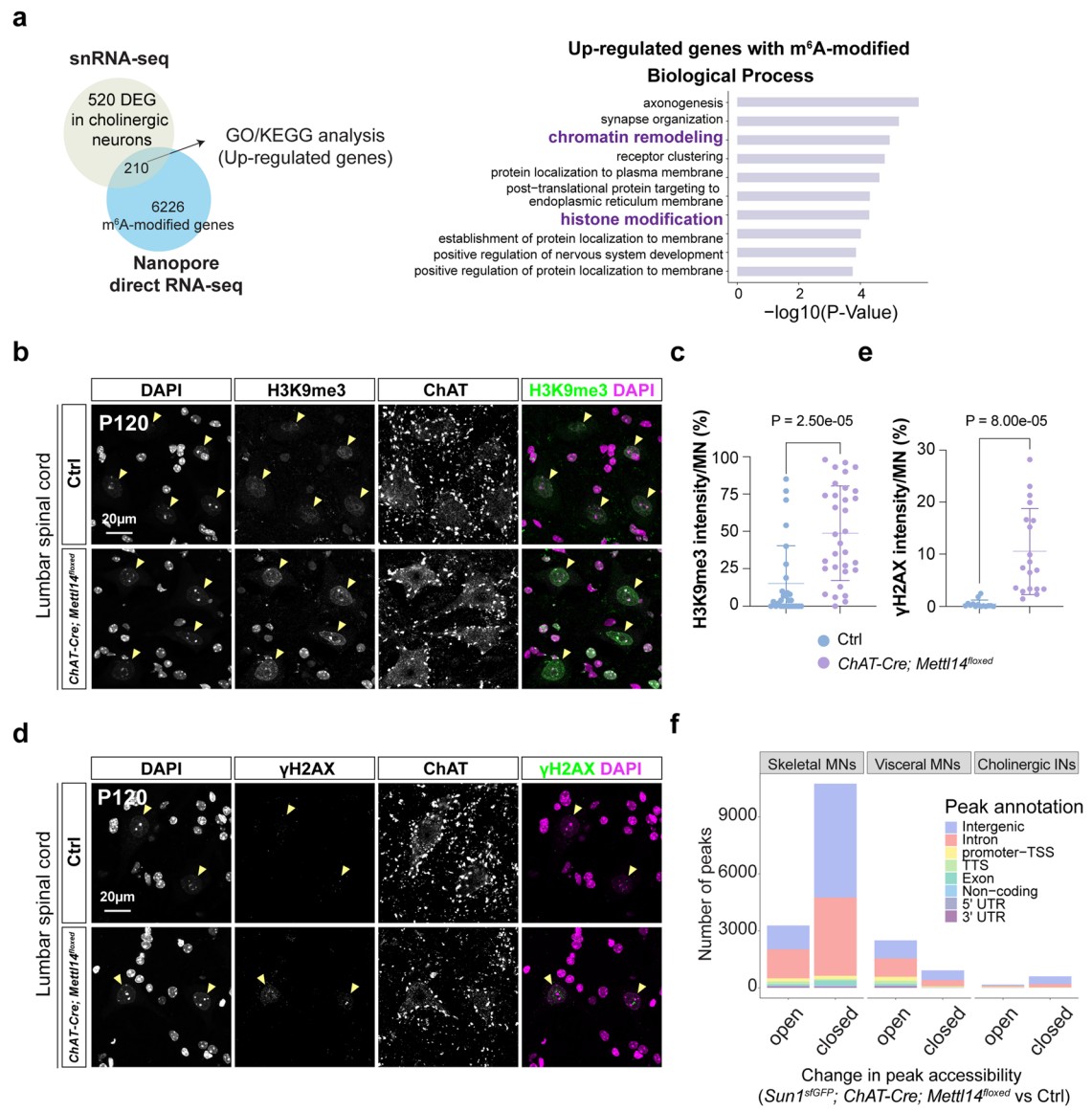

**Fig. 7 | Increase of repressive histone modification marks and closed chromatin regions in *ChAT-Cre; Mettl14^floxed* mice. a** Schematic for cross-referencing of DEGs, particularly those up-regulated in *Sun1^sfGFP; ChAT-Cre; Mettl14^floxed* MNs, and the m⁶A-modified MN epitranscriptome, with the outcome revealing distinct biological pathways (Gene Ontology, right) that might increase repressive histone modification and the DNA damage response (highlighted in bold purple) in the *Sun1^sfGFP; ChAT-Cre; Mettl14^floxed* mice. **b**–**e** Representative images illustrate a dramatic increase in repressive H3K9me3 mark (**b**) and DNA damage γH2AX (**d**) signals. Quantifications of lumbar H3K9me3^on (**c**) and lumbar γH2AX^on (**e**) signal

intensities in the ventral regions from the spinal cord of P120 *ChAT-Cre; Mettl14^floxed* mice compared to littermate controls (Ctrl: *n* = 5 mice, *ChAT-Cre; Mettl14^floxed*: *n* = 6 mice, quantification for all MN nuclei from all views of captured images; Data are presented as mean ± SD with significant *P* values from two-tailed *t*-tests; Scale bars, 20 μm). **f** The bar plot shows changes in the number of peaks and distribution of their annotated genomic locations in cholinergic neuronal subtypes derived from *Sun1^sfGFP; ChAT-Cre; Mettl14^floxed* mice and control (Ctrl) snATAC-seq data. Source data are provided as a Source data file.

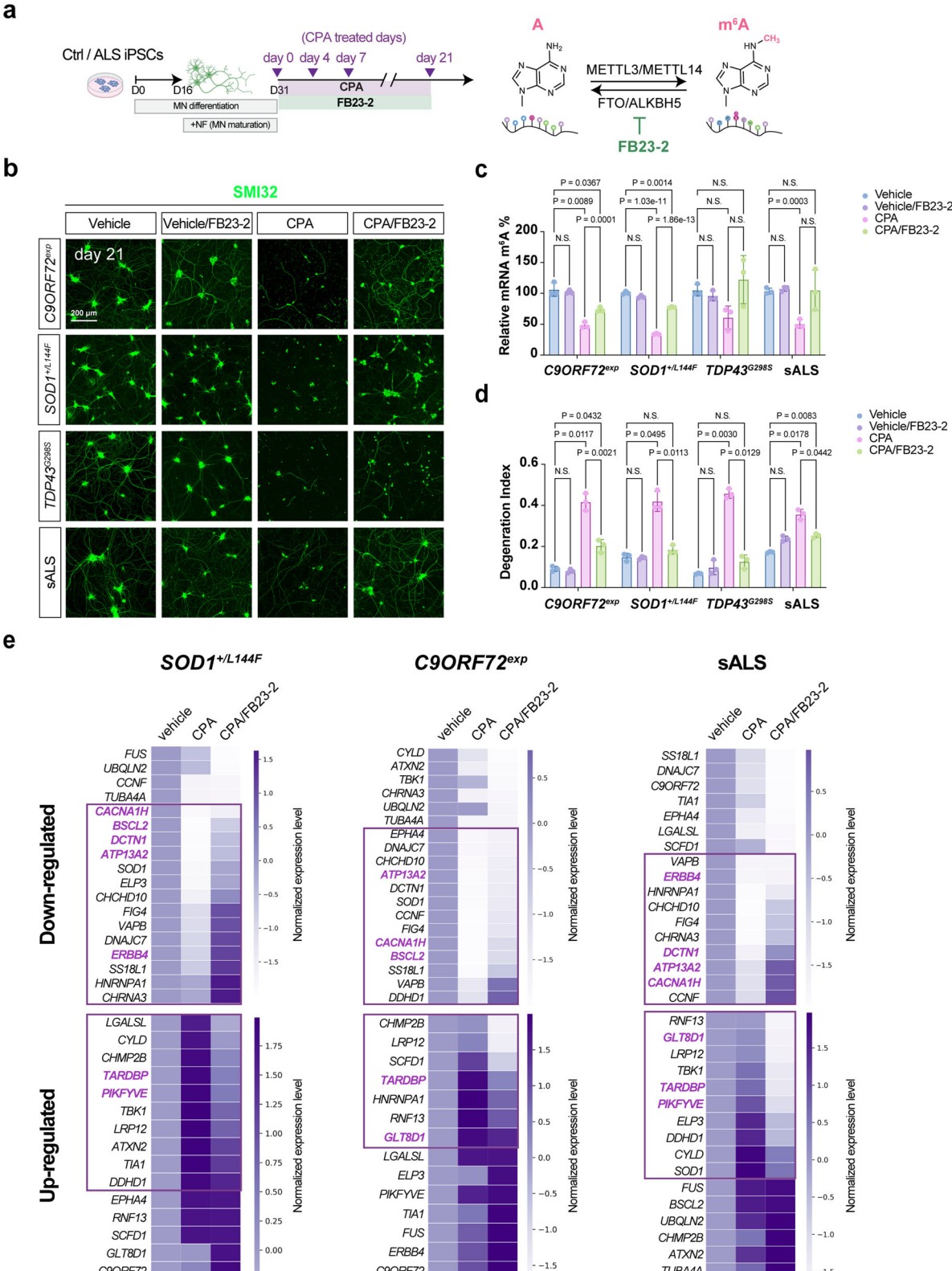

## Fto gene therapy extends the lifespan of SOD1^G93A mice and ameliorates their MN degeneration

Although human iPSCs may be used as an effective platform for drug screening, they preclude further investigation of treatment efficacy at the tissue/organ/behavioral levels. Moreover, the penetrance of FTO inhibitors through the blood-brain barrier is poor and so systematic appliance of such inhibitors might cause adverse effects[51]. To circumvent this issue, we adopted gene therapy as an alternative approach and delivered self-complementary adeno-associated vector serotype 9 (scAAV9) of *Fto*-shRNA to knock down *Fto* expression in the spinal cord (Fig. 9a, b and Supplementary Fig. 12). We first verified the knockdown efficiency of three *Fto*-shRNAs and their restoration of m⁶A level in C2C12 cells (Supplementary Fig. 12a–c). Then we subcloned the *Fto*-shRNAs into scAAV9 driven by the H1 promoter (scAAV9-sh*Fto*)[38].

**Fig. 8 | An m⁶A eraser inhibitor efficiently rescues human ALS iPSC-MNs from premature death by restoring dysregulated genes caused by hypo-m⁶A.**
**a** Schematic illustration of the m⁶A biogenesis pathway and the applied FTO inhibitor (FB23-2) with their corresponding targeting pathways. Created in BioRender. Chen, J. (2025) https://BioRender.com/3nizfu7. **b** Representative images of FB23-2 rescuing the MN degeneration associated with ALS. Scale bar, 200 μm. Quantifications of m⁶A mRNA methylation levels (**c**) and degeneration index values (**d**) at an indicated time point and compared to the CPA treatment. Note the significant rescue of the degeneration index upon applying FB23-2 to CPA-stressed *C9ORF72^exp, SOD1^{+/L144F}, TDP43^{G298S}*, and sALS iPSC-MNs. Data are presented as mean ± SD, *n* = 3, significant *P* values from two-way ANOVA. N.S. non-significant.

**e** Heatmaps of normalized expression level between stress-treated (CPA) ALS-relevant lines with or without subsequent FB23-2 treatment, revealing restorations of many ALS risk genes with m⁶A modifications (highlighted with rectangles) to control (vehicle) levels for the FB23-2-treated groups. A z-score normalization was performed on the normalized read counts across samples for each gene after stress treatment (CPA) with or without subsequent FB23-2 treatment. Samples were normalized to the vehicle control to reveal the normalized expression level. Notably, following FTO inhibitor treatment, these genes were restored to levels comparable to controls among those ALS iPSC-MNs are highlighted in bold purple. Source data are provided as a Source data file.

To test the efficiency of viral infection, we injected scAAV9-*EGFP* into control mice and observed sustained GFP expression in their spinal MNs and in some dorsal cells at 40 days post-injection (Supplementary Fig. 12d). Subsequently, scAAV9-sh*Fto* was injected into *SOD1^{G93A}* mice and their wild-type littermates. Then we verified a dramatic reduction in Fto level in the mouse spinal cords (Fig. 9c). Prominently, scAAV9-sh*Fto* gene therapy delayed the disease onset (Fig. 9d) and robustly prolonged the median survival of *SOD1^{G93A}* mice to ~14 days (Fig. 9e). In the *SOD1^{G93A}* ALS mouse model, we observed a reduction in m⁶A methylation by m⁶A immunostaining (Fig. 9f). Moreover, we also found that m⁶A levels can be upregulated via intrathecal delivery of scAAV9-sh*Fto* in the *SOD1^{G93A}* mouse model (Fig. 9f). The enhanced MN survival (Fig. 9g, h) with a significant reduction of gliosis (Fig. 9g, i) is also shown in *SOD1^{G93A}*; scAAV9-sh*Fto* mouse model.

To evaluate the mice in a more clinically relevant setting, we assayed MN and gastrocnemius (GA) muscle connectivity by measuring the compound motor action potential (CMAP). We performed the CMAP assay from P60 (directly before AAV treatment) to P160 (Fig. 9j). Consistent with a previous study, CMAP amplitude is already reduced in *SOD1^{G93A}* mice at P60 and gradually declines further over time, whereas scAAV9-sh*Fto* treatment mildly ameliorates neuromuscular function at P160. (Fig. 9j). This moderate improvement was also reflected by the enhanced behavioral performance of *SOD1^{G93A}* mice from P120 following scAAV9-sh*Fto* treatment (Fig. 9k, Supplementary Movie 4), with a substantial increase of muscle strength (Fig. 9l), a major clinical measurement for ALS motor score.

To determine if scAAV9-sh*Fto* treatment rescues *SOD1^{G93A}* mice through m⁶A-mediated molecular changes, such as histone modifications in H3K9me3 and γH2AX, we assessed these markers in four groups—wild-type control (Ctrl), Ctrl treated with scAAV9-sh*Fto* (Ctrl; scAAV9-sh*Fto*), *SOD1^{G93A}*, and *SOD1^{G93A}* treated with scAAV9-sh*Fto* (*SOD1^{G93A}*; scAAV9-sh*Fto*)—at postnatal day 140 (P140), i.e., when *SOD1^{G93A}* mice are at an early symptomatic stage (Fig. 10). Remarkably, H3K9me3 levels were significantly elevated in the ventral horn MNs of the *SOD1^{G93A}* mice and were restored to control levels following scAAV9-sh*Fto* treatment (Fig. 10), suggesting that epigenetic dysregulation may be a novel hallmark of ALS pathogenesis mediated by hypo-m⁶A. In contrast, no significant changes in γH2AX were observed in the spinal cords of the *SOD1^{G93A}* mice (Supplementary Fig. 13).

Overall, our results support that m⁶A hypomethylation promotes ALS and, significantly, augmenting the m⁶A reservoir can mitigate the disease phenotypes of sporadic and familial human ALS iPSC-MNs. In a *SOD1*-linked ALS context, enhancing m⁶A levels in adult MNs either via small molecule treatment or scAAV9 delivery delays the onset of MN degeneration and enhances motor function. The discoveries we have presented herein reveal m⁶A as a potential prognostic indicator for MN degeneration and a promising therapeutic candidate in individuals afflicted with ALS. We discuss the significance and implications of our study in detail below.

## Discussion

In this study, we generated two conditional spinal MN-*Mettl14* knockout mouse lines, using the *Olig2-Cre; Mettl14^{floxed}* mice to impair m⁶A

homeostasis from the embryonic stage and the *ChAT-Cre; Mettl14^{floxed}* mice to disrupt m⁶A levels from the late embryonic/early postnatal stage. Although *Olig2-Cre; Mettl14^{floxed}* mice exhibit early postnatal lethality (~P24 to P28), their MNs are relatively normal. The *ChAT-Cre; Mettl14^{floxed}* mice also display normal MN development and ordinary gross appearance at the postnatal and juvenile stages. These results indicate that m⁶A homeostasis appears to be less critical for embryonic spinal MN development. This scenario is different to conditional *Mettl14* knockout in mouse neural progenitor cells (NPCs) using *Nestin-Cre* that impairs NPC differentiation, prolongs cell cycle progression of radial glia, and extends cortical neurogenesis into postnatal stages[18,19]. Thus, our results, together with these previously published results, emphasize the neuron-type context-dependent role of m⁶A homeostasis, prompting the notion that the m⁶A epitranscriptome is both dynamic and diverse, and it operates in a cell and developmental context-dependent manner. Our *Olig2-Cre; Mettl14^{floxed}* mice display a shivering phenotype, likely a reflection of compromised oligodendrocytes, consistent with a previous study using a different *Olig2-Cre* line[40]. We have shown previously that the *Olig2-Cre* line we deployed in this study displays more efficient floxed allele removal activity than the other line[52] used in that previous study, so it is understandable that we observed a more lethal shivering postnatal phenotype than the other study that only showed oligodendrocyte defects at the adult stage[40].

Despite sporadic ALS cases predominating in the real world, current animal models often rely on familial genetic mutations. The expanding genetic spectrum of ALS has prompted continuous efforts to establish rodent models that emulate human physiological deficits and pathological manifestations[30,48,53]. Nevertheless, translation of effective treatment outcomes in rodent models to clinical success in humans remains limited, emphasizing the crucial need for an authentic rodent ALS model that mirrors human symptoms. Thus, even though rodent models of familial ALS are prevalent, creating sporadic ALS animal models is pivotal for pathomechanistic insights and establishing novel therapeutics. Various sporadic ALS pathologies in MNs have been simulated in mice, such as reduced ADAR2 protein levels or filament alterations[54,55]. Environmental toxin exposure, including metals and β-N-methylamino-L-alanine, has also been explored alongside patient-derived fluid injections[56]. These models partially replicate ALS features, but their status as true sporadic ALS models is arguable. Here, our *ChAT-Cre; Mettl14^{floxed}* mice exhibit several salient features of human ALS pathologies and symptoms (Supplementary Fig. 14a). At the phenotypic level, our *ChAT-Cre; Mettl14^{floxed}* mice exhibit late-stage paralysis with early adult lethality, diminished body weight, kyphosis with muscle atrophy, as well as coordination and motor deficits. At the pathological level, our *ChAT-Cre; Mettl14^{floxed}* mice display gradual MN loss, neuroinflammation, axonopathy, neuromuscular junction abnormality, and muscle denervation. Of paramount significance, our novel mouse model distinctly manifests early-onset cytoplasmic aggregations of Tardbp (Tdp43) and Fus, representing a principal hallmark observed in the post-mortem tissues of both familial and sporadic ALS cases, yet this phenotype has been conspicuously rare in existing rodent models. Moreover, we observed consistent m⁶A hypomethylation in several familial and sporadic ALS

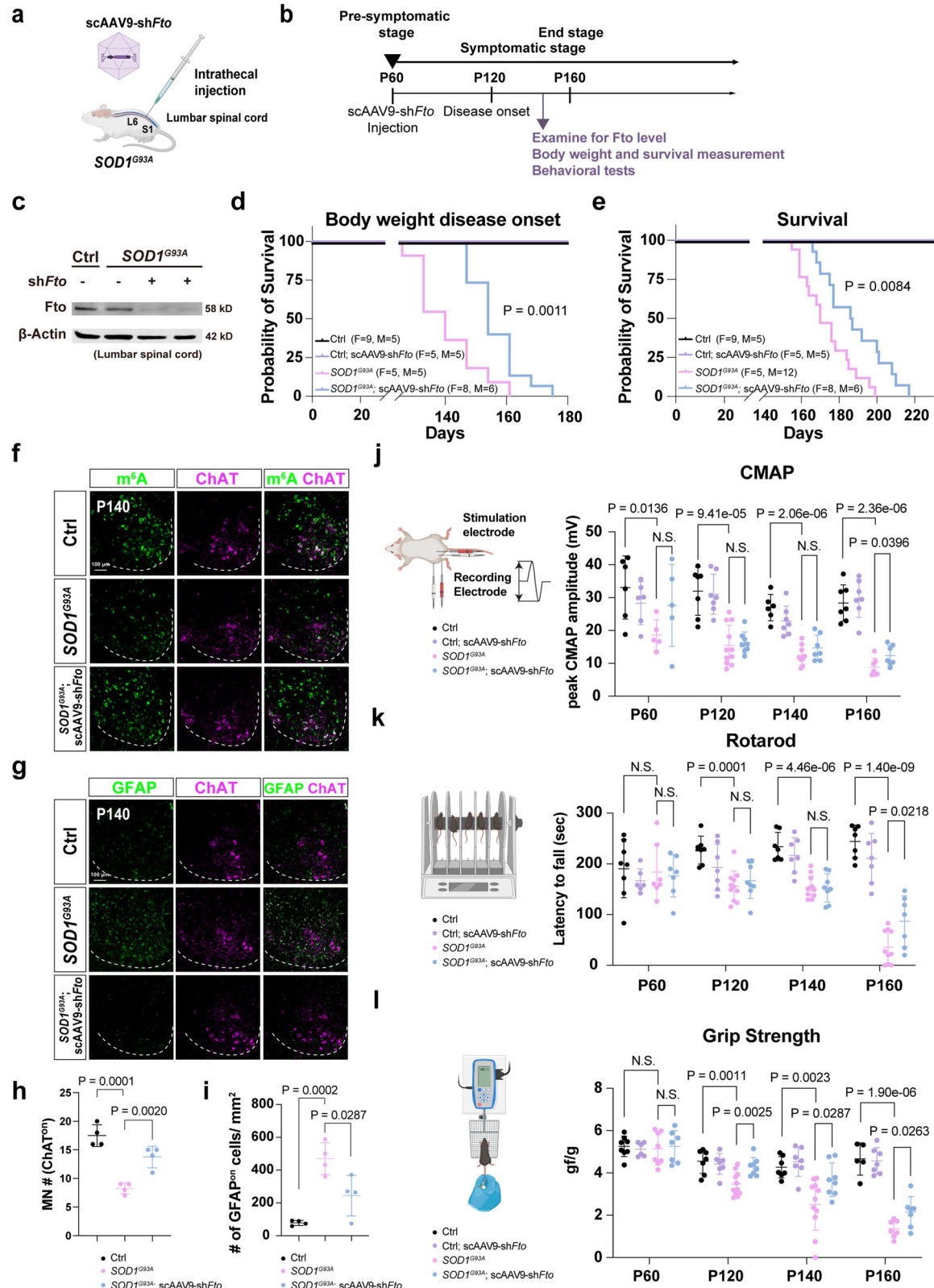

lines we tested, congruent with a recent study indicating that the expression of m⁶A methyltransferases and levels of m⁶A RNA modification are down-regulated in C9ORF72-ALS/FTD patients[36]. Thus, taken together, we assert that m⁶A hypomethylation leads to ALS and that impairing m⁶A modification in adult MNs has provided us with one of the most reliable rodent models of ALS. Although *ChAT-Cre; Mettl14^{floxed}* mice exhibited several ALS-like phenotypes, we were

unable to determine whether these effects were specifically attributable to C-boutons synapsing on the soma and proximal dendrites of MNs, the MNs themselves, or both. This is an intriguing question, as recent reports have suggested that spinal inhibitory neurons, including C-boutons, might degenerate before MNs in a mouse model of ALS[57,58]. Currently, the lack of a Cre driver that specifically targets adult MNs limits examining this question. However, our single-nuclei ATAC-

**Fig. 9 | Gene therapy of adult *SOD1^G93A* mice by overexpressing *Fto*-shRNA is sufficient to protect neuromuscular function and delay disease onset.**
**a, b** Overview of the experimental strategy. Created in BioRender. Chen, J. (2025) https://BioRender.com/3m2vxum. **c** Western blot reveals that Fto protein level is reduced in mouse lumbar spinal cords after scAAV9-sh*Fto* injection (*n* = 3 mice). **d** and **e** Kaplan-Meier survival curves with log-rank test revealing prolongation of the onset of weight decline in *SOD1^G93A* mice (from ~140 to ~155 days), with lifespans extended by ~10% (from ~170 days to ~187 days), following scAAV9-sh*Fto* injection. m⁶A levels is upregulated (**f**), MN number is rescued (**f**, with quantification in **h**), and gliosis is reduced (**g**, with quantification in **i**) after scAAV9-sh*Fto* injection of *SOD1^G93A* mice at P140. Scale bars, 100 μm. (mean ± SD, *n* = 4 mice; two-tailed *t*-tests). **j** The CMAP amplitude is reduced in *SOD1^G93A* mice at P60 and gradually declines further over time, whereas scAAV9-sh*Fto* treatment significantly ameliorates neuromuscular function at P160 (mean ± SD, Ctrl: *n* = 5/1, 4/3, 3/3, and 4/

3 mice, Ctrl; scAAV9-sh*Fto*: *n* = 5/1, 5/2, 5/2, and 5/2 mice; *SOD1^G93A*: *n* = 2/3, 5/6, 4/5, and 4/3 mice; *SOD1^G93A*; scAAV9-sh*Fto*: *n* = 3/2, 5/3, 4/3, and 4/3 mice at P60, 120, 140, and 160 from males/females respectively; two-tailed *t*-tests) (right). **k, l** Motor coordination and muscle strength are enhanced by scAAV9-sh*Fto* injection, as assayed by rotarod test at P60-P160 (**k**) (mean ± SD, Ctrl: *n* = 6/2, 4/3, 4/3, and 4/3 mice, Ctrl; scAAV9-sh*Fto*: *n* = 5/1, 5/2, 5/2, and 5/2 mice; *SOD1^G93A*: *n* = 3/4, 5/6, 5/6, and 5/5 mice; *SOD1^G93A*; scAAV9-sh*Fto*: *n* = 5/2, 5/3, 5/3, and 5/3 mice at P60, 120, 140, and 160 from males/females respectively; two-tailed *t*-tests), and by grip strength test (**l**) (mean ± SD, Ctrl: *n* = 6/2, 4/3, 4/3, and 4/1 mice, Ctrl; scAAV9-sh*Fto*: *n* = 5/1, 5/2, 5/2, and 5/2 mice; *SOD1^G93A*: *n* = 4/4, 5/6, 5/6, and 4/3 mice; *SOD1^G93A*; scAAV9-sh*Fto*: *n* = 5/2, 5/3, 5/3, and 5/3 mice at P60, 120, 140, and 160 from males/females respectively; two-tailed *t*-tests). N.S. non-significant. Illustrations in **j**–**l** were created in BioRender. Chen, J. (2025) https://BioRender.com/mrtnwlk. Source data are provided as a Source data file.

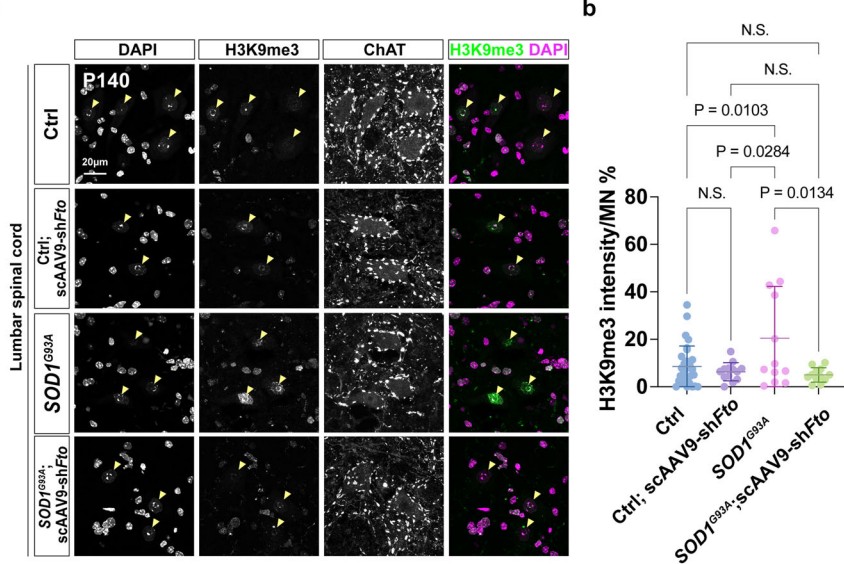

**Fig. 10 | An increase of H3K9me3 is observed in MNs of *SOD1^G93A* and is reduced after scAAV9-sh*Fto* treatment. a** Representative image illustrating H3K9me3 marks (yellow arrowheads) in the ventral horn of different sets of Ctrl and *SOD1^G93A* mice, with scAAV9-sh*Fto* intrathecal injections. **b** Quantifications of lumbar H3K9me3^on MNs (Ctrl: *n* = 7 mice, Ctrl; scAAV9-sh*Fto*: *n* = 3 mice, *SOD1^G93A*:

*n* = 4 mice, *SOD1^G93A*; scAAV9-sh*Fto*: *n* = 4 mice, quantification for all MN nuclei from all views of captured images; Scale bars, 20 μm). Data are presented as mean ± SD with significant *P* values from two-way ANOVA. N.S. non-significant. Source data are provided as a Source data file.

---

seq data offer a promising avenue for identifying enhancers specific to adult cholinergic INs and MNs. Generating Cre lines driven by these enhancers could allow for a more precise dissection of the respective contributions of INs and MNs to the observed phenotypes.

As a hallmark of ALS, why do Tardbp (Tdp43) and Fus move out of the nucleus to form cytoplasmic aggregations in the *ChAT-Cre; Mettl14^floxed* mice? A previous study has revealed that a lack of specific RNA modifications may affect global and/or local translation rates, consequently increasing protein aggregation[59]. Thus, it has been proposed that RNA modifications serve as conduits of information linking a cell's metabolic condition with its translational productivity[60]. Consequently, any disruption in regulating RNA modifications could potentially perturb the equilibrium between metabolic processes and protein synthesis. Further work is needed to disentangle the causal relationship between dysregulation of RNA modifications and Tdp43 translocation. Additionally, the nuclear m⁶A reader YTHDC1 has been shown to exert an important role in modulating many biological processes and contributing to disease, especially cancers. YTHDC1 might be the main mediator of a series of m⁶A readers, thereby controlling their activity in neuronal functions[61]. Moreover, *YTHDC1* RNAs have been discovered as binding to TDP43 protein in human SH-SY5Y neuroblastoma cells[62], and peripheral blood sample transcriptional profiling of a huge heterogeneous ALS cohort (not only sporadic cases)

revealed YTHDC1 to be differentially expressed[63]. These results point to YTHDC1-mediated m⁶A nuclear events and nucleocytoplasmic trafficking as contributing to ALS. Further experiments are warranted to dissect this potential disease mechanism in detail.

Based on four lines of evidence, we believe m⁶A hypomethylation is one of the key factors leading to ALS. First, the reduced presence of the m⁶A writer complex (METTL3/METTL14) is manifested in a random yet comprehensive (n > 800) selection of familial and sporadic ALS iPSC-MNs and postmortem tissues, consistent with a recently published study[36]. Second, a global reduction in the m⁶A epitranscriptome of several familial ALS iPSC-MNs precedes MN degeneration. Third, *ChAT-Cre; Mettl14^floxed* mice phenocopy ALS symptoms at molecular, cellular, and phenotypic levels. Fourth, restoring the m⁶A repertoire significantly mitigates ALS pathology in human iPSC and mouse models. However, the discrepancy between the current study's findings (i.e., the impact of hypo-m⁶A) and another recent study reporting m⁶A hypermethylation in the spinal cord of sporadic ALS patients warrants critical examination[35]. Several plausible explanations may account for this disparity. For instance, experimental model limitations, with previous studies predominantly utilizing cellular models and postmortem tissues, thereby potentially limiting direct correlations to in vivo pathology. The timing of sample collection, particularly at the end stage of the disease, might not adequately capture the initial

contributors to ALS pathology. We suggest that studies involving animal models are necessary to establish a more direct link between manipulating m⁶A levels and ALS progression. Alternatively, the discrepancy could be attributable to the multifaceted and complex nature of sALS, encompassing diverse pathological mechanisms. To detect m⁶A levels, different assays have been used by our study and those of others, including antibody-based pull down, dot blot, ELISA, and microarray method[51]. We have further applied Oxford Nanopore direct RNA-seq to uncover m⁶A stoichiometry, which appears to provide better sensitivity. Each of these methods has advantages and disadvantages in terms of the sensitivity and feasibility of large-scale studies[51]. We advocate screening m⁶A levels in a larger cohort of sALS cases using advanced techniques such as direct Mass Spectrometry in future. Our study acknowledges the pivotal role of an optimal m⁶A reservoir in maintaining adult MN function, highlighting that both m⁶A hypomethylation and hypermethylation could be detrimental[3]. This nuanced perspective underscores the need for a balanced understanding of m⁶A modifications in ALS pathophysiology and it calls for further investigation using diverse methodologies and models to comprehensively understand the role of m⁶A dysregulation in ALS progression.

In our proposed model (depicted in Supplementary Fig. 14b), we postulate that the aberrant down-regulation of the two m⁶A methyltransferases, namely METTL3 and METTL14, in ALS reflects an anomaly likely associated with the natural aging process compounded by unidentified exacerbating factors[29]. This reduced expression of m⁶A methyltransferases has multifaceted implications. Firstly, it instigates widespread dysregulation across the mRNA transcriptome, notably enriching pathways associated with ALS risk genes, synaptic activity, and neuronal functional pathways. Simultaneously, m⁶A modification of chromatin regulators, crucial for modulating heterochromatin repression, is diminished. This global dysregulation of m⁶A-mediated epitranscriptomic processes disrupts RNA stability and perturbs the repressive chromatin landscape, potentially exacerbating the degeneration of MNs in the context of ALS. Strikingly, we observed a significant increase in closed chromatin regions in the *ChAT-Cre; Mettl14^floxed* mice, with only a small fraction of these chromatin changes correlating with dysregulated gene expression. Given that previous studies have shown FTO-mediated LINE1 RNA m⁶A demethylation plays a role in regulating chromatin state and gene expression during mouse oocyte and embryonic development[64], and that retrotransposon reactivation has been observed in some ALS postmortem tissues[62], it would be intriguing to explore in future studies whether *Mettl14*-mediated m⁶A impairment leads to retrotransposon reactivation through its interaction with chromatin state.

In this study, we adopted two approaches to test possible treatments of ALS by bolstering the m⁶A reservoir. First, by applying a small molecule FTO inhibitor (FB23-2) in our familial and sporadic ALS iPSC-MNs, we prevented the MNs from degenerating. FTO inhibition notably restored to control levels the expression patterns of several genes associated with ALS risk and chromatin-regulated pathways. This outcome strongly suggests that the FTO inhibitor's efficacy in preserving MNs from degeneration primarily stems from its ability to restore m⁶A-modified genes relevant to ALS pathology. Second, we observed a significant enhancement of motor and neuromuscular functions, together with a significant delay in disease onset following scAAV9-sh*Fto* intrathecal injection in our *SOD1^G93A* mice. Building on this exciting outcome, we are currently testing some new synthesized FTO inhibitors with stronger activity and better blood-brain barrier penetrance. We knocked down FTO using a ubiquitous promoter as an initial attempt. Future optimization to knock down FTO in a cell type-specific manner might minimize possible adverse effects, although targeting FTO might also represent a relatively safe treatment strategy as *Fto* knockout mice are morphologically normal and only display mild learning defects[65,66]. Future experiments to develop a new drug

cocktail by combining small molecules together with gene therapy to fortify the m⁶A reservoir of both familial and sporadic ALS patients offer tantalizing treatment prospects. While we have validated several ALS risk genes with m⁶A-modified sites that are dysregulated in the *ChAT-Cre; Mettl14^floxed* mice, the direct contribution of these genes as a cohort to MN degeneration in ALS remains to be systematically tested. Furthermore, FTO is not only an m⁶A demethylase, but also functions as a demethylase for the m₂-isoform ($N^6$,2'-$O$-dimethyladenosine, m⁶Am) in small nuclear RNAs (snRNAs)[14] and mRNAs. Although we have demonstrated that FTO inhibition rescues the expression of many ALS pathway- and chromatin regulator-related genes, it remains to be investigated if this therapeutic effect involves m⁶Am-modified snRNAs.

Additionally, though we identified an increase in H3K9me3 and closed chromatin regions as potential new features of ALS, the precise mechanism of how hypo-m⁶A induces these chromatin changes remains unclear. It is also uncertain if the increase in H3K9me3 directly correlates with closed chromatin regions. Future large-scale investigations using human ALS iPSC-MNs will be crucial to further explore these potential pathological features. Moreover, it would be valuable to assess if combinatorial treatments targeting both H3K9me3 and m⁶A modifications could offer enhanced benefits in preventing MN degeneration. Extending beyond the challenge of ALS, perturbations of the m⁶A reservoir is a recurring theme of other neurodegenerative conditions, including Alzheimer's disease, Parkinson's disease, and multiple sclerosis[3]. As these pioneering paradigms relating to m⁶A modification ripple through the scientific landscape, we envisage their transformative potential will extend to diverse ailments in the future.

## Methods

### Ethical compliance
All mice experimental procedures were performed in accordance with guidelines approved by the Institutional Animal Care and Use Committee (IACUC) at Academia Sinica (protocol number 13-06-559 and 23-07-2022).

### Human iPSC culture
The *SOD1^{+/L144F}* ALS iPSC mutant line (female) and a healthy control line were acquired from the Harvard Stem Cell Institute iPSC Core Facility. *TDP43^{G298S}* (male), *C9ORF72^{exp}* (male), an isogenic control line of *C9ORF72^{exp}*, and sporadic ALS iPSC lines (male) were acquired from the Answer ALS project of Cedars-Sinai. Cells were maintained in feeder-free Essential 8 (Life Science) conditions and subcultured by 0.5 mM EDTA treatment. Cells were cultured in a 5% $CO_2$ humidified atmosphere at 37 °C. The influence of sex was not assessed in this study. Rather the effect of CPA treatment was compared between isogenic lines; the FB23-2 treatment to rescue ALS iPSC lines was compared with the vehicle control.

### Differentiation and survival assay of human iPSC-MNs
Human iPSCs were differentiated into MNs using an improved protocol[67]. Specifically, iPSCs were dissociated into single cells using accutase (Gibco) at day 0. Next, $1.5$-$2 \times 10^5$ cells were resuspended in 10 mL N2/B27 medium [1:1 of DMEM-F12 and Neurobasal medium containing N2 (Life Technologies), B27 (Life Technologies), 1% penicillin-streptomycin, 200 mM Glutamax, 0.2 mM 2-mercaptoethanol, and 0.5 mM ascorbic acid (Sigma-Aldrich)], supplemented with 10 mM Y-27632 (STemGent), 20 mM SB431542 (Merck), 0.1 mM LDN 193189 (Sigma-Aldrich), 3 mM CHIR-99021 (Merck), and 10 ng/mL bFGF (Peprotech). Small embryoid bodies (EBs) should become visible after two days of differentiation. SB431542 and LDN 193189 were kept for four days and supplemented with 100 nM retinoic acid and 0.5 mM smoothened agonist from day 2 to day 16. Then, 10 ng/mL BDNF (Peprotech) was added from day 7, and 10 mM DAPT (Calbiochem) was included from day 9 to day 16. After HB9^on nascent MNs had been generated (day 11), EBs were dissociated utilizing accutase, and the

dissociated MNs were plated onto poly-L-Ornithine/laminin-coated four-well plates at a density of $5 \times 10^4$ cells per well and $7.5 \times 10^5$ cells per 6 cm plate. Dissociated MNs were maintained in MN culture medium [CultureOne Supplement medium (Thermo Fisher Scientific Inc.) containing 10 mM Y-27632, 10 ng/mL BDNF, 10 ng/mL GDNF (Peprotech) and 10 mM 5-fluoro-20-deoxyuridine/Uridine (to inhibit proliferating cells) (Merck)]. The medium was replenished every three to five days.

### FTO inhibitor treatments

To accelerate MN degeneration, the CultureOne Supplement medium of day 31 MN culture was replaced with N2/B27 medium only (no BDNF and GDNF), for which CPA was supplemented to accelerate degeneration for another seven days. FB23-2 is a small molecule that inhibits the demethylase activity to elevate $m^6A$ levels. FB23-2 (1 nM) was treated together with the CPA (50 μM) for 15-30 days (change the small molecules and medium every two days) in ALS iPSC-MNs with the increased $m^6A$ mRNA methylation level and extended disease onset or rescue of the degenerative process. Then MNs were revealed by SMI32 immunostaining and captured by an ImageXpress® Micro XLS High-Content Imaging System (Molecular Devices). The degeneration index was calculated at the indicated time point after CPA treatment.

### Quantification of neurite degeneration

To ensure accurate measurement of neurites, images were captured from blindly selected regions with well-separated axon tracts. Neurite fragmentation was then quantified using an automated image analysis method. The extent of neurite degeneration was expressed as a degeneration index (DI), defined as the ratio of the fragmented neurite area to the total neurite area. To process images for DI calculation, gray intensity in images was first normalized using the auto-level function of GNU Image Manipulation Program (GIMP) software, ensuring consistent background intensity across all images. Subsequently, ImageJ and Ilastik software were employed to binarize the images and remove cell bodies, resulting in a black-and-white rendering of neurites. While intact neurites exhibit continuity, degenerating neurites display disrupted, particulate structures due to blebbing and fragmentation. To quantify the fragmented areas of these degenerating neurites, the Particle Analyzer algorithm in ImageJ was used, with detection parameters set for size (20–10,000 pixels) and circularity (0.2–1.0). The total area of identified neurite fragments was then divided by the overall black neurite area to calculate the DI. Consistent with previous studies, the DI ranged from 0, representing completely intact neurites, to 1.0, indicating complete degeneration into fragmented particles[68,69].

### $m^6A$ methylation and quantification

The total RNA or mRNA was extracted from samples using Trizol (Life Technologies) and $m^6A$ methylation was quantified using the $m^6A$ RNA Methylation Assay Kit (ab185912, Abcam), with absorbance read at 450 nm in biological triplicates using an EnSpire Multimode reader and EnSpire software. To quantify relative $m^6A$ RNA methylation, the percentage $m^6A$ content in total RNA was calculated as:

$$m^6A\% = \frac{(Sample\ OD - negative\ control\ OD)/\text{the amount of input sample RNA}}{\times 100\% (positive\ control\ OD - negative\ control\ OD)/\text{the amount of input positive control}}$$

### $m^6A$ dot blot assay

Total RNA was extracted with Trizol (Life Technologies) and purified into mRNA using Invitrogen PolyA$^+$ RNA selection (Dynabeads mRNA Purification Kit). After mRNA purification, 100-200 ng mRNA was dropped onto the Hybond-N+ membrane (Amersham Hybond™ -N+ Membranes) for UV crosslinking (1200 μJ X100; UV Stratalinker 2400). Hybridized mRNA was blocked in 5% milk in 0.5% Triton X-100/PBS for one hour and then incubated with anti-$m^6A$ antibody (1:1,000,

Synaptic) in a blocking solution at 4 °C overnight. After three TBST washes, the mRNA was incubated with an anti-HRP secondary antibody (1:10,000; Santa Cruz) in a blocking solution for 30 minutes at room temperature. The signal was developed with enhanced chemiluminescence (ImageQuant LAS 4000). The hybridized mRNA was stained with 0.2% methylene blue in 0.3 M sodium acetate (pH 5.2) as the loading control of mRNA.

### Mouse crosses

Mice carrying the mutant human $SOD1^{G93A}$ transgene (B6SJL-Tg(SOD1*G93A)1Gur/J) were purchased from the Jackson Laboratory (JAX:002726). $Mettl14^{floxed}$ mice were obtained from Chuan He in the University of Chicago[18], and $Olig2$-Cre was a gift from Tom Jessell in Columbia University. $ChAT$-$IRES$-$Cre$ ($\Delta neo$) (Jackson Laboratory stock: 031661), in which the neomycin cassette was removed to avoid the ectopic expression sometimes observed in the $ChAT$-$IRES$-$Cre$ line, were bred with the $Mettl14^{floxed}$ allele to generate $ChAT$-$Cre$; $Mettl14^{f/+}$ and $ChAT$-$Cre$; $Mettl14^{floxed}$ mice. WT and conditional knockout mice were generated by crossing $ChAT$-$Cre$; $Mettl14^{f/+}$ males and $Mettl14^{floxed}$ females. CAG-Sun1/sfGFP mice were purchased from JAX (stock no. 021039; B6;129-$Gt(ROSA)26Sor^{tm5(CAG-Sun1/sfGFP)Nat}$/J) and crossed with $ChAT$-$Cre$; $Mettl14^{floxed}$ mice. F1 heterozygous reporter mice were maintained to P100-P120 and then sacrificed for subsequent single-nucleus multiome experiments. All live animals were kept in an SPF animal facility, housed at ~ 55% humidity, 25 °C, on a 12:12-hour light/dark cycle, and approved and overseen by IACUC, Academia Sinica.

### Immunostaining

All adult Spinal cord sections were permeabilized in 0.5% Triton X-100/PBS for one hour and then blocked in 3% bovine serum albumin (BSA) in 0.5% Triton X-100/PBS for one hour. The sections were incubated with indicated primary antibodies in blocking solutions at 4 °C for two days. After five PBS washes, the sections were incubated with secondary antibodies and DAPI in a blocking solution for 1.5-2 h at room temperature. After five PBS washes, the sections were mounted with Aqua-Poly/Mount (18606-5; Polysciences Inc.). Dissociated MNs from iPSC differentiation were fixed and permeabilized in 0.1% Triton X-100/PBS for 5-10 min and then blocked in 10% FBS in 0.1% Triton X-100/PBS for 30 min. The sections were incubated with indicated primary antibodies in a blocking solution at 4 °C overnight. After three PBS washes, the sections were incubated with secondary antibodies and DAPI in a blocking solution for 45 min at room temperature.

### Spinal motor neuron quantification

At each indicated stage, ChAT$^{on}$ MNs in the lumbar ventral horns with DAPI nuclear signal were counted on one side of a 20-μm-thick sectioned spinal cord. MNs that did not show regular nuclear shapes were excluded. The quantification bar charts represent average MN counts from at least three sections per mouse ($n \geq 3$ mice) of the same age and genotype.

### Survival analyses

Mice were assessed weekly for baseline weight from P60. The onset of weight decline is defined as the age at which the animal lost 5% of peak body weight. We used the Kaplan−Meier method with a log-rank test to compare onset-free survival for each group. The disease end-point was defined as the day when mice could not right themselves within 15 s[70], when they were sacrificed. Survival analysis was performed using Kaplan−Meier analysis.

### NMJ analysis

To reveal neuromuscular junctions (NMJs) in whole-mount muscles, tissues were permeabilized and blocked in 3% bovine serum albumin (BSA) in 2% Triton X-100/PBS at room temperature for two hours, followed by incubation with anti-neurofilament antibody (DSHB, 1:

250) together with anti-SV2 (DSHB, 1: 500) to label axonal endfeet in blocking buffer at 4 °C for three days. After five 2% Triton X-100/PBS washes, the muscle tissues were incubated overnight at 4 °C with secondary antibodies, together with Alexa Fluor 555 labeled α-bungarotoxin [α-BTX, Thermo Fisher Scientific, B13422] to detect nicotinic acetylcholine receptors (AChR). Muscles were teased apart and flattened before mounting on slides. Z-stack images were acquired using a Zeiss LSM780 confocal microscope. All figures containing confocal images are projections of Z stacks. Innervated NMJs were counted when the AChR[on] endplates overlapped with axon terminals (as revealed by neurofilament staining), whereas the denervation ratio was calculated according to the colocalization between neurofilament, SV2 and the α-BTX signals of each picture, with values normalized against total endplate area. All NMJ areas were analyzed in ImageJ using threshold adjustment. Herein, we only examined the total denervation ratio, as represented by marker non-colocalization in NMJs.

### Behavioral assays
Locomotor activity was measured according to open field, rotarod, and treadmill tests. Age-matched wild-type (WT) mice from the littermate control line were used for experimental comparisons. Both sexes of indicated ages (P40, P70, P100, P130, and P160 for *ChAT-Cre; Mettl14^floxed* mice) were used in this study. The rotarod, grip strength, and CMAP tests were conducted in *SOD1^G93A* mice one week prior to intrathecal injection (P60) and subsequently assessed at the specified time points (P120, P140, and P160). The experimenters conducting all behavioral assays were blind to mouse genotypes.

### Rotarod
A commercially available rotarod apparatus (47600 Rota-Rod, Ugo Basile, Italy) with a rotating rod of 5 cm diameter was used. Mice were transferred to the testing room and habituated in the home cage at least 15 minutes before testing. In the training phase, three trials with a constant speed of 4 rpm and a 60-s cut-off time were used to ensure that all test mice could stay on the rod for a training trial before moving to the test phase. After a 30-minute rest interval, the mice were evaluated during the test phase, with the rod accelerating from 4 to 40 rpm with a 300-second cut-off time in a series of three trials. The longest falling latency, as well as the rotating speed when the mouse fell off the apparatus, were used to represent the motor coordination of each mouse[71].

### Grip strength
A grip strength meter (MK-380CM/R; Muromachi) was used to measure forelimb grip strength. As a mouse grasped the bar, the peak pull force in grams was recorded on a digital force transducer. During the test, a mouse was allowed to grasp the bar mounted on the force gauge. We performed four consecutive measurements per test at one-minute intervals.

### Open-field test
A square arena with opaque walls (area 48 × 48 cm and height 35 cm) was used. Mice were transferred to the testing room and habituated in the home cage one hour before testing and then allowed to explore the test chamber for another hour during which all behaviors were videotaped and tracked using a video-tracking system mounted on top of the arena (Clever System, Reston, VA). Total distance and average velocity were analyzed for each mouse[71].

### Treadmill locomotion analysis
A TreadScan apparatus (CleverSys, Reston, VA) was used to analyze gait. Mice were placed on a stationary treadmill for acclimation and trained at a speed of 10 cm/s for 5 minutes before testing. Four test speeds were analyzed (10, 15, 20, and 25 cm/s) for each trial, which were recorded at 79 frames/s for 10 s using TreadScan software. For data analyses, the successful trials in which a mouse was able to maintain treadmill speed with continuous locomotion for each 10-s recording was selected and further analyzed using TreadScan software. Only gait analyses from the trials conducted at 15 cm/s are shown in the present study. The gait parameters of stride for each limb were automatically and unbiasedly calculated, and average values were used for statistical analysis.

### Mouse motor neuron differentiation
Mouse embryonic stem cells were differentiated into MNs following published protocols[38,41]. Specifically, in this condition, we cultured the 3D embryoid bodies for five days, then attached the culture of the embryoid bodies to poly-ornithine and laminin-coated plates with an enrichment of neurotrophic factors (30 ng/mL GDNF, Peprotech) and cultured them for an extra 5 days (Day12). Change the medium every two days. Mouse ESC-MNs were used for imaging-based experiments, qPCR, and Nanopore direct RNA-seq.

### Quantification of neurite thickness, neurite complexity, and synaptic puncta numbers
Using an automated image analysis method, the synaptic puncta labeled with SynI and the neurite labeled with Smi32 were measured. For synapse quantification, images were processed with ImageJ and ilastik software to binarize the images and eliminate the background and embryonic body, resulting in an image featuring black synapses and neurites on a white background. The Particle Analyzer algorithm in ImageJ was utilized to quantify the number of synaptic puncta within the neurites, setting parameters for size (0–100 pixels) and circularity (0.2–1.0). The neurite length was measured using the Analyze Skeleton algorithm in ImageJ. The total number of the identified synaptic puncta was then divided by the overall neurite length to calculate the average number of synapses. For the neurite complexity and thickness measurement, the neurite outside the embryonic body was imaged. Neurite diameter was measured manually at the segment of each neurite within the fields with ImageJ's line tool. The image was processed with ImageJ and ilastik software to quantify the branch point and analyzed using the Analyze Skeleton algorithm in ImageJ. The total number of branch points was then divided by the overall neurite length to calculate the complexity.

### Nanopore direct RNA-seq and data analysis
Nanopore direct RNA-seq was conducted following instructions provided by Oxford Nanopore Technologies (Oxford, UK) using a Direct RNA Sequencing Kit (SQK-RNA002) and MinION flowcells (FLO-MIN106 R9 version). After live base-calling using Guppy (22.10.7) in MinKNOW, reads that passed the quality threshold were subjected to post-run base-calling with the latest version of Guppy (6.3.9) under default parameters. A reference transcriptome file was generated from the mouse genome reference and corresponding annotation. The Nanopore data provided in FASTQ format were initially aligned to the mouse genome reference from GenCode M20 using the minimap2 aligner. The resulting SAM format output was then converted to a sorted BAM format using samtools. To predict m⁶A modification sites and stoichiometry, we employed both EpiNano[42] and m6Anet[43]. For EpiNano prediction, the output was first filtered with the DRACH motif. Then, the probability of modification was taken from the 'ProbM' column and the results from the 'prediction' column, and then values > 0.5 were filtered for the results. For m6Anet prediction, the probability was yielded from the 'probability_modified' column and then filtered for values > 0.9 for the results. Each tool predicted m⁶A sites independently, which were subsequently converted into BED format for further analysis. To refine our predictions, we cross-referenced predicted sites from EpiNano and m6Anet for each sample. We then identified the shared sites across EpiNano and m6Anet, finalizing our list of predicted m⁶A sites. Each m⁶A modified site was

annotated using Homer v4.11 to determine its genomic location, categorizing them as Transcription Start Site (TSS), Transcription Termination Site (TTS), Exon (Coding), 5′ UTR Exon, 3′ UTR Exon, Intronic, or Intergenic. To estimate the predicted site with stoichiometry, the modification results from the "mod_ratio" column to quantify the percentage of reads in a given site.

## m⁶A immunoprecipitation and quantitative real-time PCR

To verify m⁶A sites, total RNA was isolated from mESC-MNs using Trizol (Life Technologies). Total RNA samples were pooled to a total RNA amount of ~75 μg/sample and processed using Invitrogen PolyA+ RNA selection (Dynabeads mRNA Purification Kit, Cat. #61006) according to the manufacturer's protocol. Input mRNA (1%) was reserved for reverse transcription. Magnetic A beads (#88802, Thermofisher Scientific) were prepared and washed, adding 2 μl of N⁶-Methyladenosine Antibody/per sample. m⁶A-modified transcripts of interest were pulled down using a rabbit polyclonal anti-m⁶A antibody (Synaptic Systems). The m⁶A-modified mRNAs were competitively eluted from beads using N⁶-Methyladenosine 5′-monophosphate sodium salt. Input m⁶A pull-down mRNA was reverse transcribed by using the SuperScript III First-Strand Synthesis System for RT-PCR (Thermo Fisher). cDNA was then used for SYBR-green-based quantitative real-time PCR. Enrichment of m⁶A-modified genes in m⁶A pull-down over input was calculated by comparing relative concentrations using Ct values ($2^{-Ct}$) and dividing each concentration by the relative concentration of the input. The concentrations of the immunoprecipitated RNA were then divided by the concentration in the input RNA and multiplied by 100, to obtain the percentage of transcripts in the m⁶A immunoprecipitate relative to the input. This value was then normalized to low enrichment in m⁶A-modified genes, which was also calculated using relative concentrations to determine a percentage of the input. Primers used are listed in Supplementary Data 6.

## Nuclei collection for single-nucleus multiome

Mice were euthanized with isoflurane according to IACUC Academica Sinica guidelines. Mice were decapitated and spinal cords were extracted using a hydraulic extrusion approach. To do so, a blunt 25G ¼ inch needle filled with ice-cold 1 × PBS was placed into the caudal end of the vertebral column and a rapid hydraulic pressure was introduced to extract the entire spinal cord. Lumbar segments were dissected based on morphology and dissociated two at a time in 2 mL pre-chilled lysis buffer with a 7 mL Dounce Homogenizer (pestle A: eight strokes; pestle B: 5 strokes). An additional 3 mL lysis buffer was added, and the entire homogenate was filtered through a 100 μm strainer. Then, 5 mL of 50% iodixanol was added to create a 10 mL suspension of 25% iodixanol. The suspension was inverted to mix well before equally splitting into two 15 mL tubes. Iodixanol (40%, 2 mL) was layered gently at the bottom of the homogenate and centrifuged at $1000 \times g$ for 12 minutes at 4 °C, with a swing bucket centrifuge (brand). The interphase nuclei were transferred to a new 15 mL tube and resuspended with 1:1 volume of 10 × wash buffer (10 mM Tris-HCl pH7.4, 10 mM NaCl, 3 mM MgCl₂, 1% BSA, 0.1% Tween 20, 1 mM DTT, and 1 U/μL RNase inhibitor), followed by centrifuging at $500\,g$ for 10 minutes at 4 °C. The supernatant was discarded and the pellet was resuspended in 800 μL diluted nuclei buffer (20× Nuclei Buffer from Chromium Next GEM Single Cell Multiome ATAC + Gene Expression Kit supplemented with 1 mM DTT, and 1 U/μL RNase inhibitor), before conducting FANS (fluorescence-activated nuclei sorting) using a BD FACS Aria III cell sorter (BD BioSciences, USA). GFP^on nuclei were collected in diluted nuclei buffer and centrifuged at $500 \times g$ for 7 minutes at 4 °C. Nuclei were adjusted to an appropriate concentration following the manufacturer's protocol (10x Genomics, 1000285) with diluted nuclei buffer. The quality of nuclei was examined using an AxioImager Z1 upright microscope (Zeiss, Germany) at 40-fold magnification.

## Single-nucleus RNA and ATAC sequencing

Transposition, nuclei isolation, and single-nucleus multiome libraries were prepared following the Chromium Next GEM Single Cell Multiome ATAC + Gene Expression protocol CG000338 Rev F (10x Genomics, Pleasanton, CA). The generated snRNA and ATAC libraries were sequenced to a minimum depth of 25,000 and 35,000 mean paired-end reads per nuclei, respectively, using a Nextseq 500 sequencer (RNA: 28 – 10 – 10 – 90 bps (base pairs), ATAC: 50 – 8 – 8 using an Illumina Nextseq 500 sequencer (Illumina, Inc., USA) and 16 – 49 bps). Raw sequencing data were processed using the standard Cell Ranger ARC pipeline (version 7.0.0, 10x Genomics) for demultiplexing, mapping to the mm10 reference genome, barcode, and UMI (unique molecular identifier) counting, and generating the gene count matrix.

## Joint processing of snRNA and snATACseq data

Analysis was performed in RStudio and R (version 3.6.1) using multiple packages: Seurat, Signac, DoubletFinder, ClusterProfiler, ggplot2. Sample quality was examined to ensure the nuclei conformed to the following criteria: based on gene expression data - nFeature_RNA < 10,000, nFeature_RNA > 200, and nCount_RNA > 500; and based on chromatin accessibility data - pct_reads_in_peaks > 15, blacklist_fraction <5, nucleosome_signal <4, and TSS.enrichment > 1. Doublets were estimated using DoubletFinder and removed from the analysis. The ATAC counts were normalized with TFIDF (term frequency-inverse document frequency), followed by dimensionality reduction with SVD (singular value decomposition). Clustering was performed using the second to tenth LSI (Latent Semantic Indexing) components. The gene expression counts were normalized whereby the expression value of each gene was divided by the total expression in each cell, multiplied by 10,000, and log-transformed. Highly variable genes across nuclei were selected and scaled. PCA (principal component analysis) was performed, and the top 30 principal components were selected for downstream clustering, as determined using an elbow plot. ATAC and RNA libraries for each sample were integrated using reciprocal LSI and MNN-CCA (Mutual Nearest Neighbor - Canonical Correlation Analysis). A WNN (weighted nearest neighbor) graph represented a weighted combination of RNA and ATAC assays with the first 60 and the second to tenth dimensions, respectively. Unbiased clustering of the RNA-ATAC integrated data at a resolution of 0.3 was used for cluster annotation. Major clusters were annotated using Seurat label transfer prediction scores with published single-cell RNA-seq clusters as reference[45,46].

## Differential expression and chromatin accessibility

Based on gene expression data, differential expression analysis was performed using the Seurat function FindMarkers and applying a likelihood-ratio test. Genes with Bonferroni-adjusted p-values of less than 0.05 were considered to be differentially expressed between *Sun1^sfGFP; ChAT-Cre; Mettl14^floxed* and littermate control. Differential chromatin accessibility analysis, based on chromatin accessibility data, was conducted using the Presto function wilcoxauc. In Presto, the p-value of the Wilcoxon rank sum test is computed based on Gaussian approximation and further adjusted by the Benjamini-Hochberg method. A peak is considered to be open in one condition and closed in the other if the adjusted p-value is less than 0.01, the logarithmic (base 2) fold-change is greater than 0.1, and the percentage of cells with a non-zero value in the first condition is greater than 5. Otherwise, the peak was denoted as neutral.

## Peak annotation

Peak annotation, i.e., to determine if a peak is in the transcription start site (TSS), transcription termination site (TTS), Exon, 5′ UTR Exon, 3′ UTR Exon, intronic, or intergenic region was performed by HOMER using the annotatePeaks.pl command on the mm10 genome reference.

## Identification of gene-to-peak linkage

Identification of gene-to-peak linkage was performed for each cell type using ArchR's framework. First, to reduce the sparsity of chromatin data, aggregation of similar cells was performed using the k-nearest neighbor graph based on LSI reduction, and the cell aggregate-by-peak and cell aggregate-by-gene matrices were created and log-normalized. Aggregates with greater than 80% overlap with any other aggregates were filtered out. Then, all possible peak-to-gene pairs were selected if the peak was located within ± 250 kilobase pairs flanking the gene start. Finally, the Pearson correlation was calculated across all cell aggregates for each possible peak-gene pair. Peak-to-gene linkages were then defined as those with Pearson correlation coefficients no less than 0.45 and adjusted $p$-values (by Benjamini–Hochberg method) no greater than 0.0001.

## H3K9me3 and γH2AX quantification

At the P120 stage, the signal of H3K9me3 or γH2AX was quantified (ImageJ) after training by machine learning (ilastik), and divided into the area of DAPI in each ChAT[on] MN. Both the strength of the laser and the exposure time for H3K9me3 or γH2AX were fixed to facilitate quantification. In Figs. 7c, e, 10b, and Supplementary 13b, each point represents the quantified result for each ChAT[on] MN, and those quantifications were counted from at least three mice of the same age and genotype.

## Bulk RNA-seq

For bulk RNA-seq, RNA was extracted from ALS iPSC-MNs, and its quality was evaluated using a Bioanalyzer 2100 RNA pico kit. The cDNA libraries were prepared from the human iPSC-MNs according to the manufacturer's instructions for the TruSeq Stranded mRNA library prep kit (Illumina). The concentration and size distribution of the completed libraries were determined using an Agilent Bioanalyzer 2100 DNA high-sensitivity kit and Qubit fluorometry (Invitrogen). Libraries were sequenced by following Illumina's standard protocol using the Illumina NextSeq500 HighOutput kit V2.5. For the gene expression z score analysis (including published data revealed in Supplementary Fig. 1) by aligning raw RNA-seq data to the GRCh38 genome using the STAR aligner. Subsequently, gene expression was estimated using the GenCode v41 gene annotation file (gtf), focusing on gene types categorized as protein_coding and lncRNA. Genes originating from the mitochondrial genome (chrM) were also excluded from further analysis. The expression levels of each gene were then quantified using the salmon tool. Next, differential expression analysis was conducted using the edgeR package. Raw read count information was processed to identify genes showing significant changes in expression across different conditions. The filterByExpr function was applied to remove genes with low expression levels to ensure robust results. The remaining genes were subjected to TMM normalization and dispersion estimation methods to identify differentially expressed genes accurately. Subsequently, genes were categorized as upregulated or downregulated based on fold change values. The expression trends of the same genes across different samples were analyzed by calculating z-scores, allowing for the observation of consistent or divergent expression patterns among compared samples. This analysis pipeline enabled the identification of differentially expressed genes and provided insights into their expression trends across various conditions.

## scAAV9 plasmid construction

To construct scAAV9-sh*Fto* plasmid, a 65 basepair short hairpin sequence targeting the fat mass and obesity-associated (*Fto*) gene (RNAi # TRCN0000183897) was subcloned into scAAV9-H1-CB-EGFP plasmid. A scAAV9-*EGFP* plasmid with the same backbone was provided by the AAV Core Facility in Academia Sinica. To verify the expression of the scAAV9-sh*Fto* construct, we used pLKO.1-sh*Fto* and pLKO.1-sh-Scramble (as a control group) and packaged them into lentivirus from the RNAi Core Facility of Academia Sinica. We transduced lentivirus (at a multiplicity of infection of 100) with 8 mg/mL protamine sulfate (Sigma-Aldrich) in a growth medium (DMEM) containing 20% fetal bovine serum. After 48 h of transduction, the medium was changed with the selection drug (8 μg/mL puromycin) for one week. We collected RNA/protein of puromycin-selected C2C12 cells, and levels of *Fto* were determined by qPCR/Western blotting, with m⁶A mRNA methylation levels being determined by m⁶A ELISA.

## scAAV9 virus preparation and intrathecal injection

scAAV9-sh*Fto* and scAAV9-*EGFP* were packaged by the AAV Core Facility in Academia Sinica. For intrathecal injection, mice at P60 were anesthetized by isoflurane. The lumbar spine was exposed through a 1.5 cm window surgically cut on the back of the mice. We injected 20 μL of either scAAV9-*EGFP* or scAAV9-sh*Fto* ($5 \times 10^9$ vg/μL) at a rate of 4 μL/minute into the groove between the L6 and S1 segments using a 27 G needle. A flick of the mouse's tail indicated a successful injection.

## Compound muscle action potential (CMAP)

Evoked CMAP of gastrocnemius muscle was measured on isoflurane-anesthetized mice at the indicated age (P60, P120, P140, P160). Before recording, hairs on the right hind limb and the lower back were shaved and completely removed using a depilatory cream. A reference recording electrode was then placed on the ankle of the right hind limb, and an active recording electrode was placed on the belly of the right gastrocnemius. To stimulate the sciatic nerve, a pair of stimulating anode and cathode needles were inserted near the sciatic nerve at the ipsilateral paraspinal site and the region of the proximal hind limb, respectively. A ground electrode was placed on the tail to minimize artifacts. To obtain maximal CMAP responses, we gradually increased the stimulus intensity from 3 mA and determined the supramaximal stimulation as ~120% of the stimulus intensity that no longer increased the response amplitude. The baseline-to-peak and peak-to-peak CMAP amplitudes evoked by supramaximal stimulation (mostly ~7 to 8 mA) were summed. We recorded at least 30 maximal CMAP responses for each mouse at each stage. The top five successful and maximal CMAP responses in each mouse were averaged and used for comparison between groups.

## Reporting summary

Further information on research design is available in the Nature Portfolio Reporting Summary linked to this article.

## Data availability

Requests for further information or resources should be directed to and will be filled by the lead authors, Y.-P.Y. and J.-A.C. All the sequencing data generated in this study have been deposited in the Gene Expression Omnibus repository under the GSE accession codes GSE290242, GSE290244, and GSE290245. Previously published human postmortem cortex RNA-seq and human iPSC-MNs are available under the accession codes GSE122649, GSE122650, GSE132972, and GSE173115. Transcriptome data from Answer ALS can be requested via the website: https://www.answerals.org/. Source data are provided with this paper.

## Code availability

The codes and processed data are available on GitHub at https://github.com/jaclab-multiomic/Yen-et-al.-2025-Nat-Comm and figshare at https://figshare.com/projects/Yen-et-al_-2025-Nat-Comm/238025.

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

## Acknowledgements

We appreciate Answer ALS for sharing the ALS patient's omics data. We thank the genomics core facility at the Institute of Molecular Biology, Academia Sinica for the Illumina sequencing and nanopore direct RNA sequencing. The flow cytometry core facility and imaging core at the Institute of Molecular Biology, Academia Sinica for cell sorting service and confocal imaging. The *Olig2-Cre* line was a kind gift from Prof. Tom Jessell (University of Columbia, USA). We particularly thank Prof. Hongjun Song and Prof. Guo-Li Ming (University of Pennsylvania) for sharing details of the initial *Mettl14^floxed* mouse strain. We thank the National RNAi Core Facility at Academia Sinica in Taiwan for providing *Fto*-shRNA-related plasmids and technical support for LV vector packaging. We also acknowledge the Taiwan Mouse Clinic and the AAV Core Facility of IBMS Academia Sinica for their technical support with mouse behavioral analyses (open field, rotarod, grip strength), X-ray imaging experiments, and scAAV9 vector packaging. We thank the Human Disease iPSC Service Consortium for iPSC generation and technical support. Y.-P.Y. is supported by an Academia Sinica Postdoctoral Fellowship (23-5g). The consortium is funded by the Ministry of Science and Technology (MOST) (MOST 110-2740-B-001-003). This work is funded by Academia Sinica (AS-GCP-113-L02 and AS-BRPT-113-01), the National Science & Technology Council (112-2326-B-001-001 -), and the National Health Research Institutes (NHRI-EX113-11330NI).

## Author contributions

Conceptualization, J.-A.C. and Y.-P.Y.; Methodology, E.-S.L., T.-H.L.,C.-C.W., G.-L.H., F.-Y.H., M.C., and Y.-P.Y., Software and Formal Analysis, E.-S.L., T.-H.L.,C.-C.W., G.-L.H., F.-Y.H., Z.-D.Y., C.-Y.H., W.Z., J.-H.H., Q.N., and Y.-P.Y.; Investigation, T.-H.L.,C.-C.W., G.-L.H., F.-Y.H., M.C., and Y.-P.Y., and J.-A.C.; Resources, Z.Z. and C.H.; Writing – Original Draft, J.-A.C. and Y.-P.Y.; Writing – Review & Editing, J.-A.C., Y.-P.Y., and C.H.; Funding Acquisition, J.-A.C. and Y.-P.Y.; Supervision, J.-A.C. and Y.-P.Y.

## Competing interests

The authors declare no competing interests. J.-A.C. and Y.-P.Y. declare a PCT and a US patent application related to this work.
