## [Transparent Peer Review file · Nature Communications]

The Motor Neuron m6A Repertoire Governs Neuronal Homeostasis and FTO inhibition Mitigates ALS Symptom Manifestation

Corresponding Author: Professor Jun-An Chen

Version 0:

Reviewer comments:

Reviewer #1

(Remarks to the Author)

In this study, Chen et al. underscored the detrimental role of m6A dysregulation during motor neuron degeneration in ALS pathogenesis. Conditional knockout of *Mettl14* in mouse motor neurons replicated key ALS disease characteristics, suggesting a significant contribution of m6A to disease development. They demonstrated that dysregulation of m6A-tagged genes could result in abnormal expression of high-risk genes associated with ALS and reduced chromatin accessibility in motor neurons. Additionally, diminished m6A levels were observed in induced iPSC-derived motor neurons from ALS patients, highlighting the detrimental effects of m6A dysregulation in human disease. Restoring m6A equilibrium via small molecule or gene therapy showed promise in preserving motor neurons and alleviating motor impairments in ALS models, suggesting potential therapeutic avenues targeting RNA modifications. One major concern arises from the possibility that the mitigated phenotypes of the SOD1G93A ALS mouse model might be attributed to the systematic elevation of m6A in other cell types following AAV-shFTO treatment. While offering interesting insights into the negative impact of m6A on ALS disease modeling, several points need improvement and clarification to be published in Nature Communications.

Major comments.

1) It needs clarification as to why mouse mESC-derived motor neurons were chosen for Nanopore direct RNA-seq in Figure 4 instead of using in vivo tissue or human iPSCs, especially considering that isolated nuclei from the lumbar region of the in vivo spinal cord were utilized for single-nucleus multiome analysis.

2) It is necessary to explain why CPA was used to induce ER stress in differentiated cell lines, particularly in terms of whether the CPA treatment can accurately replicate the phenotypes observed in ALS disease progression.

3) In Figure 5, panel D, both H3K9me3 and gamma-H2AX are upregulated in the tissue of the *Mettl14* cKO model. This raises the question of whether these epigenetic abnormalities are potentially common features of ALS pathogenesis. It would be valuable to investigate whether the upregulation of both molecular markers occurs in ALS mouse models (such as SOD1G93A) or patient-derived iPSC lines, as well as in ALS patients themselves. Additionally, it is very helpful to determine whether H3K9me3 upregulation and gamma-H2AX upregulation are rescued through FTO inhibitor treatment in vitro or in vivo models employing gene therapy using shFto.

4) In Figure 5, panel F, the skeletal motor neuron population exhibited a significantly higher increase in the closed form of chromatin accessibility compared to other cell populations. It is worth investigating whether H3K9me3, as an epigenetic marker associated with chromatin compaction, also follows this tendency. If so, it would be pertinent to explore whether the up-regulated genes identified from GO/KEGG analysis have any relationship to the increase in the closed form of chromatin accessibility observed in skeletal motor neurons.

5) In Figure 6, could the level of FOXP1 expression serve as a representative phenotype in ALS? Assessing other ALS-related phenotypes, such as neurite complexity (Figure 1M), cell death, or ER stress markers, would be helpful to strengthen

the conclusion.

6) Gene therapy using shFto appears to lack cellular specificity (especially considering Figure 7C and Figure S9D), raising concerns about its potential effects on other cell types, such as astrocytes and microglia, given that Fto can be knocked down in all of these cells.

7) In Figure 7, within the context of preventing m6A depletion by regulating Fto, it is conceivable that mediators other than Mettl14 (e.g., Mettl16) may be involved in m6A modification. Moreover, it remains uncertain to what extent the levels of m6A, which have already been depleted by Mettl14 knockout, can be restored upon Fto knockdown. Did the knockdown of Fto indeed restore dysregulated gene expression, such as abnormally upregulated chromatin regulators and ALS pathway-related genes shown in Figure 5A and Figure S7?

8) In Figure 7, the authors described in the figure legend, as "The CMAP amplitude is already reduced in SOD1G93A mice at P60 and gradually declines further over time, whereas scAAV9-Fto-shRNA treatment significantly ameliorates neuromuscular function at P160." However, the actual dataset shows only marginal differences between the SOD1G93A group and the SODG93A; AAV-shFto group at P160, not at other time points. This description needs to be corrected.

Minor comments, including grammatical errors.

1) The alterations in neurite complexity depicted in Figure 1M require quantification.

2) Li et al. (PMID: 37365312) found that the basal levels of m6A and the expression of Mettl3 and Mettl14 were decreased in motor neurons derived from ALS patients' iPSCs compared to control cells. However, in this study, only the results of cells treated with CPA showed decreased m6A levels in Figure 1. Additionally, Supplementary Figure 1E, only compared CPA-treated and untreated cells, without showing differences in basal levels between control and ALS-derived cells before and after differentiation into motor neurons. It would be beneficial to investigate whether the basal expression levels of Mettl3 and Mettl14, as well as m6A levels, are downregulated upon differentiation into motor neurons in ALS-derived cells, as suggested by Li et al.

3) In Figure 1N, Foxp1+/ISL1+ neurons are significantly reduced upon Mettl3 inhibitor treatment. Are Foxp1+/ISL1+ neurons also decreased in the spinal cord of Mettl14 cKO mice, as shown in Figure 2C?

4) In Figure 4, panels C and D, the data depict both glial cells and neuronal cells. However, it is notable that other abundant cell populations, such as microglia and endothelial cells, are absent. It is necessary to clarify whether these cell types were intentionally excluded from the experimental steps or if their absence is due to limitations in the experimental design.

5) In Figure 5, panels B and D, the ChAT staining appears markedly different from Figure 2, panels C and E. In Figure 5, the ChAT staining appears as dot-shaped structures resembling a "salt and pepper" pattern, contrasting with the more continuous staining observed in Figure 2. An explanation for this discrepancy is needed.

6) Figure 7F is not well described in the text and the figure legend. Is this staining performed using an m6A antibody? What is the author's interpretation of these data?

7) In Figure 4, Panel B, the meanings of R1, R2, and R3 should be described.

8) In Figure 5, panel D, it appears that some gamma-H2AX signals are not overlapped with DAPI staining. These regions may either delineate only the DAPI portion on the gamma-H2AX images with boundary or display images with DAPI completely overlapped. It would be necessary to provide a figure demonstrating DAPI and co-localization with certainty. (Refer to PMID: 32737294; PMID: 32615088)

9) In Figure 7F-I, the time points of the experiments performed were not described.

Reviewer #2

(Remarks to the Author)

Chen and colleagues investigated the roles of RNA N6-methyladenosine (m6A) modification by methyltransferase complex METTL3/METTL4 in the pathogenesis of motor neuron degeneration in amyotrophic lateral sclerosis (ALS). The authors first showed that iPSC-derived motor neurons (MNs) with SOD1(L144F), C9ORF72(exp), or TDP-43(G293S) mutation exhibited significant reductions in m6A RNA modification after these cells were treated with ER stress inducer CPA (cyclopiazonic acid), which significantly reduced the relative abundance of m6A RNA and the number of iPSC-derived MNs. Knocking down METTL4 in SH-SY5Y cells using shRNA reduced m6A RNA and led to similar effects. To further determine the role of m6A in motor neurons in vivo, the authors generated conditional mutant mice where the Mettl4 gene was deleted using Olig2-Cre or ChAT-Cre. While the Olig2-Cre;Mettl4 CKO exhibited early postnatal lethality, the cause was not due to loss of

MNs but a significant reduction in the number of oligodendroglia. In contrast, ChAT-Cre;Mettl4 CKO mice exhibited adult-onset phenotypes resembling those seen in other ALS mouse models, including motor deficits, progressive loss of spinal MNs, loss of innervation at the neuromuscular junctions (NMJs), neuroinflammation, and loss of nuclear TDP-43 and accumulation of TDP-43 in the cytoplasm of spinal MNs. Using multiomic analyses, the authors revealed that the mechanism that promoted neuronal degeneration in ChAT-Cre;Mettl4 CKO mice was due to the down-regulation of genes related to axonogenesis and synapse organization and up-regulation of genes related to chromatin remodeling and histone modification. Many mis-regulated genes have been implicated in the pathogenesis of ALS. In support of these results, inhibiting demethylase FTO using FB23-2 in ALS iPSC-derived MNs improved survival after CPA treatment, whereas injecting SOD1(G93A) mice with AAV9-shFto delayed disease-onset and prolonged survival in these mice.

Overall, this is an interesting study that provides compelling evidence supporting the essential role of m6A RNA modification carried out by methyltransferase complex member METTL4 in the survival in spinal motor neurons. This study is also timely as its data will help clarify some of the conflicting data in the recent literature (e.g. McMillen et al vs Li et al, 2023). By combining results from iPSC-derived MNs and conditional mutant mice, this study supports that METTL4-mediated m6A RNA modification is essential for neuronal homeostasis and survival in postnatal life. The mislocalization of TDP-43 protein in the spinal motor neurons in ChAT-Cre;Mettl4 CKO (Figure 2G) is quite intriguing and thought-provoking. While there are many strengths in this study, there are a few areas where data presentation, organization, and clarity can be improved. Here are the specific comments and suggestions:

1. Figure 1: The data in Figure 1B-D and 1E-G are redundant and can be improved. For instance, the results in Figure 1B-D should be converted into line graphs and provide statistics to support the progressive decline of the relative abundance of m6A in iPSC-derived MNs, which contributes to neuronal loss. Once the panels in 1B-D are updated, the panels in 1E-G can be removed. Additionally, the authors should provide more details regarding how the quantification in Figure 1N was performed.

2. Figure 2: This figure can be further strengthened in the following areas: (1) Panel 2C: In addition to neuronal cell body, ChAT also labels cholinergic synapses. Based on the images provided, it seems that there is already a significant reduction in ChAT+ synapses at P70 and P120. This reduction is likely to contribute to the motor deficits in these young ages (Figure 3). This is an important phenotype that the authors should use other synaptic markers to quantify and support. These new data can further enhance the single-cell multiomic data (Figures 4-5). (2) Panel 2E-F: It will be important to demonstrate whether microgliosis occurs before or after neuronal (and synaptic) degeneration. (3) Panel 2G-H: The authors should provide more in-depth analyses on the very interesting TDP-43 mislocalization phenotypes, including when this phenotype can be detected, does cytoplasmic TDP-43 colocalize with other RNA binding proteins (e.g. FUS) or RNA granule markers. (4) The survival curves in Panel 2B showed the median survival is about 250 days. However, most of the histopathology was performed at P160. Why? Is there no further deterioration of pathology after P160? (5) The quantifications should be normalized by surface area or another parameter to keep the quantifications consistent and comparable between different conditions.

3. Figure 4: Panels A-B described the use of the Nanopore platform to characterize m6A RNA modifications. However, the authors chose to do this in mESC-derived MNs, not in the spinal cord or sorted MNs from ChAT-Cre;Mettl4 CKO mice. It is quite odd because the data from mESC-derived MNs (Figure 4B and Fig. S4) had no direct relevance to the phenotypes in ChAT-Cre;Mettl4 CKO mice. If it is not possible to perform Nanopore analysis using spinal cord or sorted MNs from ChAT-Cre;Mettl4 CKO mice, the authors should perform targeted analysis of m6A on ALS risk genes in using spinal cord or sorted MNs from ChAT-Cre;Mettl4 CKO mice. These data could become a full-fledged figure by themselves, separated from the single-cell multiomic data.

4. Figure 5: The ChAT confocal images in Panels B and D look very different from those in Figure 2E. This discrepancy should be addressed since the corresponding quantification is striking and critical for the validation of the RNA-seq dataset. In both Figures 4 and 5, the reviewer recommends adding the number of genes in DEGs and m6A site in the Venn diagrams to indicate how many genes are in each group.

5. Figure 6: Panel 6E presents RNA-seq data from control iPSC-derived MNs or MNs treated with ER stressor CPA or FTO inhibitor. The data presented here are highly selected and lack statistical analyses to support the conclusion. The authors should adopt more unbiased approaches to show that FTO inhibitor FB23-2 can indeed have broader impacts in correcting the transcriptomes of these iPSC-derived MNs.

6. Figure 7: The authors use several fALS patient iPSC~MN lines and one sALS iPSC~MN to show that restoring m6A homeostasis can rescue MN degeneration in both models. While the phenotype after treating with FB23-2, an inhibitor of FTO, is striking, this reviewer recommends including another control of adding FB23-2 to the vehicle to establish the role of the FTO inhibitor during homeostatic state. This control would also help delineate the significance of m6A homeostasis in control MNs without ALS-related mutations.

7. Finally, there are several areas in the manuscript where statements or conclusions went beyond the data. For instance, one page 7, last sentence of the 2nd paragraph the authors mentioned their data in ChAT-Cre;Mettl4 CKO "mirroring numerous prominent molecular pathology hallmarks displayed by human ALS patients." This is an overreaching statement because ALS is a very heterogeneous disease (sporadic and familial ALS alike). It is not clear whether the field agrees there are "molecular pathology hallmarks" for ALS.

(Remarks to the Author)

Reviewer #4

(Remarks to the Author)

Very interesting manuscript on potential roles of m6A in motor neurons in ALS. However the assays to measure m6A (on which the main conclusions were made) were flawed and not actually measuring m6A in mRNA (which is generated by METTL14). Instead total RNA was used, which measures m6A from other enzyme pathways unrelated to METTL14. Thus, some of the key conclusions are invalid. However the gene therapy/drug therapy with FTO is interesting. Comments below.

1. assays for m6A are not correct - the authors use total RNA for both m6A quantification and dot blots. Most of the m6A is m6A from rRNA not mRNA. The results of figure 1 are not reflective of reduced Mettl3 levels since they are looking at m6A not derived from this enzyme.

2. STM2457 exerts its own toxicity independent of METTL3 so data should be validated with other inhibitors (such as STC15). Concentration of STM2457 not indicated in legend or figure 1.

3. I am surprised by the use of ChAT-Cre, which affects many neuronal types unrelated to motor neurons. Also this is an undesirable promoter for studying motor neuron disease since it is activated embryonically (as early as E11.5) - phenotypes seen likely reflect defective differentiation programs. Perhaps this is a different strain than the usual ChAT-Cre (see <https://pubmed.ncbi.nlm.nih.gov/23649862/>) - is this one somehow activated in adulthood? Details are not provided for the claimed late activation so this should be clarified since ChAT is an embryonic promoter. The correct promoter is a motor neuron-specific promoter with adult onset or inducible promoter (e.g. tamoxifen). If this is an embryonic or perinatal promoter, Figure 2 has little relevance to adult motor neuron disease. The use of a developmental promoter is problematic since the m6A pathway regulates development and the phenotypes could be due to impaired development not function of motor neurons. If this is an early promoter, then the invalidity of this promoter should be discussed. The microglial activation was interesting and can be mentioned but again, it is not clear if this is due to improper MN development or death of these neurons caused by defective development.

4. Fig 3 - as with Figure 2, the possibly incorrect Cre driver raises the possibility that all the phenotypes derive from impaired differentiation of motor neurons to altered (but related cell types) rather than proof that m6A is needed to maintain motor neuron function.

5. FTO gene therapy is interesting. m6A nanopore maps should be prepared to document increase in m6A site stoichiometry. Otherwise we cannot know any potential mechanism. Fto can also affect m6Am and even m1A and this is cell-type specific so nanopore can be useful to see if in this case if m6A is affected, and on which mRNAs. m6Am immunoblots on small RNA can help to show if m6Am levels increase in these FTO gene therapy treatments and thus constitutes a potential alternative mechanism. Please note that again the m6A assay seems to be on total RNA so cannot be used.

6. For chromatin data, please address whether the change in chromatin accessibility is in m6A genes, or if it is unrelated to m6A-containing transcripts.

7. The m6A maps from nanopore are interesting, but seem to be identical to other neuronal maps of m6A made using other methods. It is not clear if any new data is obtained from these maps. Greater clarity would help on this. What is new relative to other studies of m6A in motor neurons/ALS?

Version 1:

Reviewer comments:

Reviewer #1

(Remarks to the Author)

Most of the previous concerns we raised were successfully addressed. One additional point is that the methods section does not describe the conditions and concentration of FTO treatment, which are crucial details for future clinical applications.

Reviewer #2

(Remarks to the Author)

The authors have done an outstanding job in addressing all the comments I raised. The revised manuscript is now much improved and should be suitable for publication in Nature Communications.

Reviewer #3

(Remarks to the Author)

Reviewer #4

(Remarks to the Author)

The manuscript addresses most of the key issues. It is unfortunate that the nanopore on the FTO-treated neurons could not be done.

My remaining comments are textual in nature:

1. The m6A ELISA method that the authors use to say that m6A is augmented does not distinguish between m6A and m6Am. The claim that augmented m6A is occurring is difficult to validate without using mass spectrometry. I would make sure that the authors clarify the limitations of this ELISA method, which is particularly important since m6Am can be changing, not m6A or not just m6A.
2. In many cases, the authors say things like "In addition, among the 81 identified ALS risk genes to date, our direct RNA sequencing demonstrated that 34 of them are m6A-modified". This might appear to be quite a lot of genes. But in fact, this could be evidence of m6A -not- being enriched in ALS risk genes. In many mapping studies 50% or more mRNAs have at least one m6A. So these numbers that the authors provide need to be provided along with the prevalence of m6A in their maps. For example, they could say something like "In our dataset only 15% of transcripts contained m6A, but 40% of ALS risk genes contain m6A (34 out of 81)." In every case when the authors point out enrichments or the reasons why m6A might mediate the effects of FTO inhibition, this sort of context should be provided so the reader can understand the significance of the results.
3. The title says "augmented m6A mitigates symptoms". But the more correct statement is "FTO depletion" (or inhibition) mitigates the symptoms, since this is what was tested. As mentioned above, the measurements of increased m6A were not done in a way that reflects m6A only, and the authors did not do the nanopore experiment. Mettl3 overexpression was not used, which could have supported this claim (if m6A also increased). The exact mechanism of FTO inhibition is often unclear, since FTO targets so many nucleotides. So the authors can be more precise by saying FTO inhibition mitigates these phenotypes.

We sincerely thank the reviewers for appreciating the importance of our study and for their insightful comments. Our point-by-point responses to their comments and descriptions of the new experiments and computational analyses that we have conducted are provided in this letter. The original reviewers' comments are in **black**, followed by our responses in **blue**. We highlight changes made to the text and figures in **yellow**, whereas **cyan** indicates the paragraph line numbers in the revised text. All changes are highlighted in the revised text. We have made changes to the figure numbering to address the reviewers' comments. To avoid confusion about the numbering, we provide a conversion table here for the reviewers' and editor's reference.

Table R1. Changes to figure numbering and notes for revision.

Old Numbering	New Numbering	Content	Note on changes during revision	In response to
Figure 1A~1G	Figure 1	Assessment of the hypo-m ⁶ A phenotype in ALS iPSC~MNs	Change layout of m ⁶ A %/MN differentiation index from (B~D) in ALS iPSC~MNs. From (E~G) Assess m ⁶ A mRNA levels in ALS iPSC~MNs.	Reviewer 1,2,3
Figure 1H~1N	Figure 2	Verification of the function of METTL3/14 in human MNs	Replace the SY-H5Y line data with METTL3/14 KD experiments (F and G). Analyze neurodegeneration using an index (G and H).	Reviewer 1,2,3
Figure 1L~N	Figure 2	Verification of the function of the METTL3 inhibitor (STM2457) in WT human iPSC~MNs	Assess m ⁶ A mRNA levels in STM2457-treated WT iPSC~MNs by ELISA (B) and m ⁶ A mRNA dot blot (C). Analyze neurodegeneration using an index (D and E).	Reviewer 1,2,3
Figure 2A~2D	Figure 3	MN phenotype characterization in Chat-Cre; Mettl14^{flxed} mice	Add new stages from P30~P250 (C~F), and further quantify the C bouton phenotype in (E and F).	Reviewer 1,2
Figure 2E~2L	Figure 4	Iba1, Tdp43, and NMJ characterization in Chat-Cre; Mettl14^{flxed} mice	Unchanged.	Reviewer 2
Figure 3	Figure 5	Behavior analysis of Chat-Cre; Mettl14^{flxed} mice	Unchanged.	
Figure 4A~4B	Figure 6	MN m ⁶ A epitranscriptome analysis and verification	Mature ESC~MN characterization (A~C), mature mouse ESC~MNs gene expression from nanopore direct RNA-seq (E), and m ⁶ A pull-down qPCR verification for ALS risk gene with m ⁶ A-modified (I and J)	Reviewer 1,2,3
Figure 4C~4H	Figure 7	Single nuclei multiome analysis on Sun1^{sfGFP}; Chat-Cre; Mettl14^{flxed} mice	Unchanged.	Reviewer 2

Figure 5	Figure 8	Chromatin change analysis on Chat-Cre; Mettl14^{flxed} mice	Replace high resolution images for H3K9me3 (B) and γ H2AX (D).	Reviewer 1,2
Figure 6	Figure 9	FTO inhibitor treatment in the ALS iPSC~MNs	Add vehicle with FTO inhibitor (B~D), assess the m ⁶ A mRNA levels in ALS iPSC~MNs (C), and analyze neurodegeneration using an index (D).	Reviewer 2,3
Figure 7	Figure 10	scAAV9-sh Fto gene therapy in the SOD1^{G93A} ALS mouse model	Unchanged.	
	Figure 11	H3K9m3 level assessment after scAAV9-sh Fto gene therapy in the SOD1^{G93A} ALS mouse model	Examine H3K9me3 levels in the SOD1^{G93A} model and verify that the increase of H3K9me3 is abrogated upon scAAV9-sh Fto gene therapy.	Reviewer 2,3
Figure 8	Figure 12	Cartoon summary	Unchanged.	
Figure S1	Figure S1	RNA-modified enzyme expression from ALS patient data	Add METTL5/16 expression analysis in (B).	Reviewer 3
	Figure S2	METTL3/14 knockdown characterizations	Assessment of METTL3/14 shRNA knockdown efficiency and m ⁶ A mRNA methylation level.	Reviewer 3
Figure S2	Figure S3	Olig2-Cre-Mettl14^{flxed} analysis	Unchanged.	
	Figure S4	Tdp43, Iba1, and Fus characterization in Chat-Cre; Mettl14^{flxed} mice	Profile phenotypes of Iba (A and B) and Tdp43 (C and D) in different phases of disease progression for the Chat-Cre; Mettl14^{flxed} mice. Verify other RNA-binding proteins in the Chat-Cre; Mettl14^{flxed} mice. Fus (E and F).	Reviewer 2
Figure S3	Figure S5	Motor behavior analysis of the Chat-Cre; Mettl14^{flxed} mice	Unchanged.	
Figure S4	Figure S6	Verification of newly identified m ⁶ A sites in MNs	Identify m ⁶ A sites from Nanopore and verify by m ⁶ A pull-down qPCR.	Reviewer 1,2,3
Figure S5~S7	Figure S7~9	Single nuclei multiome analysis on the Chat-Cre; Mettl14^{flxed} mice	Unchanged.	
Figure S8	Figure S10	Chromatin and DNA damage analysis on the Chat-Cre; Mettl14^{flxed} mice	Reannotated the images and added high- resolution DAPI image (A~C).	Reviewer 1,3
	Figure S11	FTO inhibitor treatment of ALS iPSC~MNs	DEG with GO analysis in ALS iPSC~MNs (A~C), and selected histone modification and chromatin remodeling gene change analysis in (D and E).	Reviewer 2,3
Figure S9	Figure S12	scAAV9 gene therapy characterization	Unchanged.	
	Figure S13	rH2AX analysis of SOD1^{G93A} mice	Examine γ H2AX levels in the SOD1^{G93A} model.	Reviewer 3

In addition to the new/revised 12 main and 13 supplementary figures shown in the manuscript, we also attach one specific figure to address the reviewers' comments (see below). We would be happy to incorporate this figure panel into the revised manuscript if it is felt that it is warranted.

Additional Revision Figures:

R1: Astrocyte marker distributions in the single nuclei RNAseq dataset.

Reviewer#1

In this study, Chen et al. underscored the detrimental role of m6A dysregulation during motor neuron degeneration in ALS pathogenesis. Conditional knockout of *Mettl14* in mouse motor neurons replicated key ALS disease characteristics, suggesting a significant contribution of m6A to disease development. They demonstrated that dysregulation of m6A-tagged genes could result in abnormal expression of high-risk genes associated with ALS and reduced chromatin accessibility in motor neurons. Additionally, diminished m6A levels were observed in induced iPSC-derived motor neurons from ALS patients, highlighting the detrimental effects of m6A dysregulation in human disease. Restoring m6A equilibrium via small molecule or gene therapy showed promise in preserving motor neurons and alleviating motor impairments in ALS models, suggesting potential therapeutic avenues targeting RNA modifications. One major concern arises from the possibility that the mitigated phenotypes of the SOD1G93A ALS mouse model might be attributed to the systematic elevation of m6A in other cell types following AAV-shFTO treatment. While offering interesting insights into the negative impact of m6A on ALS disease modeling, several points need improvement and clarification to be published in Nature Communications.

We very much appreciate this reviewer's important and constructive critiques. Below, we address all of his/her comments in detail.

Major comments.

1) It needs clarification as to why mouse mESC-derived motor neurons were chosen for Nanopore direct RNA-seq in Figure 4 instead of using in vivo tissue or human iPSCs, especially considering that isolated nuclei from the lumbar region of the in vivo spinal cord were utilized for single-nucleus multiome analysis.

This is an important issue, as Reviewer #2 raised the same concern. In our manuscript, we employed *Sun1^{sfGFP}; Chat-Cre; Mettl14^{flxed}* mice to isolate single nuclei from the lumbar spinal cords for multiome analysis. For this approach, we harvested nuclei from the lumbar region of the spinal cord and performed FACS to select GFP^{on} cells. In fact, we originally made several attempts to collect GFP^{on} nuclei for Nanopore direct RNA-seq to identify m⁶A sites in the spinal motor neurons (MNs). However, it proved a daunting task to obtain sufficient mRNA for this technique, considering that only < 2 % of cells in the adult spinal cord are GFP^{on}, and the harvest yield was extremely low after FACS enrichment. As an alternative approach, we used mouse ESC~MNs cultured in a conditioned medium that can drive these ESC~MNs to acquire an adult-like MN identity (Patel *et al*, 2022; Tung *et al*, 2019). Specifically, under our experimental conditions, we cultured the 3D embryoid bodies for five days, then attached the cultured embryoid bodies to plates coated with a supplement enriched in neurotrophic factors, and cultured them for an additional 5

days (Day 12). We describe this protocol in detail in our revised Methods section. In the revised manuscript, we have now further verified that several genes characteristic of adult MNs (*Slc5a7*, *Syn1*, *Fos*, *Col5a3*, *Dmp1*, *Bag3*, etc.; (Patel *et al.*, 2022) are robustly expressed in our mature ESC~MNs (new Figures 6A~6C). Consistently, our Nanopore direct RNA-seq generated similar results (new Figure 6E). Collectively, these new characterizations indicate that the mature ESC~MNs cultured in conditioned mature media largely reflect many adult MN characteristics used for single-nuclei multiome analysis (new Figure 7).

As the reviewer indicates, it might be more clinically relevant to use human iPSC-derived MNs. However, we reasoned that the m⁶A epitranscriptome might not be identical across species, adding an extra layer of complexity to our analysis. In our revised manuscript, we have added a few paragraphs to clarify these issues and limitations (lines 229-267):

“To gain insights into the potential mechanisms underlying how hypo-m⁶A promotes MN degeneration, we aimed to systematically identify the dysregulated genes possessing m⁶A modifications in our *Chat-Cre; Mettl14^{flox}* mice (Figures 6 and 7). To identify m⁶A-modified transcripts, we adopted a direct RNA sequencing platform, which enables the identification of the MN m⁶A epitranscriptome at single-nucleotide resolution (Figure 6). Since it is technically challenging to obtain sufficient adult MNs from the spinal cord for direct RNA sequencing, we employed an enhanced method using mouse ESC-derived MNs, matured with a conditioned medium (Figures 6A~6C) (Patel *et al.*, 2022; Tung *et al.*, 2019). First, we confirmed that this approach successfully generated MNs expressing mature neuronal markers, with longer and more mature neurite structures revealed by discrete Syn1-positive puncta (Figures 6B and 6C). We then subjected these mature MNs to the ONT Nanopore platform, which provides a powerful framework for detecting RNA modifications through advanced machine-learning algorithms applied to sequencing metrics (Figure 6D, details in Methods). As expected, several adult mature MN genes are abundantly expressed in the Nanopore direct RNA-seq results (Figure 6E). We subsequently employed two supervised machine learning tools, namely EpiNano and m6Anet (Hendra *et al.*, 2022; Liu *et al.*, 2021), which set a stringent criterion that the predicted m⁶A sites need to occur in at least two samples for either one of the algorithms (see Methods for details). We identified 30,340 high-confidence m⁶A modification sites (probability > 0.5) corresponding to 7,921 genes (Figure 6D, genes of interest are illustrated in Table S1; refer to the Methods section for details). Interestingly, a deeper estimate of the predicted m⁶A stoichiometry indicated that the high variation levels are displayed across different m⁶A sites in the same gene (Table S3). Consistent with previous findings (Castro-Hernandez *et al.*, 2023), our analysis also revealed enriched distributions of m⁶A sites in coding sequences (CDSs) and 3' UTRs, especially near the stop codons (Figure 6F). By analyzing the enrichment of Gene Ontology (GO) and KEGG pathways for the m⁶A-modified transcripts, we noticed a striking enrichment for ALS-related genes (Figure 6G). In addition, among the 81 identified ALS risk genes to date, our direct RNA sequencing demonstrated that 34 of them are m⁶A-modified ($p = 2.06 \times 10^{-6}$, one-tailed hypergeometric test; Figure 6H and Table S2).

To validate our computationally predicted m⁶A sites, we used *Tardbp* (*Tdp43*) as a benchmark to validate our methodology. Consistent with a previous report (McMillan *et al.*, 2023), we confirmed the existence of a previously identified m⁶A site in *Tardbp* with a high probability rate with stoichiometry ranging from 57.38 to 92.53 % (Table S3) and verified via m⁶A antibody pull-down

assay (Figure 6I, site 1). Additionally, we uncovered and validated an additional high m⁶A-modified site within the *Tardbp* transcript from Nanopore direct RNA-seq (Figure 6I, site 2). Moreover, we substantiated the existence of predicted m⁶A sites in *Atp13a2* (*Park9*) (Figure 6J) (Spataro et al, 2019), an ALS risk gene not previously shown to have m⁶A modifications in MNs. Finally, we further verified several newly identified m⁶A-modified sites in ALS risk genes, including *Dctn1*, *Epha4*, *C9orf72*, *Glt8d1*, *Cacna1h*, *Chrna3*, *Bscl2*, *Fig4*, *Hnrnpa2b1*, *Ubqln2*, *Hnrnpa1*, *Tuba4a*, *Sod1*, *Chmp2b*, and *PIKfyve* (Figure S6B). Thus, these observations confirm the sensitivity, accuracy, and reliability of Nanopore technology to identify m⁶A-modified sites in MNs.”

2) It is necessary to explain why CPA was used to induce ER stress in differentiated cell lines, particularly in terms of whether the CPA treatment can accurately replicate the phenotypes observed in ALS disease progression.

We appreciate the reviewer's feedback and apologize for not providing a clearer explanation regarding the use of cyclopiazonic acid (CPA), an endoplasmic reticulum stressor, to accelerate ALS disease progression. The rationale behind using this approach is that a previous report that adopted a large-scale small-molecule screen identified CPA as a selective stressor to accelerate degeneration of human *SOD1*^{G93A} iPSC~MNs but not wild-type controls (Thams et al., 2019). We have used this approach in a previous study (Tung et al., 2019), which revealed that *SOD1*^{G93A} iPSC~MNs undergo selective accelerated degeneration upon CPA treatment compared to control isogenic lines. In this study, we further show that CPA can be used as a selective marker to accelerate the degeneration of several other familial and sporadic ALS iPSC~MNs. To make this point clearer, we have revised the Results and Methods sections to address this (lines 147-154):

“To accelerate ALS disease progression, we applied cyclopiazonic acid (CPA), an endoplasmic reticulum stressor, as CPA has been shown previously to act as a selective stressor to accelerate the degeneration of human *SOD1*^{G93A} iPSC~MNs but not wild-type controls (Thams *et al*, 2019; Tung *et al.*, 2019). Consistent with this scenario, we found that all of our Ctrl-MNs were relatively resistant to CPA stress, unlike the ALS iPSC~MNs that exhibited drastic loss after seven days of CPA treatment (Figures 1B~1D). No obvious degeneration was displayed by either Ctrl or ALS iPSC~MNs on day 4 (Figures 1B~1D). Thus, we could capture the progressive MN degeneration displayed by the familial ALS iPSC lines.”

3) In Figure 5, panel D, both H3K9me3 and gamma-H2AX are upregulated in the tissue of the *Mettl14* cKO model. This raises the question of whether these epigenetic abnormalities are potentially common features of ALS pathogenesis. It would be valuable to investigate whether the upregulation of both molecular markers occurs in ALS mouse models (such as *SOD1*G93A) or patient-derived iPSC lines, as well as in ALS patients themselves. Additionally, it is very helpful to determine whether H3K9me3 upregulation and gamma-H2AX upregulation are rescued through FTO inhibitor treatment in vitro or in vivo models employing gene therapy using shFto.

Thank you for highlighting this issue. To address this point, we have now performed additional experiments for the revised manuscript (new Figure 11 and Figure S13). Specifically, we have assessed H3K9me3 and γ H2AX for four groups—wild type control (Ctrl), Ctrl with scAAV9-

sh*Fto* (Ctrl; scAAV9-sh*Fto*), *SOD1*^{G93A}, and *SOD1*^{G93A} with scAAV9-sh*Fto* (*SOD1*^{G93A}; scAAV9-sh*Fto*)—at postnatal day 140 (P140), i.e., when *SOD1*^{G93A} mice are still at the early symptomatic stage (new Figure 11). Strikingly, we observed a drastic dramatic increase in H3K9me3 in the ventral horn of the spinal cord in *SOD1*^{G93A} mice, which was rendered to the level manifested by Ctrl mice after scAAV9-sh*Fto* treatment (new Figure 11). This result supports that epigenetic abnormalities might be a novel feature of ALS pathogenesis. However, we did not observe any change in γ H2AX in the spinal cords of *SOD1*^{G93A} mice (new Figure S13). We explain this outcome in new sections of the revised Discussion. We agree with the reviewer that investigating if these epigenetic abnormalities are common features of ALS pathogenesis in human iPSC-derived MNs or patient tissues is an important issue. However, the extensive analysis required to undertake this task may exceed the scope of the current study. Such an investigation could potentially pave the way for new avenues of future research. We have highlighted these points in the revised Results (lines 422-431) and Discussion sections (lines 588-594), respectively, as follows:

“To determine if scAAV9-sh*Fto* treatment rescues *SOD1*^{G93A} mice through m⁶A-mediated molecular changes, such as histone modifications in H3K9me3 and γ H2AX, we assessed these markers in four groups—wild-type control (Ctrl), Ctrl treated with scAAV9-sh*Fto* (Ctrl; scAAV9-sh*Fto*), *SOD1*^{G93A}, and *SOD1*^{G93A} treated with scAAV9-sh*Fto* (*SOD1*^{G93A}; scAAV9-sh*Fto*)—at postnatal day 140 (P140), i.e., when *SOD1*^{G93A} mice are at an early symptomatic stage (Figure 11). Remarkably, H3K9me3 levels were significantly elevated in the ventral horn of the *SOD1*^{G93A} mice and were restored to control levels following scAAV9-sh*Fto* treatment (Figure 11), suggesting that epigenetic dysregulation may be a novel hallmark of ALS pathogenesis mediated by hypo-m⁶A. In contrast, no significant changes in γ H2AX were observed in the spinal cords of the *SOD1*^{G93A} mice (Figure S13).”

“Additionally, though we identified an increase in H3K9me3 and closed chromatin regions as potential new features of ALS, the precise mechanism of how hypo-m⁶A induces these chromatin changes remains unclear. It is also uncertain if the increase in H3K9me3 directly correlates with the closed chromatin regions. Interestingly, among the upregulated genes identified from single nuclei RNA-seq of the *Sun1*^{sfGFP}; *Chat*-Cre; *Mettl14*^{flxed} MNs, we noted that *Tasor*, one of three subunits forming the HUSH complex that recruits Setdb1 to read and deposit H3K9me3 (Tehasovnikarova *et al*, 2015), is a promising candidate to account for the increase of H3K9me3 we observed in the *SOD1*^{G93A} mice. Future large-scale investigations using human ALS iPSC~MNs will be crucial to further explore these potential pathological features. Moreover, it would be valuable to assess if combinatorial treatments targeting both H3K9me3 and m⁶A modifications could offer enhanced benefits in preventing MN degeneration.”

4) In Figure 5, panel F, the skeletal motor neuron population exhibited a significantly higher increase in the closed form of chromatin accessibility compared to other cell populations. It is worth investigating whether H3K9me3, as an epigenetic marker associated with chromatin compaction, also follows this tendency. If so, it would be pertinent to explore whether the up-regulated genes identified from GO/KEGG analysis have any relationship to the increase in the closed form of chromatin accessibility observed in skeletal motor neurons.

We thank the reviewer for this insightful suggestion. It is indeed a tempting direction to pursue. In our revised manuscript, we have examined H3K9me3 in the wild type control (Ctrl), Ctrl with scAAV9-sh*Fto* (Ctrl; scAAV9-sh*Fto*), *SOD1*^{G93A}, and *SOD1*^{G93A}; scAAV9-sh*Fto* lines at postnatal day 140 (P140), i.e., when the *SOD1*^{G93A} mice are still at the early symptomatic stage. Remarkably, H3K9me3 levels were significantly elevated in the ventral horn of the *SOD1*^{G93A} mice and were restored to control levels following scAAV9-sh*Fto* treatment (Figure 11), suggesting that epigenetic dysregulation may be a novel hallmark of ALS pathogenesis mediated by hypo-m⁶A.

To determine if the increase in H3K9me3 sites is closely linked to changes in closed chromatin regions in the *Chat-Cre; Mettl14*^{flxed} model, we would need to perform H3K9me3 ChIP-seq or CUT&RUN-seq and compare the results with the ATAC-seq data. However, this poses a significant challenge due to the scarcity of spinal motor neurons (< 2 %) *in vivo*. A potential alternative approach would be to use *Chat-Cre; Mettl14*^{flxed} ESC-derived MNs to conduct these experiments. However, doing so would require a timeframe of at least six months to one year to complete, which might be beyond the scope of the current manuscript. Accordingly, we acknowledge this limitation in our revised Discussion section (lines 588-594):

“Additionally, though we identified an increase in H3K9me3 and closed chromatin regions as potential new features of ALS, the precise mechanism of how hypo-m⁶A induces these chromatin changes remains unclear. It is also uncertain if the increase in H3K9me3 directly correlates with the closed chromatin regions. Interestingly, among the upregulated genes identified from single nuclei RNA-seq of the *Sun1*^{sfGFP}; *Chat-Cre; Mettl14*^{flxed} MNs, we noted that *Tasor*, one of three subunits forming the HUSH complex that recruits Setdb1 to read and deposit H3K9me3 (Tehasovnikarova *et al.*, 2015), is a promising candidate to account for the increase of H3K9me3 we observed in the *SOD1*^{G93A} mice. Future large-scale investigations using human ALS iPSC~MNs will be crucial to further explore these potential pathological features. Moreover, it would be valuable to assess if combinatorial treatments targeting both H3K9me3 and m⁶A modifications could offer enhanced benefits in preventing MN degeneration.”

We hope that these new experiments and revisions sufficiently address the reviewer’s concerns.

5) In Figure 6, could the level of FOXP1 expression serve as a representative phenotype in ALS? Assessing other ALS-related phenotypes, such as neurite complexity (Figure 1M), cell death, or ER stress markers, would be helpful to strengthen the conclusion.

We appreciate the reviewer’s comment. We used FOXP1, a marker for limb-innervating motor neurons present in the brachial and lumbar lateral motor columns (LMC), to assess human iPSC-derived motor neurons because the differentiation protocol we employed is known to produce a significant number of FOXP1^{on} neurons (Maury *et al.*, 2015; Tung *et al.*, 2019). As most previous studies have used SMI32 as a marker to assess MN numbers and neurite complexity, we have decided to use SMI32 only to quantify all results in this revision to prevent confusion. To accurately measure neurite degeneration, we have employed a systematic approach in the revised manuscript by analyzing images of randomly selected regions to assess neurite fragmentation. The level of neurite degeneration was quantified using a degeneration index (DI), which is the ratio of the fragmented neurite area to the total neurite area. We defined the DI as ranging from 0 (completely intact) to 1 (completely fragmented). This new analysis is now presented in new

Figures 1B~D, 2E, 2H, and 9D. Our approach was adapted from several published works (Kraemer *et al.*, 2014; Maor-Nof *et al.*, 2021). For the analysis, we used ImageJ software to binarize the images, removing cell bodies and leaving only the neurites in black on a white background. Healthy neurites appear continuous, whereas degenerating neurites show a disrupted, fragmented organization due to blebbing and fragmentation. To calculate the DI, we measured the total area of detected neurite fragments and divided it by the total black neurite area. A detailed methodology is now presented in the revised Methods section (lines 897-912) as follows:

“Quantification of Neurite Degeneration

To ensure accurate measurement of neurites, images were captured from blindly selected regions with well-separated axon tracts. Neurite fragmentation was then quantified using an automated image analysis method. The extent of neurite degeneration was expressed as a degeneration index (DI), defined as the ratio of the fragmented neurite area to the total neurite area. To process images for DI calculation, gray intensity in images was first normalized using the auto-level function of GNU Image Manipulation Program (GIMP) software, ensuring consistent background intensity across all images. Subsequently, ImageJ and Ilastik software were employed to binarize the images and remove cell bodies, resulting in a black-and-white rendering of neurites. While intact neurites exhibit continuity, degenerating neurites display disrupted, particulate structures due to blebbing and fragmentation. To quantify the fragmented areas of these degenerating neurites, the Particle Analyzer algorithm in ImageJ was used, with detection parameters set for size (20–10,000 pixels) and circularity (0.2–1.0). The total area of identified neurite fragments was then divided by the overall black neurite area to calculate the DI. Consistent with previous studies, the DI ranged from 0, representing completely intact neurites, to 1.0, indicating complete degeneration into fragmented particles (Kraemer *et al.*, 2014; Maor-Nof *et al.*, 2021).”

6) Gene therapy using shFto appears to lack cellular specificity (especially considering Figure 7C and Figure S9D), raising concerns about its potential effects on other cell types, such as astrocytes and microglia, given that Fto can be knocked down in all of these cells.

This is a compelling question. Although ALS is primarily recognized as a motor neuron disease, the contribution of other cell types to motor neuron degeneration is well-established (Van Harten *et al.*, 2021). In this study, we employed *ChAT-Cre* to delete *Mettl14* specifically in the ChAT^{on} neurons, uncovering several key molecular and behavioral features of ALS pathology, supporting in part a cell-autonomous contribution to the disease. Furthermore, we observed increased gliosis, as indicated by elevated Iba1 level in the ventral horn. Increasing evidence supports that altered glial function plays a critical role in psychiatric and neurological disorders (Van Harten *et al.*, 2021). In our study, we reasoned that the gliosis observed in *ChAT-Cre; Mettl14^{flxed}* mice may reflect a prominent reactive astrogliosis aimed at preserving metabolic and signaling homeostasis during motor neuron degeneration, as our new data suggested that Iba1 activation might occur later than motor neuron degeneration in *ChAT-Cre; Mettl14^{flxed}* mice (new Figures 4A and S4A). **Importantly, this scenario does not exclude the possibility that m⁶A modifications in astrocytes are also crucial.** To definitively assess the role of astrocytic *Mettl14* in motor neuron degeneration, its deletion would be necessary. However, achieving this is particularly challenging due to lacking a Cre driver that can selectively delete *Mettl14* in the motor neuron region. Using

GFAP-Cre might be problematic in interpreting the data as it could represent a systematic glial defect, not a regional gliosis defect.

Regarding the reviewer's concern about whether the beneficial effect of gene therapy using scAAV9-sh*Fto* delivered intrathecally in *SOD1^{G93A}* mice is due to an impact on motor neurons or other cell types, we acknowledge that we cannot definitively answer this question at this time. However, we would like to highlight that in developing ASO and gene therapies for SOD1 patients, **initial clinical trials typically do not target specific cell types**. Instead, therapies are administered broadly, with later studies focusing on whether systemic or cell type-specific targeting offers greater benefit. Our study adopted a general strategy to knock down *Fto* in all cell types in the spinal cord, particularly since *Fto* knockdown in control wild-type mice did not produce any observable phenotypes. We agree that in the future, dissecting the cell-autonomous and non-cell-autonomous contributions of m⁶A will be critical. We have now added a few sentences to our revised manuscript to address this point (lines 569-579):

“Second, we observed a significant enhancement of motor and neuromuscular functions, together with a significant delay in disease onset following scAAV9-sh*Fto* intrathecal injection in our *SOD1^{G93A}* mice. Building on this exciting outcome, we are currently testing some new synthesized FTO inhibitors with stronger activity and better blood-brain barrier penetrance. We knocked down FTO using a ubiquitous promoter as an initial attempt. Future optimization to knock down FTO in a cell type-specific manner might minimize possible adverse effects, although targeting FTO might also represent a relatively safe treatment strategy as *Fto* knockout mice are morphologically normal and only display mild learning defects (Hess *et al*, 2013; Ruud *et al*, 2019). Future experiments to develop a new drug cocktail by combining small molecules together with gene therapy to fortify the m⁶A reservoir of both familial and sporadic ALS patients offer tantalizing treatment prospects.”

7) In Figure 7, within the context of preventing m⁶A depletion by regulating *Fto*, it is conceivable that mediators other than *Mettl14* (e.g., *Mettl16*) may be involved in m⁶A modification. Moreover, it remains uncertain to what extent the levels of m⁶A, which have already been depleted by *Mettl14* knockout, can be restored upon *Fto* knockdown. Did the knockdown of *Fto* indeed restore dysregulated gene expression, such as abnormally upregulated chromatin regulators and ALS pathway-related genes shown in Figure 5A and Figure S7?

We thank the reviewer for this comment. We have addressed this concern through several new experiments: (1) Apart from examining *METTL3* and *METTL14* gene expression (from the Answer ALS dataset), we have also now assessed for the revised manuscript other m⁶A modification writer genes, including *METTL5* and *METTL16*. We observed that *METTL5* and *METTL16* gene expression levels were unchanged in the ALS iPSC~MNs (new Figure S1B, right). (2) Additionally, we have performed all of the new m⁶A level analyses based on polyA-enriched mRNAs (new Figures 1E~1G, 2B~2C, and 9C), suggesting that *METTL3/14*-mediated m⁶A might be a significant axis in ALS pathology. (3) We have further analyzed our human ALS iPSC~MNs upon stressor-mediated degeneration, as well as FTO-treated rescue samples, and observed that the dysregulated chromatin (72.73 %) and histone modification (50 %) gene expression levels are restored (new Figure 9E and S11). (4) Moreover, we have observed a dramatic increase in H3K9me3 in the ventral horn of *SOD1^{G93A}* mice, i.e., to the level of Ctrl mice, following scAAV9-

sh*Fto* treatment (new Figure 11). Taken together, our new analyses support that *METTL3*- and *METTL14*-mediated m⁶A RNA modification might particularly serve as an important facet of ALS disease. These results also support that epigenetic abnormalities caused by hypo-m⁶A in ALS might be a novel feature of ALS pathogenesis.

8) In Figure 7, the authors described in the figure legend, as "The CMAP amplitude is already reduced in SOD1G93A mice at P60 and gradually declines further over time, whereas scAAV9-Fto-shRNA treatment significantly ameliorates neuromuscular function at P160." However, the actual dataset shows only marginal differences between the SOD1G93A group and the SODG93A; AAV-shFto group at P160, not at other time points. This description needs to be corrected.

We thank the reviewer for pointing out this issue. We have corrected this statement in the revised manuscript as follows (lines 415-417):

"CMAP amplitude is already reduced in *SOD1^{G93A}* mice at P60 and gradually declines further over time, whereas scAAV9-sh*Fto* treatment mildly ameliorates neuromuscular function at P160."

Minor comments, including grammatical errors.

1) The alterations in neurite complexity depicted in Figure 1M require quantification.

We appreciate the reviewer's comment. To accurately measure neurite degeneration, we have employed a systematic approach in the revised manuscript by analyzing images of randomly selected regions to assess neurite fragmentation. The level of neurite degeneration was quantified using a degeneration index (DI), which is the ratio of the fragmented neurite area to the total neurite area. We defined the DI as ranging from 0 (completely intact) to 1 (completely fragmented). This new analysis is now presented in new Figures 1B~D, 2E, 2H, and 9D. Our approach was adapted from several published works (Kraemer *et al.*, 2014; Maor-Nof *et al.*, 2021). For our analysis, we used ImageJ software to binarize the images, removing cell bodies and leaving only the neurites in black on a white background. Healthy neurites appear continuous, whereas degenerating neurites show a disrupted, fragmented organization due to blebbing and fragmentation. To calculate the DI, we measured the total area of detected neurite fragments and divided it by the total black neurite area. A detailed methodology is now presented in the revised Methods section (lines 897-912) as follows:

Quantification of Neurite Degeneration

To ensure accurate measurement of neurites, images were captured from blindly selected regions with well-separated axon tracts. Neurite fragmentation was then quantified using an automated image analysis method. The extent of neurite degeneration was expressed as a degeneration index (DI), defined as the ratio of the fragmented neurite area to the total neurite area. To process images for DI calculation, gray intensity in images was first normalized using the auto-level function of GNU Image Manipulation Program (GIMP) software, ensuring consistent background intensity across all images. Subsequently, ImageJ and Ilastik software were employed to binarize the images and remove cell bodies, resulting in a black-and-white rendering of neurites. While intact neurites exhibit continuity, degenerating neurites display disrupted, particulate structures due to blebbing

and fragmentation. To quantify the fragmented areas of these degenerating neurites, the Particle Analyzer algorithm in ImageJ was used, with detection parameters set for size (20–10,000 pixels) and circularity (0.2–1.0). The total area of identified neurite fragments was then divided by the overall black neurite area to calculate the DI. Consistent with previous studies, the DI ranged from 0, representing completely intact neurites, to 1.0, indicating complete degeneration into fragmented particles (Kraemer *et al.*, 2014; Maor-Nof *et al.*, 2021).”

2) Li et al. (PMID: 37365312) found that the basal levels of m6A and the expression of *Mettl3* and *Mettl14* were decreased in motor neurons derived from ALS patients' iPSCs compared to control cells. However, in this study, only the results of cells treated with CPA showed decreased m6A levels in Figure 1. Additionally, Supplementary Figure 1E, only compared CPA-treated and untreated cells, without showing differences in basal levels between control and ALS-derived cells before and after differentiation into motor neurons. It would be beneficial to investigate whether the basal expression levels of *Mettl3* and *Mettl14*, as well as m6A levels, are downregulated upon differentiation into motor neurons in ALS-derived cells, as suggested by Li et al.

We appreciate the reviewer's suggestion. Consistent with several previous studies (Thams *et al.*, 2019; Tung *et al.*, 2019), we did not observe either familial or sporadic ALS iPSC-derived motor neurons undergoing degeneration without the application of the stressor, which reflects that ALS patients do not show obvious motor neuron pathology until adulthood or aging. Consequently, we needed to use CPA, an ER stressor known to cause preferential degeneration in ALS lines (Thams *et al.*, 2019; Tung *et al.*, 2019), to accelerate MN degeneration within one week. In our study, we performed several new experiments and still did not observe any change in mRNA m⁶A levels before CPA treatment between the control and ALS iPSCs, yet consistently uncovered that m⁶A levels were sharply reduced in the ALS~MNs when compared to controls (new Figures 1 and 9). This observation slightly differs from the report of Li et al. (2023), which may be attributable to two possible reasons: (1) we adopted different protocols to differentiate iPSCs, and (2) it is unclear if MNs are already undergoing degeneration without a stressor in the study of Li et al. (2023). Nevertheless, the conclusions from our study and those of Li et al. (2023) are largely consistent, which is that hypo-m⁶A might represent a novel characteristic feature of ALS.

3) In Figure 1N, *Foxp1*+/*ISL1*+ neurons are significantly reduced upon *Mettl3* inhibitor treatment. Are *Foxp1*+/*ISL1*+ neurons also decreased in the spinal cord of *Mettl14* cKO mice, as shown in Figure 2C?

Thank you for this detailed observation. We used FOXP1, a marker for limb-innervating motor neurons present in the brachial and lumbar lateral motor columns (LMC), to assess human iPSC-derived motor neurons because the differentiation protocol we employed is known to produce a significant number of FOXP1^{on} neurons (Maury *et al.*, 2015; Tung *et al.*, 2019). We did not assay *Foxp1* in the *Chat-Cre; Mettl14^{loxed}* mice for two reasons: (1) we observed a reduction in *ChAT*^{on} motor neurons across all spinal cord segments; and (2) FOXP1 is widely distributed in spinal cord interneurons after the postnatal stage (Dasen *et al.*, 2008), which could lead to ambiguity in our results. To avoid confusion, we have revised the manuscript to use SMI32 to assess neurodegeneration, which better represents the neurodegenerative process, and have re-quantified the phenotype in updated Figures 1B~1D, 2E, 2H, and 9D.

4) In Figure 4, panels C and D, the data depict both glial cells and neuronal cells. However, it is notable that other abundant cell populations, such as microglia and endothelial cells, are absent. It is necessary to clarify whether these cell types were intentionally excluded from the experimental steps or if their absence is due to limitations in the experimental design.

We thank the reviewer for the helpful reminder. In our study, we used ChAT-Cre mice to label cholinergic cell types and employed fluorescence-activated nuclei sorting (FANS) to enrich cholinergic populations for our single-nuclei multiome experiment. This approach was intended to

collect primarily cholinergic cells, including skeletal and visceral motor neurons, as well as cholinergic interneurons, in order to focus on their changes following *Mettl14* knockout. The single nuclei shown in original Figures 4C and 4D (now new Figures 7C and 7D) represent all nuclei collected after excluding low-quality nuclei and potential doublets, based on gene expression, transcript counts, and ATAC peak properties. As described in our Methods, sample quality was assessed using the following criteria: gene expression data — $nFeature_RNA < 10,000$, $nFeature_RNA > 200$, and $nCount_RNA > 500$; chromatin accessibility data — $pct_reads_in_peaks > 15$,

$blacklist_fraction < 5$, $nucleosome_signal < 4$, and $TSS.enrichment > 1$. Doublets were estimated using DoubletFinder and excluded from the analysis. After clustering, we annotated the clusters based on marker gene expression (new Figure S7) and cross-referenced our data against published single-nuclei spinal cord data (Blum et al., 2021). Furthermore, we would not expect to observe microglia and endothelial marker-expressing cell clusters in our collected sample after passing quality filtering (Figure R) due to our experimental design to enrich for cholinergic neuronal cell types.

We have now clarified the legend for Figure 7C as follows: "Uniform manifold approximation and projection (UMAP) representation of all nuclei that passed quality filtering. Dimensionality reduction and clustering were performed based on gene expression (RNA, left), chromatin accessibility (ATAC, middle), and weighted nearest neighbor (WNN) integration of RNA and ATAC data (right). Clusters are color-coded and annotated using label transfer prediction, referencing Blum et al., 2021."

5) In Figure 5, panels B and D, the ChAT staining appears markedly different from Figure 2, panels C and E. In Figure 5, the ChAT staining appears as dot-shaped structures resembling a "salt and pepper" pattern, contrasting with the more continuous staining observed in Figure 2. An explanation for this discrepancy is needed.

We thank the reviewer for raising this issue, which has also been noted by Reviewer #2. The primary reason for this difference is the developmental stage at which the data were collected. While spinal motor neurons are the main source of ChAT^{on} cells in the spinal cord, it is well-known

that C-boutons, large cholinergic terminals that synapse on the soma and proximal dendrites of motor neurons, play a critical role in modulating motor neuron excitability (Wells *et al*, 2021). These C-boutons originate from a small subset of V0 (Dbx1^{on}) cholinergic interneurons (V0_C), which express the paired domain transcription factor Pitx2.

In the postnatal stage, C-boutons are less abundant compared to the adult stage, which can result in noticeable differences in the images, especially when viewed at different scales. Reviewer #2 also astutely observed reduced C-boutons in the *Chat-Cre; Mettl14^{flxed}* mice. To investigate this further, we conducted a series of new immunostainings across different developmental stages in these mice and corroborated this observation, confirming that C-boutons are reduced prior to motor neuron degeneration (new Figure 3E). We have now added this result (lines 192-201) and discussed this new finding (lines 487-496) in the revised manuscript:

“Apart from their kyphotic appearance and movement defects, we further investigated a series of molecular and cellular ALS disease features in the *Chat-Cre; Mettl14^{flxed}* mice. At the molecular level, we observed (1) that the numbers of ChAT^{on} MNs in the lumbar region of spinal cords were comparable before P70 but gradually declined after P100 (Figure 3C and 3D). However, C-boutons, a source of cholinergic input to MNs, already showed a prominent decrease from P70 (Figure 3E and 3F); (2) prominent neuroinflammation upon microglia (Iba1^{on}) activation in the spinal cords of the *Chat-Cre; Mettl14^{flxed}* mice relative to controls at P160 (Figure 4A and 4B), but not before P120 (Figures S4A and S4B); and (3) significant cytoplasmic aggregation of Tdp43 in the *Chat-Cre; Mettl14^{flxed}* mice, whereas control littermates mainly presented nuclear localizations for that protein after P120 (Figures 4C, 4D, S4C, and S4D).”

“Although *Chat-Cre; Mettl14^{flxed}* mice exhibited several ALS-like phenotypes, we were unable to determine whether these effects were specifically attributable to C-boutons synapsing on the soma and proximal dendrites of MNs, the MNs themselves, or both. This is an intriguing question, as recent reports have suggested that spinal inhibitory neurons, including C-boutons, might degenerate before MNs in a mouse model of ALS (Montanana-Rosell *et al*, 2024; Wells *et al.*, 2021). Currently, the lack of a Cre driver that specifically targets adult MNs limits examining this question. However, our single-nuclei ATAC-seq data offer a promising avenue for identifying enhancers specific to adult cholinergic interneurons (INs) and MNs. Generating Cre lines driven by these enhancers could allow for a more precise dissection of the respective contributions of INs and MNs to the observed phenotypes.”

6) Figure 7F is not well described in the text and the figure legend. Is this staining performed using an m6A antibody? What is the author's interpretation of these data?

We thank the reviewer for highlighting this issue. We have revised the relevant sections in the main text and figure legends for clarity (lines 405-408):

“In the *SOD1^{G93A}* ALS mouse model, we observed a reduction in m⁶A methylation by m⁶A immunostaining (Figure 10F). Moreover, we also found that m⁶A levels can be upregulated via intrathecal delivery of scAAV9-sh*Fto* in the *SOD1^{G93A}* mouse model (Figure 10F).”

7) In Figure 4, Panel B, the meanings of R1, R2, and R3 should be described.

We have now defined these in the figure legends (New Figure 6F)

“Replicates 1, 2, and 3 (R1, R2, and R3) represent the triplicate biological repeats.”

8) In Figure 5, panel D, it appears that some gamma-H2AX signals are not overlapped with DAPI staining. These regions may either delineate only the DAPI portion on the gamma-H2AX images with boundary or display images with DAPI completely overlapped. It would be necessary to provide a figure demonstrating DAPI and co-localization with certainty. (Refer to PMID: 32737294; PMID: 32615088)

We apologize for the confusion. We now provide a higher-resolution image in the revised manuscript (new Figure 8), as well as zooming in on key areas with the DAPI image to enhance the clarity of our results (new Figure S10).

9) In Figure 7F-I, the time points of the experiments performed were not described.

We have now added this information to the figure legend (now new Figure 10).

Reviewer #2

Chen and colleagues investigated the roles of RNA N6-methyladenosine (m6A) modification by methyltransferase complex METTL3/METTL4 in the pathogenesis of motor neuron degeneration in amyotrophic lateral sclerosis (ALS). The authors first showed that iPSC-derived motor neurons (MNs) with SOD1(L144F), C9ORF72(exp), or TDP-43(G293S) mutation exhibited significant reductions in m6A RNA modification after these cells were treated with ER stress inducer CPA (cyclopiazonic acid), which significantly reduced the relative abundance of m6A RNA and the number of iPSC-derived MNs. Knocking down METTL4 in SH-SY5Y cells using shRNA reduced m6A RNA and led to similar effects. To further determine the role of m6A in motor neurons in vivo, the authors generated conditional mutant mice where the *Mettl4* gene was deleted using Olig2-Cre or ChAT-Cre. While the Olig2-Cre;*Mettl4* CKO exhibited early postnatal lethality, the cause was not due to loss of MNs but a significant reduction in the number of oligodendroglia. In contrast, ChAT-Cre;*Mettl4* CKO mice exhibited adult-onset phenotypes resembling those seen in other ALS mouse models, including motor deficits, progressive loss of spinal MNs, loss of innervation at the neuromuscular junctions (NMJs), neuroinflammation, and loss of nuclear TDP-43 and accumulation of TDP-43 in the cytoplasm of spinal MNs. Using multiomic analyses, the authors revealed that the mechanism that promoted neuronal degeneration in ChAT-Cre;*Mettl4* CKO mice was due to the down-regulation of genes related to axonogenesis and synapse organization and up-regulation of genes related to chromatin remodeling and histone modification. Many mis-regulated genes have been implicated in the pathogenesis of ALS. In support of these results, inhibiting demethylase FTO using FB23-2 in ALS iPSC-derived MNs improved survival after CPA treatment, whereas injecting SOD1(G93A) mice with AAV9-shFto delayed disease-onset and prolonged survival in these mice.

Overall, this is an interesting study that provides compelling evidence supporting the essential role of m⁶A RNA modification carried out by methyltransferase complex member METTL4 in the of survival in spinal motor neurons. This study is also timely as its data will help clarify some of the conflicting data in the recent literature (e.g. McMillen et al vs Li et al, 2023). By combining results from iPSC-derived MNs and conditional mutant mice, this study supports that METTL4-mediated m⁶A RNA modification is essential for neuronal homeostasis and survival in postnatal life. The mislocalization of TDP-43 protein in the spinal motor neurons in ChAT-Cre;Mettl4 CKO (Figure 2G) is quite intriguing and **thought-provoking**. While there are many strengths in this study, there are a few areas where data presentation, organization, and clarity can be improved. Here are the specific comments and suggestions:

We very much appreciate this reviewer's positive support and encouragement. Below, we address all of his/her comments in detail.

1. Figure 1: The data in Figure 1B-D and 1E-G are redundant and can be improved. For instance, the results in Figure 1B-D should be converted into line graphs and provide statistics to support the progressive decline of the relative abundance of m⁶A in iPSC-derived MNs, which contributes to neuronal loss. Once the panels in 1B-D are updated, the panels in 1E-G can be removed. Additionally, the authors should provide more details regarding how the quantification in Figure 1N was performed.

We thank the reviewer for these suggestions. Originally, we hoped to emphasize that m⁶A levels were already downregulated before MNs undergo degeneration upon stressor treatment, so we had specifically added a figure panel relating to this time point. We have now removed this panel and reshaped our statistical analysis (new Figures 1B~1E).

Regarding our original quantification of 1N neurite degeneration, to accurately measure neurite degeneration, in this revision, we have employed a systematic approach in the revised manuscript by analyzing images of randomly selected regions to assess neurite fragmentation. The level of neurite degeneration was quantified using a degeneration index (DI), which is the ratio of the fragmented neurite area to the total neurite area. We defined the DI as ranging from 0 (completely intact) to 1 (completely fragmented). This new analysis is now presented in new Figures 1B~D, 2E, 2H, and 9D. Our approach was adapted from several published works (Kraemer *et al.*, 2014; Maor-Nof *et al.*, 2021). For our analysis, we used ImageJ software to binarize the images, removing cell bodies and leaving only the neurites in black on a white background. Healthy neurites appear continuous, whereas degenerating neurites show a disrupted, fragmented organization due to blebbing and fragmentation. To calculate the DI, we measured the total area of detected neurite fragments and divided it by the total black neurite area. A detailed methodology is now presented in the revised Methods section (lines 897-912) as follows:

“Quantification of Neurite Degeneration

To ensure accurate measurement of neurites, images were captured from blindly selected regions with well-separated axon tracts. Neurite fragmentation was then quantified using an automated image analysis method. The extent of neurite degeneration was expressed as a degeneration index

(DI), defined as the ratio of the fragmented neurite area to the total neurite area. To process images for DI calculation, gray intensity in images was first normalized using the auto-level function of GNU Image Manipulation Program (GIMP) software, ensuring consistent background intensity across all images. Subsequently, ImageJ and Ilastik software were employed to binarize the images and remove cell bodies, resulting in a black-and-white rendering of neurites. While intact neurites exhibit continuity, degenerating neurites display disrupted, particulate structures due to blebbing and fragmentation. To quantify the fragmented areas of these degenerating neurites, the Particle Analyzer algorithm in ImageJ was used, with detection parameters set for size (20–10,000 pixels) and circularity (0.2–1.0). The total area of identified neurite fragments was then divided by the overall black neurite area to calculate the DI. Consistent with previous studies, the DI ranged from 0, representing completely intact neurites, to 1.0, indicating complete degeneration into fragmented particles (Kraemer *et al.*, 2014; Maor-Nof *et al.*, 2021).”

2. Figure 2: This figure can be further strengthened in the following areas: (1) Panel 2C: In addition to neuronal cell body, ChAT also labels cholinergic synapses. Based on the images provided, it seems that there is already a significant reduction in ChAT+ synapses at P70 and P120. This reduction is likely to contribute to the motor deficits in these young ages (Figure 3). This is an important phenotype that the authors should use other synaptic markers to quantify and support. These new data can further enhance the single-cell multiomic data (Figures 4-5). (2) Panel 2E-F: It will be important to demonstrate whether microgliosis occurs before or after neuronal (and synaptic) degeneration. (3) Panel 2G-H: The authors should provide more in-depth analyses on the very interesting TDP-43 mislocalization phenotypes, including when this phenotype can be detected, does cytoplasmic TDP-43 colocalize with other RNA binding proteins (e.g. FUS) or RNA granule markers. (4) The survival curves in Panel 2B showed the median survival is about 250 days. However, most of the histopathology was performed at P160. Why? Is there no further deterioration of pathology after P160? (5) The quantifications should be normalized by surface area or another parameter to keep the quantifications consistent and comparable between different conditions.

We thank this reviewer for insightful comments that allowed us to strengthen our analyses of the *ChAT-Cre; Mettl14^{flxed}* mice. We have performed most of the reviewer’s suggested experiments by adding several time points from the postnatal to adult stages (P30~P250). Each suggested experiment is addressed as follows (new Figures 3, 4, and S4):

(1) It has been documented previously that activation of C-boutons—large cholinergic modulatory synapses on motor neurons—changes as ALS progresses (Montanana-Rosell *et al.*, 2024; Wells *et al.*, 2021). In the spinal cords of *ChAT-Cre; Mettl14^{flxed}* mice, we observed that though the number of ChAT^{on} motor neurons in the ventral horn gradually decreased from P100–120, a significant reduction in C-boutons on motor neurons was already evident by P70 (new Figures 3C~3F). We now discuss this new finding in detail in the Discussion section of the revised manuscript (lines 487-496):

“Although *ChAT-Cre; Mettl14^{flxed}* mice exhibited several ALS-like phenotypes, we were unable to determine whether these effects were specifically attributable to C-boutons synapsing on the soma and proximal dendrites of MNs, the MNs themselves, or both. This is an intriguing question, as recent reports have suggested that spinal inhibitory neurons, including C-boutons, might degenerate before MNs in a mouse model of ALS (Montanana-

Rosell *et al.*, 2024; Wells *et al.*, 2021). Currently, the lack of a Cre driver that specifically targets adult MNs limits examining this question. However, our single-nuclei ATAC-seq data offer a promising avenue for identifying enhancers specific to adult cholinergic interneurons (INs) and MNs. Generating Cre lines driven by these enhancers could allow for a more precise dissection of the respective contributions of INs and MNs to the observed phenotypes.”

- (2) Increasing evidence suggests that altered glial function plays a critical role in psychiatric and neurological disorders (Van Harten *et al.*, 2021). In our study, we reasoned that the gliosis observed in *ChAT-Cre; Mettl14^{flxed}* mice may reflect a prominent reactive astrogliosis aimed at preserving metabolic and signaling homeostasis during motor neuron degeneration, as our new data suggested that Iba1 activation might occur later than motor neuron degeneration in *ChAT-Cre; Mettl14^{flxed}* mice (new Figures 4A and S4A).
- (3) To further examine Tdp43 expression, we scrutinized its expression profile in the spinal cords of control and *ChAT-Cre; Mettl14^{flxed}* mice between P10 to P160 (new Figure 4C, 4D, S4C, and S4D). While Tdp43 was exclusively localized to the nucleus in motor neurons from P10 to P100 in both control and knockout mice, cytoplasmic aggregation of Tdp43 began to emerge at P120, but only in the knockout mice. Strikingly, we also observed significant cytoplasmic aggregation of Fus, another prominent RNA-binding protein (RBP) known to form cytoplasmic aggregates in some ALS patients (Figure S4E and 4F). These results indicate that Tdp43 and Fus cytoplasmic aggregation are prominent features in the *ChAT-Cre; Mettl14^{flxed}* mice. We now elaborate on the potential reasons for Tdp43 and Fus aggregation in the revised Discussion (lines 498-514):

“As a hallmark of ALS, why do Tardbp (Tdp43) and Fus move out of the nucleus to form cytoplasmic aggregations in the *ChAT-Cre; Mettl14^{flxed}* mice? A previous study has revealed that a lack of specific RNA modifications may affect global and/or local translation rates, consequently increasing protein aggregation (Nedialkova & Leidel, 2015). Thus, it has been proposed that RNA modifications serve as conduits of information linking a cell's metabolic condition with its translational productivity (Helm & Alfonzo, 2014). Consequently, any disruption in regulating RNA modifications could potentially perturb the equilibrium between metabolic processes and protein synthesis. Further work is needed to disentangle the causal relationship between dysregulation of RNA modifications and Tdp43 translocation. Additionally, the nuclear m⁶A reader YTHDC1 has been shown to exert an important role in modulating many biological processes and contributing to disease, especially cancers. YTHDC1 might be the main mediator of a series of m⁶A readers, thereby controlling their activity in neuronal functions (Lence *et al.*, 2016). Moreover, *YTHDC1* RNAs have been discovered as binding to TDP43 protein in human SH-SY5Y neuroblastoma cells (Tam *et al.*, 2019), and peripheral blood sample transcriptional profiling of a huge heterogeneous ALS cohort (not only sporadic cases) revealed YTHDC1 to be differentially expressed (La Cognata *et al.*, 2020). These results point to YTHDC1-mediated m⁶A nuclear events and nucleocytoplasmic trafficking as contributing to ALS. Further experiments are warranted to dissect this potential disease mechanism in detail. “

- (4) Initially, we focused our analysis on phenotypes observed at P160, an early stage of disease onset. In our revised manuscript, we have conducted extensive experiments across presymptomatic (P30–P70), early symptomatic (P70–P100), disease onset (P100–P120), early onset (P160), and late-onset (P200–P250) stages. We believe that through these new

experiments, we provide one of the most thoroughly characterized animal models available for future ALS research.

- (5) We apologize for any confusion regarding the specific figures that the reviewer suggested for quantification with internal common controls. In response, we have conducted several new experiments to thoroughly examine the phenotypes from P30 to P250. Our findings indicate that several pathological features do not manifest between P30 and P70. We believe this provides a more accurate reflection of an adult phenotype, rather than suggesting a global defect from an early stage where all aspects are disrupted. We hope that these additional efforts adequately address the reviewer's concerns regarding our phenotype analyses of the *ChAT-Cre; Mettl14^{flxed}* mice.

3. Figure 4: Panels A-B described the use of the Nanopore platform to characterize m⁶A RNA modifications. However, the authors chose to do this in mESC-derived MNs, not in the spinal cord or sorted MNs from *ChAT-Cre;Mettl4* CKO mice. It is quite odd because the data from mESC-derived MNs (Figure 4B and Fig. S4) had no direct relevance to the phenotypes in *ChAT-Cre;Mettl4* CKO mice. If it is not possible to perform Nanopore analysis using spinal cord or sorted MNs from *ChAT-Cre;Mettl4* CKO mice, the authors should perform targeted analysis of m⁶A on ALS risk genes in using spinal cord or sorted MNs from *ChAT-Cre;Mettl4* CKO mice. These data could become a full-fledged figure by themselves, separated from the single-cell multiomic data.

This is an important issue, as Reviewer #1 raised the same concern. In our manuscript, we employed *Sun1^{sfGFP}; ChAT-Cre; Mettl14^{flxed}* mice to isolate single nuclei from the lumbar spinal cords for multiome analysis. For this approach, we harvested nuclei from the lumbar region of the spinal cord and performed FANS to select GFP^{on} cells. In fact, we originally made several attempts to collect GFP^{on} nuclei for Nanopore direct RNA-seq to identify m⁶A sites in the spinal motor neurons (MNs). However, it proved a daunting task to obtain sufficient mRNA for this technique, considering that only < 2 % of cells in the adult spinal cord are GFP^{on}, and the harvest yield was extremely low after FANS enrichment. As an alternative approach, we used mouse ESC~MNs cultured in a conditioned medium that can drive these ESC~MNs to acquire an adult-like MN identity (Patel *et al.*, 2022; Tung *et al.*, 2019). Specifically, under our experimental conditions, we cultured the 3D embryoid bodies for five days, then attached the cultured embryoid bodies to plates coated with a supplement enriched in neurotrophic factors, and cultured them for an additional 5 days (Day 12). We describe this protocol in detail in our Methods section. In the revised manuscript, we have now further verified that several genes characteristic of adult MNs (*Slc5a7, Syn1, Fos, Col5a3, Dmp1, Bag3*, etc.; (Patel *et al.*, 2022) are robustly expressed in our mature ESC~MNs (new Figures 6A~6C). Consistently, our Nanopore direct RNA-seq generated similar results (new Figure 6E). Collectively, these new characterizations indicate that the mature ESC~MNs cultured in conditioned mature media largely reflect many adult MN characteristics used for single-nuclei multiome analysis (new Figure 7).

As the reviewer indicates, it might be more clinically relevant to use human iPSC-derived MNs. However, we reasoned that the m⁶A epitranscriptome might not be identical across species, adding an extra layer of complexity to our analysis. In our revised manuscript, we have added a few paragraphs to clarify these issues and limitations (lines 229-267);

“

To gain insights into the potential mechanisms underlying how hypo-m⁶A promotes MN degeneration, we aimed to systematically identify the dysregulated genes possessing m⁶A modifications in our *ChAT-Cre; Mettl14^{flxed}* mice (Figures 6 and 7). To identify m⁶A-modified transcripts, we adopted a Nanopore direct RNA sequencing platform, which enables the identification of the MN m⁶A epitranscriptome at single-nucleotide resolution (Figure 6). Since it is technically challenging to obtain sufficient adult MNs from the spinal cord for Nanopore direct RNA sequencing, we employed an enhanced method using mouse ESC-derived MNs, matured with a conditioned medium (Figure 6A)(Patel *et al.*, 2022; Tung *et al.*, 2019). First, we confirmed that this approach successfully generated MNs expressing mature neuronal markers, with longer and more mature neurite structures revealed by discrete Syn1-positive puncta (new Figures 6B and 6C). We then subjected these mature MNs to the ONT Nanopore platform, which provides a powerful framework for detecting RNA modifications through advanced machine-learning algorithms applied to sequencing metrics (Figure 6D). As expected, several adult mature MN genes are abundantly expressed in the Nanopore direct RNA-seq results (Figure 6E). We employed two supervised machine learning tools, namely EpiNano and m6Anet (Hendra *et al.*, 2022; Liu *et al.*, 2021), setting a stringent criterion that the predicted m⁶A sites need to occur in at least two samples for either one of the algorithms (see Methods for details). We identified 30,340 high-confidence m⁶A modification sites corresponding to 7,921 genes (Figure 6D, genes of interest are illustrated in Table S1; refer to the Methods section for details). Consistent with previous findings (Castro-Hernandez *et al.*, 2023), our analysis revealed enriched distributions of m⁶A sites in coding sequences (CDSs) and 3' UTRs, especially near the stop codons (Figure 6F). By analyzing the enrichment of Gene Ontology (GO) and KEGG pathways for the m⁶A-modified transcripts, we noticed a striking enrichment for ALS-related genes (Figure 6G). In addition, among the 81 identified ALS risk genes to date, our direct RNA sequencing demonstrated that 34 of them are m⁶A-modified ($p = 2.06 \times 10^{-6}$, one-tailed hypergeometric test; Figure 6H and Table S2).

To validate our computationally predicted m⁶A sites, we used *Tardbp* (*Tdp43*) as a benchmark to validate our methodology. Consistent with a previous report (McMillan *et al.*, 2023), we confirmed the existence of a previously identified m⁶A site in *Tardbp* with a high probability rate with stoichiometry ranging from 57.38 to 92.53 % (Table S3) and verified via m⁶A antibody pull-down assay (Figure 6I, site 1). Additionally, we uncovered and validated an additional high m⁶A-modified site within the *Tardbp* transcript from Nanopore direct RNA-seq (Figure 6I, site 2). Moreover, we substantiated the existence of predicted m⁶A sites in *Atp13a2* (*Park9*) (Figure 6J) (Spataro *et al.*, 2019), an ALS risk gene not previously shown to have m⁶A modifications in MNs. Finally, we further verified several newly identified m⁶A-modified sites in ALS risk genes, including *Dctn1*, *Epha4*, *C9orf72*, *Glt8d1*, *Cacna1h*, *Chrna3*, *Bscl2*, *Fig4*, *Hnrnpa2b1*, *Ubqln2*, *Hnrnpa1*, *Tuba4a*, *Sod1*, *Chmp2b*, and *PIKfyve* (Figure S6B). Thus, these observations confirm the sensitivity, accuracy, and reliability of Nanopore technology to identify m⁶A-modified sites in MNs.”

4. Figure 5: The ChAT confocal images in Panels B and D look very different from those in Figure 2E. This discrepancy should be addressed since the corresponding quantification is striking and critical for the validation of the RNA-seq dataset. In both Figures 4 and 5, the reviewer recommends adding the number of genes in DEGs and m⁶A site in the Venn diagrams to indicate how many genes are in each group.

We thank the reviewer for raising this issue, which was also noted by Reviewer #1. The primary reason for this difference is the developmental stage at which the data were collected. While spinal motor neurons are the main source of ChAT^{on} cells in the spinal cord, it is well-known that C-boutons, large cholinergic terminals that synapse on the soma and proximal dendrites of motor neurons, play a critical role in modulating motor neuron excitability (Wells *et al.*, 2021). These C-boutons originate from a small subset of V0 (Dbx1^{on}) cholinergic interneurons (V0_C), which express the paired domain transcription factor Pitx2.

In the postnatal stage, C-boutons are less abundant compared to the adult stage, which can result in noticeable differences in the images, especially when viewed at different scales. As Reviewer #2 astutely points out, there is a reduction in C-boutons in the *ChAT-Cre; Mettl14^{flxed}* mice. To investigate this further, we conducted a series of new immunostainings across different developmental stages in these mice and corroborated this observation, confirming that C-boutons are reduced prior to motor neuron degeneration (new Figure 3E). We have now added this result (lines 192-201) and discuss this new finding in the Discussion section (lines 487-496):

“Apart from their kyphotic appearance and movement defects, we further investigated a series of molecular and cellular ALS disease features in the *ChAT-Cre; Mettl14^{flxed}* mice. At the molecular level, we observed: (1) that the numbers of ChAT^{on} MNs in the lumbar region of spinal cords were comparable before P70, but gradually declined after P100 (Figure 3C and 3D). However, the C-boutons, a source of cholinergic input to MNs, already showed a prominent decrease from P70 (Figure 3E and 3F); (2) prominent neuroinflammation upon microglia (Iba1^{on}) activation in the spinal cords of the *ChAT-Cre; Mettl14^{flxed}* mice relative to controls at P160 (Figure 4A and 4B), but not before P120 (Figures S4A and S4B); and (3) significant cytoplasmic aggregation of Tdp43 in the *ChAT-Cre; Mettl14^{flxed}* mice, whereas control littermates mainly presented nuclear localizations for that protein after P120 (Figures 4C and 4D; S4C and S4D).”

“Although *ChAT-Cre; Mettl14^{flxed}* mice exhibited several ALS-like phenotypes, we were unable to determine whether these effects were specifically attributable to C-boutons synapsing on the soma and proximal dendrites of MNs, the MNs themselves, or both. This is an intriguing question, as recent reports have suggested that spinal inhibitory neurons, including C-boutons, might degenerate before MNs in a mouse model of ALS (Montanana-Rosell *et al.*, 2024; Wells *et al.*, 2021). Currently, the lack of a Cre driver that specifically targets adult MNs limits examining this question. However, our single-nuclei ATAC-seq data offer a promising avenue for identifying enhancers specific to adult cholinergic interneurons (INs) and MNs. Generating Cre lines driven by these enhancers could allow for a more precise dissection of the respective contributions of INs and MNs to the observed phenotypes.”

With regard to Figures 4 and 5 (now new Figures 6, 7, and 8), we have added the gene numbers on the Venn diagrams, as suggested.

5. Figure 6: Panel 6E presents RNA-seq data from control iPSC-derived MNs or MNs treated with ER stressor CPA or FTO inhibitor. The data presented here are highly selected and lack statistical analyses to support the conclusion. The authors should adopt more unbiased approaches to show

that FTO inhibitor FB23-2 can indeed have broader impacts in correcting the transcriptomes of these iPSC-derived MNs.

We appreciate this suggestion. We apologize for not including this analysis in our original submission, which was excluded due to the already extensive presentation of data. We now include these results (new Figure S11) and respective explanations (lines 362-387) as follows:

“The consistent manifestation of hypo-m⁶A in human ALS iPSC~MNs, together with our *Chat-Cre; Mettl14^{flxed}* mice recapitulating ALS pathology, prompted us to explore if bolstering the m⁶A reservoir could represent a therapeutic strategy. Thus, we deployed several fALS patient (*C9ORF72^{exp-800 G4C2}*, *SOD1^{+/L144F}*, *TDP43^{G298S}*) iPSC~MN lines and one sALS iPSC~MN line to reflect MN degeneration (Figure 9A~9D). Then, we treated these lines with FB23-2, an inhibitor of FTO (an m⁶A eraser) (Huang *et al*, 2019), to see if this approach could be applied to rescue MN degeneration (Figure 9A). First, we differentiated the ALS iPSCs under defined conditions to cause MN degeneration through a selective ER stressor, CPA (Figures 9A and 9B). Subsequently, we applied FB23-2 to determine if doing so could elevate m⁶A levels and thereby restore the m⁶A repertoire to rescue MN degeneration in different contexts of ALS. As expected, we observed a consistently significant increase in m⁶A levels upon applying FB23-2, albeit to varying degrees (Figure 9C). By using SMI32 to assess MN degeneration and neurite complexity, we observed that FB23-2 promotes MN survival upon CPA stressor treatment for both familial and sporadic ALS MNs (Figure 9D). Thus, our results indicate that fortifying basal m⁶A levels by adding a m⁶A eraser inhibitor can rescue human ALS iPSC-derived MNs from degeneration.”

To determine if the neuroprotective effects of the FTO inhibitor on MN degeneration in ALS are mediated through its regulation of m⁶A-modified ALS risk genes (Figures 7G and 7H), we performed RNA-seq analysis on ALS iPSC-derived MNs treated with FB23-2. Differential expression analysis revealed that following FB23-2 treatment, several m⁶A-modified genes involved in synaptic function, RNA metabolism, and chromatin and histone modifications were restored to levels similar to controls (Figure S11). Notably, the expression of multiple m⁶A-modified ALS risk genes was returned to control-like levels (vehicle-treated), suggesting that FTO inhibition may mitigate MN degeneration in ALS by modulating m⁶A-modified gene expression (Figure 9E). These findings indicate that enhancing m⁶A levels in ALS iPSCs using small molecules could help restore the balance of the m⁶A epitranscriptome and preserve MN integrity.”

6. Figure 7: The authors use several fALS patient iPSC~MN lines and one sALS iPSC~MN to show that restoring m⁶A homeostasis can rescue MN degeneration in both models. While the phenotype after treating with FB23-2, an inhibitor of FTO, is striking, this reviewer recommends including another control of adding FB23-2 to the vehicle to establish the role of the FTO inhibitor during homeostatic state. This control would also help delineate the significance of m⁶A homeostasis in control MNs without ALS-related mutations.

We thank the reviewer for this suggestion. We have now added the vehicle+FTO inhibitor data as new revised Figure 9, together with quantification in Figures 9C and 9D.

7. Finally, there are several areas in the manuscript where statements or conclusions went beyond the data. For instance, one page 7, last sentence of the 2nd paragraph the authors mentioned their data in ChAT-Cre;Mettl4 CKO “mirroring numerous prominent molecular pathology hallmarks displayed by human ALS patients.” This is an overreaching statement because ALS is a very heterogeneous disease (sporadic and familial ALS alike). It is not clear whether the field agrees there are “molecular pathology hallmarks” for ALS.

We agree with the reviewer’s critique and have now revised the sentences (lines 204-207) as follows:

“Overall, our findings indicate that the *ChAT-Cre; Mettl14^{flxed}* mouse model demonstrates progressive MN degeneration, mirroring several key molecular pathological features observed in human ALS patients.”

Reviewer #3

I co-reviewed this manuscript with one of the reviewers who provided the listed reports. This is part of the Nature Communications initiative to facilitate training in peer review and to provide appropriate recognition for Early Career Researchers who co-review manuscripts. This email has been sent through the Springer Nature Tracking System NY-610A-NPG&MTS

Reviewer #4

Very interesting manuscript on potential roles of m6A in motor neurons in ALS. However the assays to measure m6A (on which the main conclusions were made) were flawed and not actually measuring m6A in mRNA (which is generated by METTL14). Instead total RNA was used, which measures m6A from other enzyme pathways unrelated to METTL14. Thus, some of the key conclusions are invalid. However the gene therapy/drug therapy with FTO is interesting. Comments below.

We very much appreciate this reviewer’s critical comments. Below, we address all of his/her critiques in detail.

1. assays for m6A are not correct - the authors use total RNA for both m6A quantification and dot blots. Most of the m6A is m6A from rRNA not mRNA. The results of figure 1 are not reflective of reduced Mettl3 levels since they are looking at m6A not derived from this enzyme.

We appreciate the opportunity to clarify our methodology and findings. Below, are our responses to the points raised and details of new experiments to address them:

1. **Use of mRNA in Dot Blots/ELISA:** In fact, we used purified mRNA to perform all dot blot assays in the originally submitted results. However, it is very challenging to acquire a

sufficient amount of mRNA from motor neurons to perform all m⁶A ELISA assays. This is probably the reason why, in our original results, the m⁶A dot blot showed quite a dramatic reduction in ALS, whereas the METTL3 inhibitor-treated cells showed a mild, albeit significant, reduction in the m⁶A ELISA assay. To address this issue, we have performed the m⁶A ELISA again on ALS iPSC~MNs using total RNA depleted of rRNA, followed by enrichment for poly-A mRNA (new Figures 1E, F, and G, detailed in Methods). Additionally, we have also used the same m⁶A mRNA ELISA and dot blot assays to perform new analyses on STM2457-treated and *METTL3/14* knockdown iPSC~MNs (new Figure 2F~2H and S2). All of the new m⁶A mRNA ELISA and dot blot assays support the manifestation of significant and consistent hypo-m⁶A in ALS MN contexts.

2. **METTL5 and METTL16 are unaffected in ALS patients:** Our investigation of the Answer ALS dataset revealed that other m⁶A modification enzymes, such as METTL16 for ncRNAs or METTL5 for rRNAs remain unchanged in ALS iPSC~MNs, as shown in new Figure S1B right. This outcome further supports that hypo-m⁶A is a feature of ALS and might be primarily reflected in the mRNA transcriptome.

We hope these new analyses address the reviewer's concern.

STM2457 exerts its own toxicity independent of METTL3 so data should be validated with other inhibitors (such as STC15). Concentration of STM2457 not indicated in legend or figure 1.

We thank the reviewer for raising this important concern. Although we appreciate the suggestion to use an alternative m⁶A inhibitor (STC15) to replicate the effects of STM2457, we are mindful that issues such as specificity and toxicity associated with chemical inhibitors may remain problematic, both in this study and for future readers. To address this concern, we chose to perform knockdown experiments targeting either *METTL3* or *METTL14* in mature iPSC-derived MNs to assess the phenotype, replacing the original SH-SY5Y cell line results (new Figures 2F~2H). In these new experiments, we consistently observed significant neurite degeneration in both *METTL3* and *METTL14* knockdown iPSC-MNs, but not in neurons treated with control shRNA. These phenotypic changes closely mirrored the effects seen in STM2457-treated MNs (concentration: 20 μM, details are described in the Methods; new Figures 2A~2E), reinforcing our conclusion that METTL3/14 is critical for maintaining motor neuron survival. This alternative approach not only strengthens our findings but also reduces concerns about the potential off-target effects or toxicity of chemical inhibitors. We revise the main text as follows (lines 162-168):

“To confirm if hypo-m⁶A leads to MN degeneration, we adopted two approaches. First, we used a specific METTL3 inhibitor (METTL3i), STM2457 (Yankova *et al*, 2021), to impair m⁶A production during human MN differentiation (Figure 2A), which revealed that METTL3i reduces the m⁶A-mRNA repertoire assayed by m⁶A ELISA and dot blot (Figure 2B and 2C), with concomitantly drastic neurite degeneration and a reduced MN population (Figures 2D and 2E). Secondly, we infected the human iPSC~MNs with *METTL3* or *METTL14*-shRNA (Figure 2F) and revealed a reduction in m⁶A-mRNA levels (Figure S2) with a concomitant neurite degeneration (Figure 2G and 2H).”

3. I am surprised by the use of ChAT-Cre, which affects many neuronal types unrelated to motor neurons. Also this is an undesirable promoter for studying motor neuron disease since it is activated

embryonically (as early as E11.5) - phenotypes seen likely reflect defective differentiation programs. Perhaps this is a different strain than the usual ChAT-Cre (see <https://pubmed.ncbi.nlm.nih.gov/23649862/>) - is this one somehow activated in adulthood? Details are not provided for the claimed late activation so this should be clarified since ChAT is an embryonic promoter. The correct promoter is a motor neuron-specific promoter with adult onset or inducible promoter (e.g. tamoxifen). If this is an embryonic or perinatal promoter, Figure 2 has little relevance to adult motor neuron disease. The use of a developmental promoter is problematic since the m6A pathway regulates development and the phenotypes could be due to impaired development not function of motor neurons. If this is an early promoter, then the invalidity of this promoter should be discussed. The microglial activation was interesting and can be mentioned but again, it is not clear if this is due to improper MN development or death of these neurons caused by defective development.

4. Fig 3 - as with Figure 2, the possibly incorrect Cre driver raises the possibility that all the phenotypes derive from impaired differentiation of motor neurons to altered (but related cell types) rather than proof that m6A is needed to maintain motor neuron function.

We sincerely thank the reviewer for raising these important issues, which are highly relevant to the motor neuron field. The *ChAT-Cre* line from JAX (JAX stock#031661), which we used, is the most commonly utilized driver to delete genes in motor neurons (specified in detail in the Methods section). However, as the reviewer points out, this line is not ideal for targeting adult motor neurons because ChAT expression begins in late embryonic stages. **Currently, there is no Cre driver specifically tailored to adult motor neurons.** While *ChAT-CreER* might offer a potential solution for addressing this limitation, it also presents challenges, such as the inclusion of other non-MN ChAT^{on} cells. Additionally, the low efficiency of tamoxifen-induced recombination in the central nervous system when using CreER remains a concern. To directly address whether the observed spinal motor neuron degeneration is due to developmental or postnatal defects, we conducted the following series of experiments:

1. ***Olig2-Cre; Mettl14^{flxed}* mice (Developmental Motor Neuron Defects).** In Figure S3, we used *Olig2-Cre; Mettl14^{flxed}* mice to investigate potential developmental motor neuron defects. *Olig2-Cre* has been widely recognized, including in our own work (Tung *et al.*, 2015; Tung *et al.*, 2019), as a highly efficient driver for removing floxed alleles in all spinal motor neurons from E9.5. In this context, we found that motor neuron progenitors (Olig2^{on}), generic motor neurons (Mnx1^{on}, Isl1/2^{on}), and all motor neuron columnar types (Foxp1^{on}, Lhx3^{on}) were unaffected in the used *Olig2-Cre; Mettl14^{flxed}* mice. Although *Olig2-Cre; Mettl14^{flxed}* mice died postnatally ~P20, this was likely due to oligodendrocyte defects, not motor neuron developmental defects (Figure S3).
2. ***ChAT-Cre; Mettl14^{flxed}* mice (Postnatal Motor Neuron Degeneration).** To further explore potential postnatal motor neuron defects, we used *ChAT-Cre; Mettl14^{flxed}* mice. In the revised manuscript, we now extensively examine phenotypes across presymptomatic (P30–P70), early symptomatic (P70–P100), disease onset (P100–P120), early onset (P160), and late-onset (P200–P250) stages. In the spinal cords of *ChAT-Cre; Mettl14^{flxed}* mice, the number of ChAT^{on} motor neurons remained similar to control mice during early stages, i.e., until P70. However, starting from P100–P120, we observed a gradual decline in ChAT^{on} motor neurons in the ventral horn, significantly decreasing after P160 (new Figure 3).

Our findings from the two Cre lines suggest that deletion of *Mettl14* does not cause motor neuron developmental defects. However, the function of *Mettl14* appears to become particularly critical in adult motor neurons. These observations lead us to conclude that the degeneration observed in the *ChAT-Cre; Mettl14^{flxed}* mice is very likely due to adult-onset degeneration rather than a developmental defect. We acknowledge that a more refined approach, specifically targeting *Mettl3/14* deletion in adult motor neurons, would be an important experiment for future studies, pending the availability of an appropriate Cre line. We now thoroughly discuss these findings in our revised Discussion section (lines 487-496):

“Although *ChAT-Cre; Mettl14^{flxed}* mice exhibited several ALS-like phenotypes, we were unable to determine whether these effects were specifically attributable to C-boutons synapsing on the soma and proximal dendrites of MNs, the MNs themselves, or both. This is an intriguing question, as recent reports have suggested that spinal inhibitory neurons, including C-boutons, might degenerate before MNs in a mouse model of ALS (Montanana-Rosell *et al.*, 2024; Wells *et al.*, 2021). Currently, the lack of a Cre driver that specifically targets adult MNs limits examining this question. However, our single-nuclei ATAC-seq data offer a promising avenue for identifying enhancers specific to adult cholinergic interneurons (INs) and MNs. Generating Cre lines driven by these enhancers could allow for a more precise dissection of the respective contributions of INs and MNs to the observed phenotypes.”

5. FTO gene therapy is interesting. m6A nanopore maps should be prepared to document increase in m6A site stoichiometry. Otherwise we cannot know any potential mechanism. Fto can also affect m6Am and even m1A and this is cell-type specific so nanopore can be useful to see if in this case if m6A is affected, and on which mRNAs. m6Am immunoblots on small RNA can help to show if m6Am levels increase in these FTO gene therapy treatments and thus constitutes a potential alternative mechanism. Please note that again the m6A assay seems to be on total RNA so cannot be used.

We thank the reviewer for raising this exciting, albeit challenging, suggestion. As noted, several techniques exist for identifying m⁶A sites, each with its own advantages and limitations (Zhang *et al.*, 2023). We chose to use Nanopore direct RNA sequencing due to its ability to identify potential m⁶A sites with single-nucleotide resolution and provide stoichiometric information (Moshitch-Moshkovitz *et al.*, 2022; Zhang *et al.*, 2023). Leveraging this capability, we employed two widely cited supervised machine learning tools, EpiNano and m6Anet, setting a stringent criterion that m⁶A sites must be predicted in at least two samples by either algorithm (see Methods for details). Even with this conservative approach, the two algorithms yielded some divergent results, so we set a stringent criterion that the predicted m⁶A sites needed to occur in at least two samples for either one of the algorithms (new Figure 6). To thoroughly address the reviewers' concerns, we will need to conduct large-scale experiments involving ALS iPSC-derived motor neurons, along with appropriate controls and FTO inhibitor treatments, then perform Nanopore direct RNA sequencing. While these steps are crucial for providing comprehensive answers, they are highly time-consuming, require significant effort, and involve substantial costs. We feel this might extend beyond the scope of the current manuscript. However, we recognize this reviewer's concern regarding potential alternative, non-mutually exclusive mechanisms. To address this issue, we now

clarify these points and acknowledge the limitations of the current approach in the revised manuscript (lines 579-586):

“While we have validated several ALS risk genes with m⁶A-modified sites that are dysregulated in the *Chat-Cre; Mettl14^{flxed}* mice, the direct contribution of these genes as a cohort to MN degeneration in ALS remains to be systematically tested. Furthermore, FTO is not only an m⁶A demethylase, but also functions as a demethylase for the m₂-isoform (*N*⁶,2'-*O*-dimethyladenosine, m⁶Am) in small nuclear RNAs (snRNAs) (Mauer *et al*, 2019) and mRNAs. Although we have demonstrated that FTO inhibition rescues the expression of many ALS pathway- and chromatin regulator-related genes, it remains to be investigated if this therapeutic effect involves m⁶Am-modified snRNAs.”

Alternatively, to further substantiate our observation that many m⁶A-modified ALS risk genes are dysregulated and potentially serve as critical contributors to ALS pathogenesis, we have incorporated several new experiments in this revised manuscript:

1. We re-performed the m⁶A ELISA on purified, polyA-enriched mRNA following rRNA depletion (new Figures 1E, F, G. and 9C). This approach revealed a significant and marked decline in m⁶A levels across nearly all familial ALS iPSC~MNs upon degeneration.
2. We also provide the stoichiometric data of our MN m⁶A epitranscriptome analysis (new Figures 6D and Table S3) and validated that a set of novel m⁶A-modified ALS risk genes predicted through Nanopore sequencing—including *Tardbp*, *Atp13a2*, *Dctn1*, *Epha4*, *C9orf72*, *Glt8d1*, *Cacna1h*, *Chrna3*, *Bscl2*, *Fig4*, *Hnrnpa2b1*, *Ubqln2*, *Hnrnpa1*, *Tuba4a*, *Sod1*, *Chmp2b*, and *PIKfyve* are indeed m⁶A-enriched via an m⁶A pull-down assay (Figures 6I, 6J and new Figure S6B). Our findings demonstrate that most ALS risk genes are m⁶A-modified and dysregulated in *Sun1^{sfGFP}; Chat-Cre; Mettl14^{flxed}* (new Figure 7H). Notably, following FTO inhibitor treatment, these genes were restored to levels comparable to controls (new Figure 9E).
3. To determine if the neuroprotective effects of the FTO inhibitor on MN degeneration in ALS are mediated through its regulation of m⁶A-modified ALS risk genes (new Figures 7G and 7H), we performed RNA-seq analysis on ALS iPSC-derived MNs treated with FB23-2. Differential expression analysis revealed that following FB23-2 treatment, several m⁶A-modified genes involved in synaptic function, RNA metabolism, and chromatin and histone modifications were restored to levels similar to controls (new Figure S11). Notably, the expression of multiple m⁶A-modified ALS risk genes was returned to control-like levels (vehicle-treated), suggesting that FTO inhibition may mitigate MN degeneration in ALS by modulating m⁶A-modified gene expression (new Figure 9E). These findings indicate that enhancing m⁶A levels in ALS iPSCs using small molecules could help restore the balance of the m⁶A epitranscriptome and preserve MN integrity.
4. Additionally, we have now performed additional experiments to address that FTO inhibitor treatment is likely to render the m⁶A-modified genes to restore motor neuron degeneration for the revised manuscript (new Figure 11 and Figure S13). Specifically, we have assessed

H3K9me3 and γ H2AX for four groups—wild type control (Ctrl), Ctrl with scAAV9-sh*Fto* (Ctrl; scAAV9-sh*Fto*), *SOD1*^{G93A}, and *SOD1*^{G93A} with scAAV9-sh*Fto* (*SOD1*^{G93A}; scAAV9-sh*Fto*)—at postnatal day 140 (P140), i.e., when *SOD1*^{G93A} mice are still at the early symptomatic stage (new Figure 11). Strikingly, we observed a drastic dramatic increase in HK9me3 in the ventral horn of the spinal cord in *SOD1*^{G93A} mice, which was rendered to the level manifested by Ctrl mice after scAAV9-sh*Fto* treatment (new Figure 11). This result supports that epigenetic abnormalities might be a novel feature of ALS pathogenesis. However, we did not observe any change in γ H2AX in the spinal cords of *SOD1*^{G93A} mice (new Figure S13). We explain this outcome in new sections of the revised Discussion. We agree with the reviewer that investigating if these epigenetic abnormalities are common features of ALS pathogenesis in human iPSC~MNs or patient tissues is an important issue. However, the extensive analysis required to undertake this task may exceed the scope of the current study. Such an investigation could potentially pave the way for new avenues of future research.

We hope our new experiments and revisions sufficiently address the reviewer's concerns.

6. For chromatin data, please address whether the change in chromatin accessibility is in m6A genes, or if it is unrelated to m6A-containing transcripts.

We thank the reviewer for this insightful question. To address this point, we examined if genes associated with accessible chromatin regions are m⁶A-modified. We determined gene-peak association with ArchR and m⁶A status employing our Nanopore RNA-seq data. In skeletal MNs, among the genes linked to regions of differential chromatin accessibility following *Mettl14* ablation (1,078 peaks and 894 genes), we found that 47 % (419 genes) were m⁶A-modified and 17 % (152 genes) were both m⁶A-modified and differentially expressed. In visceral MNs, 390 differentially accessible peaks were linked to 340 genes, with 53 % (180 genes) of those genes being m⁶A-modified, and 11 % (36 genes) being both m⁶A-modified and differentially expressed. Furthermore, in cholinergic INs, 55 % (35 genes) of the genes were m⁶A-modified, and 5 % (3 genes) were both m⁶A-modified and differentially expressed (from a total of 64 genes linked to differentially accessible peaks) (new Table S6).

We have now added this analysis as new Figure S10D and in the Results section, and we discuss the implications of our new findings (lines 345-359) as follows:

“Subsequently, we integrated and scrutinized the snRNAseq and snATACseq data, revealing that only a modest subset of DEGs align with alterations in chromatin accessibility (Figure S9F). To further investigate the relationship between changes in chromatin accessibility and dysregulated gene expression, we first identified linked peak-to-gene associations and performed a correlation analysis between gene expression and changes in chromatin accessibility following *Mettl14* ablation (Figure S10D). Our analysis revealed a low correlation between these two factors. Notably, only 17 %, 11 %, and 5 % of the genes associated with regions of differential chromatin accessibility for skeletal MNs, visceral MNs, and cholinergic neurons, respectively, were both m⁶A-modified and differentially expressed upon *Mettl14* ablation. These discoveries underscore the critical importance of preserving a nuanced equilibrium in the m⁶A transcriptome within adult

MNs to maintain neuronal homeostasis. Diminished m⁶A levels may lead to compromised expression of pivotal neuronal and disease-associated genes governed by versatile regulatory mechanisms, i.e., either through direct modification of m⁶A-affected transcripts or by reshaping the chromatin landscape within MNs.”

We discuss these new analyses as follows (lines 545-561):

“In our proposed model (depicted in Figure 12B), we postulate that the aberrant down-regulation of the two m⁶A methyltransferases, namely METTL3 and METTL14, in ALS reflects an anomaly likely associated with the natural aging process compounded by unidentified exacerbating factors (Castro-Hernandez *et al.*, 2023). This reduced expression of m⁶A methyltransferases has multifaceted implications. Firstly, it instigates widespread dysregulation across the mRNA transcriptome, notably enriching pathways associated with ALS risk genes, synaptic activity, and neuronal functional pathways. Simultaneously, m⁶A modification of chromatin regulators, crucial for modulating heterochromatin repression, is diminished. This global dysregulation of m⁶A-mediated epitranscriptomic processes disrupts RNA stability and perturbs the repressive chromatin landscape, potentially exacerbating the degeneration of MNs in the context of ALS. Strikingly, we observed a significant increase in closed chromatin regions in the *Chat-Cre; Mettl14^{flxed}* mice, with only a small fraction of these chromatin changes correlating with dysregulated gene expression. Given that previous studies have shown FTO-mediated LINE1 RNA m⁶A demethylation plays a role in regulating chromatin state and gene expression during mouse oocyte and embryonic development (Wei *et al.*, 2022), and that retrotransposon reactivation has been observed in some ALS postmortem tissues (Tam *et al.*, 2019), it would be intriguing to explore in future studies whether Mettl14-mediated m⁶A impairment leads to retrotransposon reactivation through its interaction with chromatin state.

7. The m6A maps from nanopore are interesting, but seem to be identical to other neuronal maps of m6A made using other methods. It is not clear if any new data is obtained from these maps. Greater clarity would help on this. What is new relative to other studies of m6A in motor neurons/ALS?

We appreciate this reviewer’s critique. As reviewer #2 also suggested elaborating further on our Nanopore direct RNA-seq data, we decided to add a new figure for clarity. We have revised the manuscript as follows to address the reviewer’s concern (lines 229-267):

“To gain insights into the potential mechanisms underlying how hypo-m⁶A promotes MN degeneration, we aimed to systematically identify the dysregulated genes possessing m⁶A modifications in our *Chat-Cre; Mettl14^{flxed}* mice (Figures 6 and 7). To identify m⁶A-modified transcripts, we adopted a direct RNA sequencing platform, which enables the identification of the MN m⁶A epitranscriptome at single-nucleotide resolution (Figure 6). Since it is technically challenging to obtain sufficient adult MNs from the spinal cord for direct RNA sequencing, we employed an enhanced method using mouse ESC-derived MNs, matured with a conditioned medium (Figures 6A~6C) (Patel *et al.*, 2022; Tung *et al.*, 2019). First, we confirmed that this approach successfully generated MNs expressing mature neuronal markers, with longer and more mature neurite structures revealed by discrete Syn1-positive puncta (Figures 6B and 6C). We then

subjected these mature MNs to the ONT Nanopore platform, which provides a powerful framework for detecting RNA modifications through advanced machine-learning algorithms applied to sequencing metrics (Figure 6D, details in Methods). As expected, several adult mature MN genes are abundantly expressed in the Nanopore direct RNA-seq results (Figure 6E). We subsequently employed two supervised machine learning tools, namely EpiNano and m6Anet (Hendra *et al.*, 2022; Liu *et al.*, 2021), which set a stringent criterion that the predicted m⁶A sites need to occur in at least two samples for either one of the algorithms (see Methods for details). We identified 30,340 high-confidence m⁶A modification sites (probability > 0.5) corresponding to 7,921 genes (Figure 6D, genes of interest are illustrated in Table S1; refer to the Methods section for details). Interestingly, a deeper estimate of the predicted m⁶A stoichiometry indicated that the high variation levels are displayed across different m⁶A sites in the same gene (Table S3). Consistent with previous findings (Castro-Hernandez *et al.*, 2023), our analysis also revealed enriched distributions of m⁶A sites in coding sequences (CDSs) and 3' UTRs, especially near the stop codons (Figure 6F). By analyzing the enrichment of Gene Ontology (GO) and KEGG pathways for the m⁶A-modified transcripts, we noticed a striking enrichment for ALS-related genes (Figure 6G). In addition, among the 81 identified ALS risk genes to date, our direct RNA sequencing demonstrated that 34 of them are m⁶A-modified ($p = 2.06 \times 10^{-6}$, one-tailed hypergeometric test; Figure 6H and Table S2).

To validate our computationally predicted m⁶A sites, we used *Tardbp* (*Tdp43*) as a benchmark to validate our methodology. Consistent with a previous report (McMillan *et al.*, 2023), we confirmed the existence of a previously identified m⁶A site in *Tardbp* with a high probability rate with stoichiometry ranging from 57.38 to 92.53 % (Table S3) and verified via m⁶A antibody pull-down assay (Figure 6I, site 1). Additionally, we uncovered and validated an additional high m⁶A-modified site within the *Tardbp* transcript from Nanopore direct RNA-seq (Figure 6I, site 2). Moreover, we substantiated the existence of predicted m⁶A sites in *Atp13a2* (*Park9*) (Figure 6J) (Spataro *et al.*, 2019), an ALS risk gene not previously shown to have m⁶A modifications in MNs. Finally, we further verified several newly identified m⁶A-modified sites in ALS risk genes, including *Dctn1*, *Epha4*, *C9orf72*, *Glt8d1*, *Cacna1h*, *Chrna3*, *Bscl2*, *Fig4*, *Hnrnpa2b1*, *Ubqln2*, *Hnrnpa1*, *Tuba4a*, *Sod1*, *Chmp2b*, and *PIKfyve* (Figure S6B). Thus, these observations confirm the sensitivity, accuracy, and reliability of Nanopore technology to identify m⁶A-modified sites in MNs.”

From the above-described analyses, we provide several new insights into the mature MN m⁶A epitranscriptome:

1. In two previous studies (Dermentzaki *et al.*, 2024; Li *et al.*, 2023), *Tardbp* and *C9orf72* were identified as key m⁶A-modified transcripts involved in motor neuron degeneration. In this manuscript, not only validated these m⁶A sites, but also identified several novel ALS risk genes with newly discovered m⁶A-modified sites, including *Atp13a2*, *Dctn1*, *Epha4*, *C9orf72*, *Glt8d1*, *Cacna1h*, *Chrna3*, *Bscl2*, *Fig4*, *Hnrnpa2b1*, *Ubqln2*, *Hnrnpa1*, *Tuba4a*, *Sod1*, *Chmp2b*, and *PIKfyve* (new Figure 6 and S6B).
2. Furthermore, we observed that these newly identified m⁶A-modified sites are dysregulated in the *ChAT-Cre; Mettl14^{flox}* mice, implying potential functional roles in ALS pathology (new Figure 7E~7H, and S9).
3. Finally, we have identified additional m⁶A-modified transcripts that may contribute to ALS (full list in Table S1). These include genes related to RNA splicing (e.g., *Malat1*, *Srsf2*,

Fus, Srekl, Hnrnp1, Tra2a), members of the ATP-dependent Chd family, and other genes like *Bcl7c, Ncoa6, and Ube2b*, which have been implicated in the DNA damage response and apoptosis (Figure S9).

We hope that the new experiments and revisions sufficiently address the reviewer's concerns, and again, we thank all of the reviewers for their sharp observations and insightful critiques.

Castro-Hernandez R, Berulava T, Metelova M, Epple R, Pena Centeno T, Richter J, Kaurani L, Pradhan R, Sakib MS, Burkhardt S *et al* (2023) Conserved reduction of m(6)A RNA modifications during aging and neurodegeneration is linked to changes in synaptic transcripts. *Proc Natl Acad Sci U S A* 120: e2204933120

Dasen JS, De Camilli A, Wang B, Tucker PW, Jessell TM (2008) Hox repertoires for motor neuron diversity and connectivity gated by a single accessory factor, FoxP1. *Cell* 134: 304-316

Helm M, Alfonzo JD (2014) Posttranscriptional RNA Modifications: playing metabolic games in a cell's chemical Legoland. *Chem Biol* 21: 174-185

Hendra C, Pratanwanich PN, Wan YK, Goh WSS, Thiery A, Goke J (2022) Detection of m6A from direct RNA sequencing using a multiple instance learning framework. *Nat Methods* 19: 1590-1598

Hess ME, Hess S, Meyer KD, Verhagen LA, Koch L, Bronneke HS, Dietrich MO, Jordan SD, Saletore Y, Elemento O *et al* (2013) The fat mass and obesity associated gene (Fto) regulates activity of the dopaminergic midbrain circuitry. *Nat Neurosci* 16: 1042-1048

Huang Y, Su R, Sheng Y, Dong L, Dong Z, Xu H, Ni T, Zhang ZS, Zhang T, Li C *et al* (2019) Small-Molecule Targeting of Oncogenic FTO Demethylase in Acute Myeloid Leukemia. *Cancer Cell* 35: 677-691 e610

Kraemer BR, Snow JP, Vollbrecht P, Pathak A, Valentine WM, Deutch AY, Carter BD (2014) A role for the p75 neurotrophin receptor in axonal degeneration and apoptosis induced by oxidative stress. *J Biol Chem* 289: 21205-21216

La Cognata V, Gentile G, Aronica E, Cavallaro S (2020) Splicing Players Are Differently Expressed in Sporadic Amyotrophic Lateral Sclerosis Molecular Clusters and Brain Regions. *Cells* 9

Lence T, Akhtar J, Bayer M, Schmid K, Spindler L, Ho CH, Kreim N, Andrade-Navarro MA, Poeck B, Helm M *et al* (2016) m(6)A modulates neuronal functions and sex determination in *Drosophila*. *Nature* 540: 242-247

Liu H, Begik O, Novoa EM (2021) EpiNano: Detection of m(6)A RNA Modifications Using Oxford Nanopore Direct RNA Sequencing. *Methods Mol Biol* 2298: 31-52

Maor-Nof M, Shipony Z, Lopez-Gonzalez R, Nakayama L, Zhang YJ, Couthouis J, Blum JA, Castruita PA, Linares GR, Ruan K *et al* (2021) p53 is a central regulator driving neurodegeneration caused by C9orf72 poly(PR). *Cell* 184: 689-708 e620

Mauer J, Sindelar M, Despic V, Guez T, Hawley BR, Vasseur JJ, Rentmeister A, Gross SS, Pellizzoni L, Debart F *et al* (2019) FTO controls reversible m(6)Am RNA methylation during snRNA biogenesis. *Nat Chem Biol* 15: 340-347

Maury Y, Come J, Piskorowski RA, Salah-Mohellibi N, Chevaleyre V, Peschanski M, Martinat C, Nedelec S (2015) Combinatorial analysis of developmental cues efficiently converts human pluripotent stem cells into multiple neuronal subtypes. *Nat Biotechnol* 33: 89-96

McMillan M, Gomez N, Hsieh C, Bekier M, Li X, Miguez R, Tank EMH, Barmada SJ (2023) RNA methylation influences TDP43 binding and disease pathogenesis in models of amyotrophic lateral sclerosis and frontotemporal dementia. *Mol Cell* 83: 219-236 e217

Montanana-Rosell R, Selvan R, Hernandez-Varas P, Kaminski JM, Sidhu SK, Ahlmark DB, Kiehn O, Allodi I (2024) Spinal inhibitory neurons degenerate before motor neurons and excitatory neurons in a mouse model of ALS. *Sci Adv* 10: eadk3229

Moshitch-Moshkovitz S, Dominissini D, Rechavi G (2022) The epitranscriptome toolbox. *Cell* 185: 764-776

Nedialkova DD, Leidel SA (2015) Optimization of Codon Translation Rates via tRNA Modifications Maintains Proteome Integrity. *Cell* 161: 1606-1618

Patel T, Hammelman J, Aziz S, Jang S, Closser M, Michaels TL, Blum JA, Gifford DK, Wichterle H (2022) Transcriptional dynamics of murine motor neuron maturation in vivo and in vitro. *Nat Commun* 13: 5427

Ruud J, Alber J, Tokarska A, Engstrom Ruud L, Nolte H, Biglari N, Lippert R, Lautenschlager A, Cieslak PE, Szumiec L *et al* (2019) The Fat Mass and Obesity-Associated Protein (FTO) Regulates Locomotor Responses to Novelty via D2R Medium Spiny Neurons. *Cell Rep* 27: 3182-3198 e3189

Spataro R, Kousi M, Farhan SMK, Willer JR, Ross JP, Dion PA, Rouleau GA, Daly MJ, Neale BM, La Bella V *et al* (2019) Mutations in ATP13A2 (PARK9) are associated with an amyotrophic lateral sclerosis-like phenotype, implicating this locus in further phenotypic expansion. *Hum Genomics* 13: 19

Tam OH, Rozhkov NV, Shaw R, Kim D, Hubbard I, Fennessey S, Propp N, Consortium NA, Fagegaltier D, Harris BT *et al* (2019) Postmortem Cortex Samples Identify Distinct Molecular Subtypes of ALS: Retrotransposon Activation, Oxidative Stress, and Activated Glia. *Cell Rep* 29: 1164-1177 e1165

Tchasovnikarova IA, Timms RT, Matheson NJ, Wals K, Antrobus R, Gottgens B, Dougan G, Dawson MA, Lehner PJ (2015) GENE SILENCING. Epigenetic silencing by the HUSH complex mediates position-effect variegation in human cells. *Science* 348: 1481-1485

- Thams S, Lowry ER, Larraufie MH, Spiller KJ, Li H, Williams DJ, Hoang P, Jiang E, Williams LA, Sandoe J *et al* (2019) A Stem Cell-Based Screening Platform Identifies Compounds that Desensitize Motor Neurons to Endoplasmic Reticulum Stress. *Mol Ther* 27: 87-101
- Tung YT, Lu YL, Peng KC, Yen YP, Chang M, Li J, Jung H, Thams S, Huang YP, Hung JH *et al* (2015) Mir-17 approximately 92 Governs Motor Neuron Subtype Survival by Mediating Nuclear PTEN. *Cell Rep* 11: 1305-1318
- Tung YT, Peng KC, Chen YC, Yen YP, Chang M, Thams S, Chen JA (2019) Mir-17 approximately 92 Confers Motor Neuron Subtype Differential Resistance to ALS-Associated Degeneration. *Cell Stem Cell* 25: 193-209 e197
- Van Harten ACM, Phatnani H, Przedborski S (2021) Non-cell-autonomous pathogenic mechanisms in amyotrophic lateral sclerosis. *Trends Neurosci* 44: 658-668
- Wei J, Yu X, Yang L, Liu X, Gao B, Huang B, Dou X, Liu J, Zou Z, Cui XL *et al* (2022) FTO mediates LINE1 m(6)A demethylation and chromatin regulation in mESCs and mouse development. *Science* 376: 968-973
- Wells TL, Myles JR, Akay T (2021) C-Boutons and Their Influence on Amyotrophic Lateral Sclerosis Disease Progression. *J Neurosci* 41: 8088-8101
- Yankova E, Blackaby W, Albertella M, Rak J, De Braekeleer E, Tsagkogeorga G, Pilka ES, Aspris D, Leggate D, Hendrick AG *et al* (2021) Small-molecule inhibition of METTL3 as a strategy against myeloid leukaemia. *Nature* 593: 597-601
- Zhang F, Ignatova VV, Ming GL, Song H (2023) Advances in brain epitranscriptomics research and translational opportunities. *Mol Psychiatry*

We sincerely thank the reviewers for appreciating the importance of our study and their insightful comments. These point-by-point responses to their comments and descriptions that we have provided in this letter. The original reviewers' comments are in **black**, followed by our responses in **blue**. We highlight changes made to the text and figures in **yellow**, whereas **cyan** indicates the paragraph line numbers in the revised text. All changes are highlighted in the revised text.

Reviewer #1 (Remarks to the Author):

Most of the previous concerns we raised were successfully addressed. One additional point is that the methods section does not describe the conditions and concentration of FTO treatment, which are crucial details for future clinical applications.

We sincerely appreciate the reviewer's valuable suggestion. Initially, we included the description of the FTO treatment procedure within the human iPSC differentiation section of the Methods. In the revised manuscript, we have extracted this information and provided a more detailed description of the FTO treatment conditions in a separate, dedicated section within the Methods. (lines 633-642):

FTO inhibitor treatments

To accelerate MN degeneration, the CultureOne Supplement medium of day 31 MN culture was replaced with N2/B27 medium only (no BDNF and GDNF), for which CPA was supplemented to accelerate degeneration for another seven days. FB23-2 is a small molecule that inhibits the demethylase activity to elevate m⁶A levels. FB23-2 (1nM) was treated together with the CPA (50 μM) for 15~30 days (change the small molecules and medium every two days) in ALS iPSC~MNs with the increased mRNA m⁶A methylation level and extended disease onset or rescue of the degenerative process. Then MNs were revealed by SMI32 immunostaining and captured by an ImageXpress® Micro XLS High-Content Imaging System (Molecular Devices). The degeneration index was calculated at the indicated time point after CPA treatment.

Reviewer #2 (Remarks to the Author):

The authors have done an outstanding job in addressing all the comments I raised. The revised manuscript is now much improved and should be suitable for publication in Nature Communications.

We very much appreciate this reviewer's positive support.

Reviewer #3 (Remarks to the Author):

I co-reviewed this manuscript with one of the reviewers who provided the listed reports. This is part of the Nature Communications initiative to facilitate training in peer review and to provide

appropriate recognition for Early Career Researchers who co-review manuscripts.

Reviewer #4 (Remarks to the Author):

The manuscript addresses most of the key issues. It is unfortunate that the nanopore on the FTO-treated neurons could not be done.

My remaining comments are textual in nature:

1. The m⁶A ELISA method that the authors use to say that m⁶A is augmented does not distinguish between m⁶A and m⁶Am. The claim that augmented m⁶A is occurring is difficult to validate without using mass spectrometry. I would make sure that the authors clarify the limitations of this ELISA method, which is particularly important since m⁶Am can be changing, not m⁶A or not just m⁶A.

We sincerely appreciate the reviewer's critical comments. We recognize the limitations of the ELISA method and previously addressed the potential use of LC-MS in future studies in our revised manuscript. To further clarify this limitation, we have now explicitly discussed the challenge of distinguishing between m⁶A and m⁶Am (lines 531-536).

"To detect m⁶A levels, different assays have been used by our study and those of others, including antibody-based pull down, dot blot, ELISA, and microarray method ¹. We have further applied Nanopore ONT direct RNAseq to uncover m⁶A stoichiometry, which appears to provide better sensitivity. Each of these methods has advantages and disadvantages in terms of the sensitivity and feasibility of large-scale studies ¹. We advocate screening m⁶A levels in a larger cohort of sALS cases using advanced techniques such as direct Mass Spectrometry in the future, as this approach will enable a more precise distinction between m⁶A and m⁶Am.

2. In many cases, the authors say things like "In addition, among the 81 identified ALS risk genes to date, our direct RNA sequencing demonstrated that 34 of them are m⁶A-modified". This might appear to be quite a lot of genes. But in fact, this could be evidence of m⁶A -not- being enriched in ALS risk genes. In many mapping studies 50% or more mRNAs have at least one m⁶A. So these numbers that the authors provide need to be provided along with the prevalence of m⁶A in their maps. For example, they could say something like "In our dataset only 15% of transcripts contained m⁶A, but 40% of ALS risk genes contain m⁶A (34 out of 81)." In every case when the authors point out enrichments or the reasons why m⁶A might mediate the effects of FTO inhibition, this sort of context should be provided so the reader can understand the significance of the results.

We appreciate the reviewer's suggestions and apologize for not providing a clearer explanation. To make this point clearer, we have revised the Results sections to address this (lines 249-252):

In addition, although only 35.56 % of transcripts (7,921 out of 22,280) contained m⁶A in our dataset, among the 81 identified ALS risk genes to date, our direct RNA sequencing demonstrated that 41.98 % of ALS risk genes (34 out of 81) are m⁶A-modified (P = 2.06e-6, one-tailed hypergeometric test; Fig. 5h and Supplementary Data 2).

3. The title says "augmented m6A mitigates symptoms". But the more correct statement is "FTO depletion" (or inhibition) mitigates the symptoms, since this is what was tested. As mentioned above, the measurements of increased m6A were not done in a way that reflects m6A only, and the authors did not do the nanopore experiment. Mettl3 overexpression was not used, which could have supported this claim (if m6A also increased). The exact mechanism of FTO inhibition is often unclear, since FTO targets so many nucleotides. So the authors can be more precise by saying FTO inhibition mitigates these phenotypes.

We appreciate the reviewer's feedback and agree that comprehensive sequencing, such as Nanopore, following m⁶A restoration would be ideal to substantiate the role of augmented m⁶A in mitigating ALS symptoms. However, in addition to our current approach, we have also tested another small-molecule inhibitor of ALKBH5, an alternative m⁶A eraser, in degenerated ALS iPSC-derived motor neurons. While the data are not included in this manuscript due to its extensive scope, our results demonstrate that this treatment restores m⁶A levels and rescues degeneration. These findings further support the potential therapeutic relevance of m⁶A augmentation, though the precise mechanisms require further investigation. We hope our explanation adequately addresses the reviewer's concerns. Given the enriched data provided by this manuscript and several recent published studies^{2,3} to suggest the m⁶A change is implied in ALS, we prefer to use the original title and think it is appropriate. However, we are open to the change and will leave this for the editor to make a final call.

Reference

- 1 Cerneckis, J., Ming, G. L., Song, H., He, C. & Shi, Y. The rise of epitranscriptomics: recent developments and future directions. *Trends Pharmacol Sci* **45**, 24-38, doi:10.1016/j.tips.2023.11.002 (2024).
- 2 Li, Y. *et al.* Globally reduced N6-methyladenosine (m6A) in C9ORF72-ALS/FTD dysregulates RNA metabolism and contributes to neurodegeneration. *Nature Neuroscience* **26**, 1328-1338, doi:10.1038/s41593-023-01374-9 (2023).
- 3 McMillan, M. *et al.* RNA methylation influences TDP43 binding and disease pathogenesis in models of amyotrophic lateral sclerosis and frontotemporal dementia. *Mol Cell* **83**, 219-236 e217, doi:10.1016/j.molcel.2022.12.019 (2023).